# Embryonic signatures of intergenerational epigenetic inheritance across paternal environments and genetic backgrounds

Mathilde Dura [1,4], Bobby Ranjan [1,2,4], Joana B Serrano [1], Rossella Paribeni[1], Violetta Paribeni[1], Laura Villacorta[3], Vladimir Benes [3], Olga Boruc[1], Ana Boskovic [1✉] & Jamie A Hackett [1✉]

## Abstract

The paternal preconception environment has been implicated as a modulator of phenotypic traits and disease risk in F1 offspring. However, the prevalence and mechanisms of such intergenerational epigenetic inheritance (IEI) in mammals remain poorly defined. Moreover, the interplay between paternal exposure, genetics, and age on emergent offspring features is unexplored. Here, we measure the quantitative impact of three paternal environments on early embryogenesis across genetic backgrounds. Using in vitro fertilisation (IVF) at scale, we capture batch-robust transcriptomic signatures of IEI with single-blastocyst resolution. Amongst these, paternal gut microbiota dysbiosis is linked with aberrant expression of (extra-)embryonic lineage regulators in blastocysts. In contrast, a paternal low-protein high-sugar diet associates with subtle preimplantation growth effects. We further identify gene expression variability as a paternally induced F1 phenotype, and highlight confounding issues for IEI, such as batch effects and undersampling. Finally, while genetic background dominantly modifies the inherited signature of paternal environment, aged fathers universally impact F1 expression programmes across genetic contexts. This study systematically characterises how paternal conditioning programs subtle but detectable molecular responses in early embryos, and proposes guiding principles to dissect intergenerational phenomenology.

Keywords Epigenetic Inheritance; Gut Microbiome; GxE; Development; Phenotypic Variation
Subject Categories Chromatin, Transcription & Genomics; Development

## Introduction

Inheritance of biological material entails the transmission of genetic and non-genetic information across generations. In mammals, mature sperm transfer a genomic DNA sequence to oocytes during fertilisation, which serves as the fundamental basis of paternal inheritance (Watson and Crick, 1953). In addition, sperm also transmit 'epigenetic' information in the form of small RNA payloads, chromatin states, DNA methylation, endocrine signals and metabolites (Adrian-Kalchhauser et al, 2020; Fitz-James and Cavalli, 2022; Skvortsova et al, 2018). These non-genetic molecular factors have the potential to influence early embryonic genome regulation, and thus to affect development and ultimately $F_1$ phenotype. Moreover, epigenetic information in the germline is relatively dynamic compared to the DNA sequence, with the potential to change in response to extrinsic cues and/or environmental perturbations (Ciabrelli et al, 2017; Murphy et al, 2020; Rechavi et al, 2014; Torres-Garcia et al, 2020). As a consequence, an increasing number of mammalian studies report that fathers exposed to adverse pre-conception environments transmit phenotype(s) to their $F_1$ offspring, referred to as intergenerational epigenetic inheritance (IEI). Such intergenerational epigenetic effects that traverse a single generation are emerging as important but overlooked contributors to phenotypic variation and disease susceptibility, yet remain poorly understood.

Multiple paternal environments have been reported to propagate offspring effects, including dietary perturbations (high-fat, low-protein, folate-deficiency), drug exposure (smoking, antidiabetics), gut dysbiosis (antibiotics) and other extrinsic stressors (trauma, endocrine disrupting chemicals) (Argaw-Denboba et al, 2024; Carone et al, 2010; Chen et al, 2016; Huypens et al, 2016; Lambrot et al, 2013; Lesch et al, 2019; Radford et al, 2014; Tomar et al, 2024; Vallaster et al, 2017). Amongst these, perturbations to the gut microbiome of prospective fathers has been shown to influence the birthweight and mortality rate of offspring (Argaw-Denboba et al, 2024; Masson et al, 2024). This effect occurs probabilistically and manifests as an increased risk of adverse $F_1$ outcomes rather than a deterministic response. Mechanistically, dysbiotic fathers affect offspring by inducing *in utero* placental insufficiency, indicating the origin of emergent F1 phenotypes can be traced back to early embryogenesis. Indeed, prior studies found that offspring of fathers exposed to a low-protein high-sugar (LPHS) diet exhibit altered expression of hepatic lipid-associated genes and metabolic dysfunction (Carone et al, 2010). This was linked with paternally-induced changes in MERVL expression as

[1]European Molecular Biology Laboratory, Epigenetics and Neurobiology Unit, via Ramarini 32, Rome 00015, Italy. [2]Collaboration for Joint PhD degree between EMBL and Heidelberg University, Faculty of Biosciences, Heidelberg, Germany. [3]European Molecular Biology Laboratory, Meyerhofstraße 1, Heidelberg 69117, Germany. [4]These authors contributed equally: Mathilde Dura, Bobby Ranjan. ✉E-mail: ana.boskovic@embl.it; jamie.hackett@embl.it

early as preimplantation development (Sharma et al, 2016). It is now crucial to obtain a fuller understanding of the presentation of molecular signatures in early embryos, towards dissecting the underlying mechanisms.

IEI does not occur *in silo*, and numerous factors can affect its propagation and/or manifestation. For instance, environmental exposures are interconnected, raising the question of whether each exposure produces a distinct outcome or whether they funnel into a limited number of generic $F_1$ responses. For example, antibiotics and LPHS administration both impact gut microbiome composition (Ross et al, 2024), but also exert treatment-specific effects on fathers and offspring (Argaw-Denboba et al, 2024; Carone et al, 2010). Beyond this bandwidth of potential $F_1$ responses, epigenetic phenomena are tightly coupled with genetics, implying that IEI might be contingent on the underlying DNA sequence (Cavalli and Heard, 2019). Consistently, phenotypic traits in individuals stem from interactions between their genetics and environment (GxE), and this multi-factorial relationship likely also modifies the signature and prevalence of IEI, yet it is poorly understood. Indeed, there is currently limited understanding of precisely which epigenetic factors are responsible for transmitting IEI phenotypes and their molecular manifestation in embryos (Santilli and Boskovic, 2023). One hypothesis is that subtle perturbations introduced by exposed sperm into fertilised eggs snowball through successive stages of development, ultimately reaching a threshold that triggers physiological abnormalities. In order to disentangle these complex relationships, a high-throughput quantitative readout is necessary to capture the earliest molecular signatures of mammalian IEI across diverse GxE contexts.

To this end, we designed a large-scale multi-paradigm study profiling intergenerational responses in preimplantation embryos across paternal exposures, ages and genetic backgrounds. We used in vitro fertilisation (IVF) coupled with high-throughput transcriptomic profiling of preimplantation blastocysts to ask the following questions: (i) Can we capture consistent and reproducible molecular signatures of IEI, (ii) Are these signatures common or specific to paternal exposure, and do they imply mechanisms/outcomes, (iii) How do paternal genetic background and age influence these signatures, and (iv) Are $F_1$ signatures detectable upstream in the paternal reproductive system?

# Results

## An IVF-based multi-paradigm model of paternal effects

We first sought to establish a tractable experimental model that could capture the earliest molecular signatures of paternal environments on progeny. We designed a multi-paradigm IVF-strategy with single-embryo transcriptomics, using sperm from males exposed to distinct environmental factors (Fig. 1A). IVF enables replicates of independent embryos to be generated at scale within a highly controlled context, across multiple parallel batches, thus increasing power whilst controlling for experimental variables. Specifically, our IVF strategy rules out confounders from natural matings such as maternal resource allocation based on judgement of male quality, indirect effects of paternal perturbation on females (e.g. via coprophagy), and/or litter-size effects (Bohacek and Mansuy, 2017). As paternal environment perturbations, we chose

to focus on factors previously linked with inducing intergenerational postnatal phenotype(s), by exposing prospective fathers to: (i) non-absorbable antibiotics (nABX) to induce gut microbiome dysbiosis, (ii) low-protein high-sugar (LPHS) isocaloric western-style diet, and (iii) control standard diet (CON). Male FVB mice were thus exposed to nABX, LPHS or control diet/water *ad libitum* from weaning for 6–7 weeks (Fig. 1A).

We first scored the direct impact of environmental factors on male physiology. Neither nABX nor LPHS impacted survival, testis size or overt phenotypes after 6 weeks of exposure (Fig. EV1A,B). We did however note a reciprocal trend towards increased (nABX) and decreased (LPHS) body mass, which reached significance during the course of treatment (Figs. 1B and EV1C). Moreover, we found major responses within the gastrointestinal tract. Specifically, gut dysbiosis induced by nABX was associated with significantly increased caecum size and weight, whereas in contrast, LPHS diet led to a reduction in caecum weight (Figs. 1C and EV1D,E). Such effects imply the two environmental paradigms have at least partially reciprocal impacts on systemic physiology in prospective fathers.

We next asked how paternal exposures impact fertilisation efficacy. High-quality sperm from treated and control males were isolated after swim-up, and IVF was performed from a randomised common pool of strain-matched oocytes for all sperm donors. All IVFs were performed in parallel on the same morning to minimise intra-batch variability. The resulting zygotes were cultured ex vivo, with no differences observed in the rate of blastocyst development or hatching across the paternal conditions (Fig. 1D). While blastocysts from all paternal treatments appeared morphologically indistinguishable, using quantitative whole-embryo flow analysis, we noted a small but significant decrease in the size of blastocysts derived from LPHS-exposed males, pointing to a potential growth defect (Figs. 1E and EV1F–H). In summary, we established a multi-paradigm paternal exposure model coupled with controlled and scalable ex vivo readouts of early $F_1$ effects. We observe direct physiological responses in exposed males, but no gross impact on fertility or rates of $F_1$ embryonic development.

## Intergenerational molecular signatures of paternal environments

To investigate whether the impacts of paternal environment could manifest as an altered molecular profile in early embryos, we undertook single-embryo transcriptome profiling. To capture the earliest effects, we assayed blastocysts, which represent the first developmental stage after embryonic and extra-embryonic lineages are established, can be cultured at scale, and can be accurately staged. These considerations minimise transcriptomic variation related to rapid developmental transitions during cleavage divisions, whilst enabling putative effects on early lineages to manifest, thus making blastocysts ideal to capture robust and consistent signatures of IEI. Individual blastocysts derived from the three paternal conditions were sequenced with Smart-seq, yielding high-quality, high-coverage transcriptomes from 157 embryos, each with 5–20 million reads (Fig. EV1I–K). The male-female ratio was broadly even across conditions (Fig. EV1L).

Global analysis of all expressed genes by principal component analysis (PCA) revealed that blastocysts did not cluster by paternal treatment, consistent with no overt developmental $F_1$ phenotype.

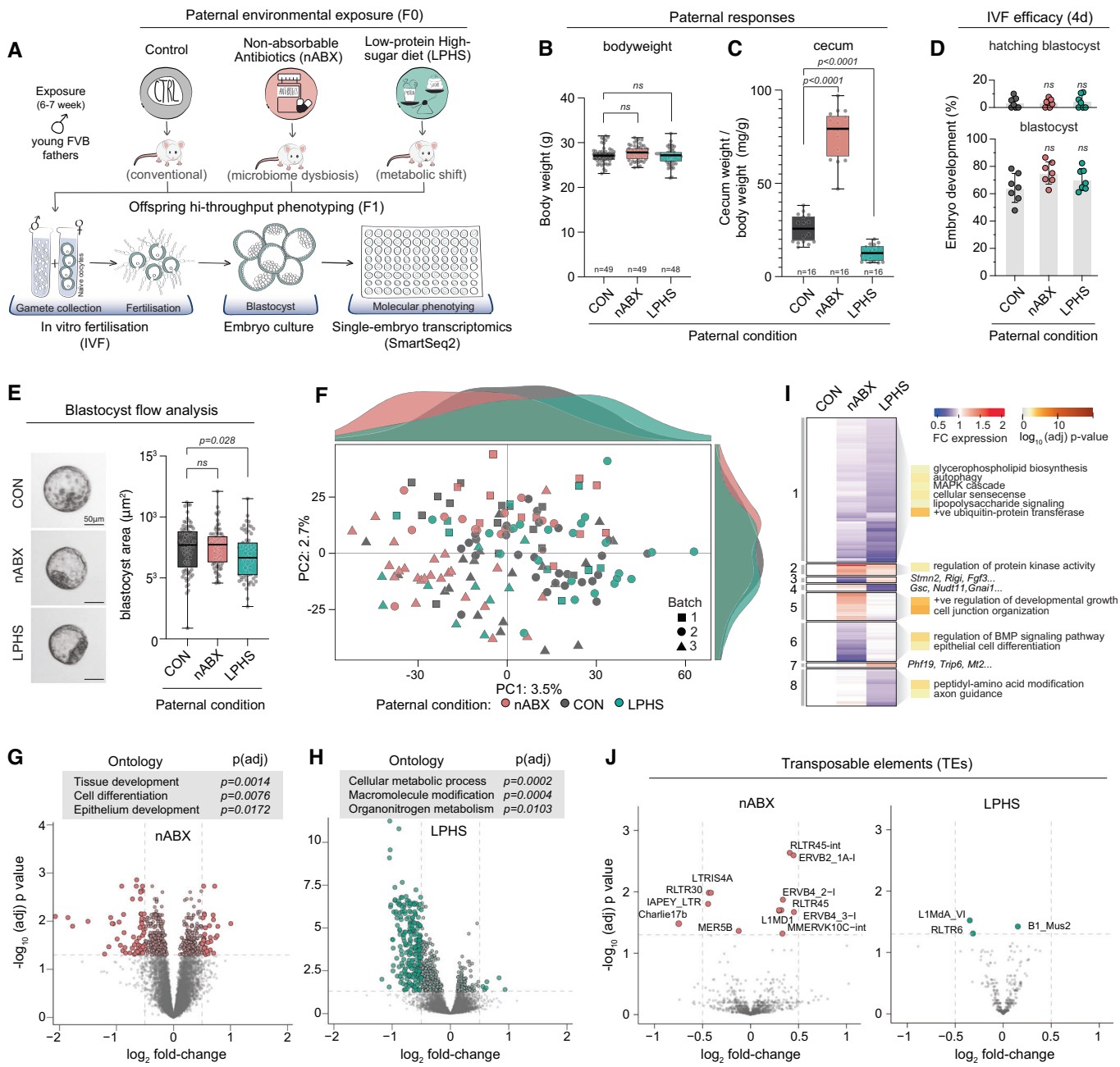

**Figure 1. Offspring signatures of paternal environmental exposure.**

(A) Schematic of the experimental design. Young FVB males are exposed to non-absorbable antibiotics (nABX) or low-protein high-sugar (LPHS) diet for 6-7 weeks. Purified sperm are then collected for parallel in vitro fertilisation (IVF), with the resulting blastocysts undergoing single-embryo transcriptomics using Smart-seq2. (B, C) Box plots showing paternal $F_O$ physiological responses to indicated treatment for (B) bodyweight and (C) caecum weight relative to bodyweight. Each datapoint represents a biological replicate (individual mouse), bars represent the median, and whiskers the minimum-maximum range (nABX: $P = 1.5e-13$; LPHS $P = P = 1.6e-08$). (D) Dot plot showing efficiency of development to blastocyst (bottom) and hatching blastocyst (top) depending on paternal sperm donor condition. (E) Representative images of blastocysts from the three paternal conditions (left), and box plot showing blastocyst size (area) by paternal condition. Each datapoint represents an individual blastocyst, bars represent the median, and whiskers the minimum–maximum range. (F) Principal component analysis (PCA) of blastocyst global transcriptome (all expressed genes) coloured according to paternal treatment. Each data point indicates a single blastocyst and histograms represent the relative density of blastocysts along PC1 (top) and PC2 (right). [#samples: CON = 61, nABX=50, LPHS = 41]. (G, H) Volcano plots showing differential gene expression in (G) nABX and (H) LPHS-derived blastocysts. Grey boxes show gene ontology (GO) terms significantly enriched in the respective differentially expressed gene sets. [#genes = 14429]. (I) Heatmap showing average expression fold-change of DE genes clustered according to F1 response pattern. Shown right selected GO terms significantly enriched in each DE gene cluster. (J) Volcano plots depicting differential expression of transposable elements in paternal nABX (left) or LPHS (right) derived blastocysts. [#TEs = 986]. (B–E) P values were computed using a two-tailed unpaired $t$ test [ns not significant].

However, we did observe a tendency of nABX- and LPHS- derived blastocysts to separate away from control embryos, with the respective exposures moving in opposite directions along the first principal component (PC1) of transcriptome variation (Fig. 1F). Indeed, the density of blastocyst projections along PC1 exhibited a trimodal distribution that reflected paternal nABX, LPHS or control status at conception (Fig. 1F). These data imply that while there is no deterministic global response to paternal condition, a probabilistic tendency towards specific molecular effects in $F_1$ embryos can be observed.

To further investigate this finding, we identified significantly differentially expressed genes (DEG), leveraging high replicate numbers across multiple $F_1$ batches to ensure robustness. This revealed that embryos derived from nABX or LPHS fathers exhibit a high-confidence (hc) DEG profile (adjusted $P < 0.05$, $\log_2$ FC > 0.5) (Fig. 1G,H; Dataset EV1). Specifically, we found 97 hcDEG in nABX-fathered blastocysts, which were enriched for functional roles in cell differentiation and developmental pathways, implying paternal dysbiosis may be linked with an $F_1$ lineage effect (Fig. 1G; Dataset EV2). Amongst the most significantly affected genes were key cell fate regulators *Dkk1*, *Gata4* and *Tbx20*. In contrast, hcDEG in LPHS-derived blastocysts ($n = 244$) were enriched for cellular metabolic processes and were mostly down-regulated (Fig. 1H). This included downregulation of many ribosome-associated genes such as *Rps2, Fbl* and *Utp18*, which is rate-limiting for proliferation (Buszczak et al, 2014). Considered together with significantly smaller blastocyst size (Fig. 1E), these data are consistent with a growth delay of embryos derived from LPHS fathers. Overall, our results indicate we can capture a molecular signature of paternal conditioning in early embryos.

Notably, there was little overlap between the two sets of hcDEG (14 genes), suggesting that distinct paternal perturbations induce distinct $F_1$ effects at the blastocyst stage (Fig. EV1M). This is in line with the reciprocal physiological response of fathers to each treatment, the opposing trend of global blastocyst transcriptomes along PC1 (Fig. 1C,F), and the divergent gene sets enriched across the conditions (Fig. EV1N,O; Dataset EV3). To further understand the specificity of $F_1$ responses, we clustered all DEGs by their fold change across conditions, obtaining eight clusters (Fig. 1I; Dataset EV4). Genes specifically downregulated in nABX-derived blastocysts (cluster 6) were highly enriched for bone morphogenetic protein (BMP) signalling, which plays a key role in extra-embryonic lineage allocation (Graham et al, 2014), while nABX-specific upregulated genes (cluster 5) were linked with cell junction proteins and epithelialisation. The genes preferentially down-regulated in LPHS-derived blastocysts (cluster 1) were enriched for senescence, ubiquitin activity, and the MAPK cascade, which has been implicated in implantation failure (Natale et al, 2004). In this case, genes determined to be preferentially LPHS-specific also trend towards downregulation in the nABX paradigm, hinting at a partially shared $F_1$ response of different magnitudes (Fig. 1I; Dataset EV5). Overall, the data suggest that while some of the intergenerational responses may reflect a general effect of adverse paternal conditioning, specific paternal exposures also trigger specific pathway responses in offspring as early as the blastocyst stage.

Finally, we investigated the paternal impact on transposable element expression. TEs have been implicated as both markers and functional regulators of successive stages of preimplantation development, and have also been previously linked with epigenetic inheritance (Modzelewski et al, 2021; Morgan et al, 1999; Todd et al, 2019). We observed only limited significant transposable element responses in LPHS-derived blastocysts (Fig. 1J). However, we note significant changes in expression of transposable elements in blastocysts derived from nABX-exposed fathers, primarily in long terminal repeat (LTR) retrotransposons, including upregulation RLTR45 and MMERVK10C and downregulation of IAPey (Fig. 1J; Dataset EV6). These LTR are dynamically expressed during embryonic progression, and have been shown to be amongst the most epigenetically responsive transposable elements (Hackett et al, 2017; Reichmann et al, 2012), pointing towards an aberrant epigenome and/or differences in development. Taken together, we were able to identify significant albeit subtle $F_1$ signatures specific to each paternal exposure, reflecting both protein-coding genes and transposable element responses.

## The regulatory basis of intergenerational signatures

To investigate potential mechanisms underlying $F_1$ gene expression changes, we assessed the upstream regulators of DEGs by searching for transcription factors (TFs) that directly bind their proximal promoters. We found TFs that confer lineage identity, such as NANOG, ZIC3 and CDX2, to be highly enriched at promoters of nABX-specific DEGs (Figs. 2A and EV2A; Dataset EV7). Moreover, the downstream effectors of BMP signalling, SMAD1, SMAD3 and SMAD4, were also amongst the most significant binders of DEG promoters, consistent with enrichment for BMP signalling pathways in nABX blastocysts. In contrast, DEGs from LPHS blastocysts were associated with MYC and TRIM28 promoter-proximal binding (Fig. 2A), which are linked with cell fitness/proliferation and targeting H3K9me3, respectively (Claveria et al, 2013; Rowe et al, 2013; Scognamiglio et al, 2016). To understand whether $F_1$ DEGs also exhibit a chromatin signature that could provide insight into their regulation, we leveraged published ChIP-Seq datasets from the ICM of blastocysts (Liu et al, 2016; Wang et al, 2018; Xu et al, 2019). We observed an enrichment of H3K9me3 preferentially in the promoters of LPHS-DEGs, consistent with those being TRIM28 targets, since TRIM28 promotes H3K9me3 (Fig. 2B). Curiously, H3K9me3 often co-occurred with H3K4me3, with this chromatin signature being absent in the promoters of nABX-specific DEG (Fig. 2B,C). The cumulative data imply that paternal LPHS diet may influence $F_1$ regulatory pathways related to growth and proliferation, whereas dysbiotic nABX-exposed fathers could affect offspring pathways involved in cell-lineage allocation.

The appropriate formation and allocation of the three primary lineages in the developing blastocyst—epiblast (EPI), primitive endoderm (PrE) and trophectoderm (TE)—is crucial to ensure implantation success. Indeed, recent studies have demonstrated that aberrant primitive endoderm is the key predictor of poor outcomes for human embryos (Chousal et al, 2024). To investigate whether paternal condition could trigger lineage-related defects, we grouped key markers for each lineage into genesets and measured their collective expression change across each embryo as proxies for lineage allocation (see 'Methods'; Dataset EV8). Blastocysts sired by nABX-treated males exhibited significant downregulation of primitive endoderm markers ($P = 0.049$) and concomitant upregulation of epiblast genesets ($P = 0.0027$) (Fig. 2D,E). This reciprocal

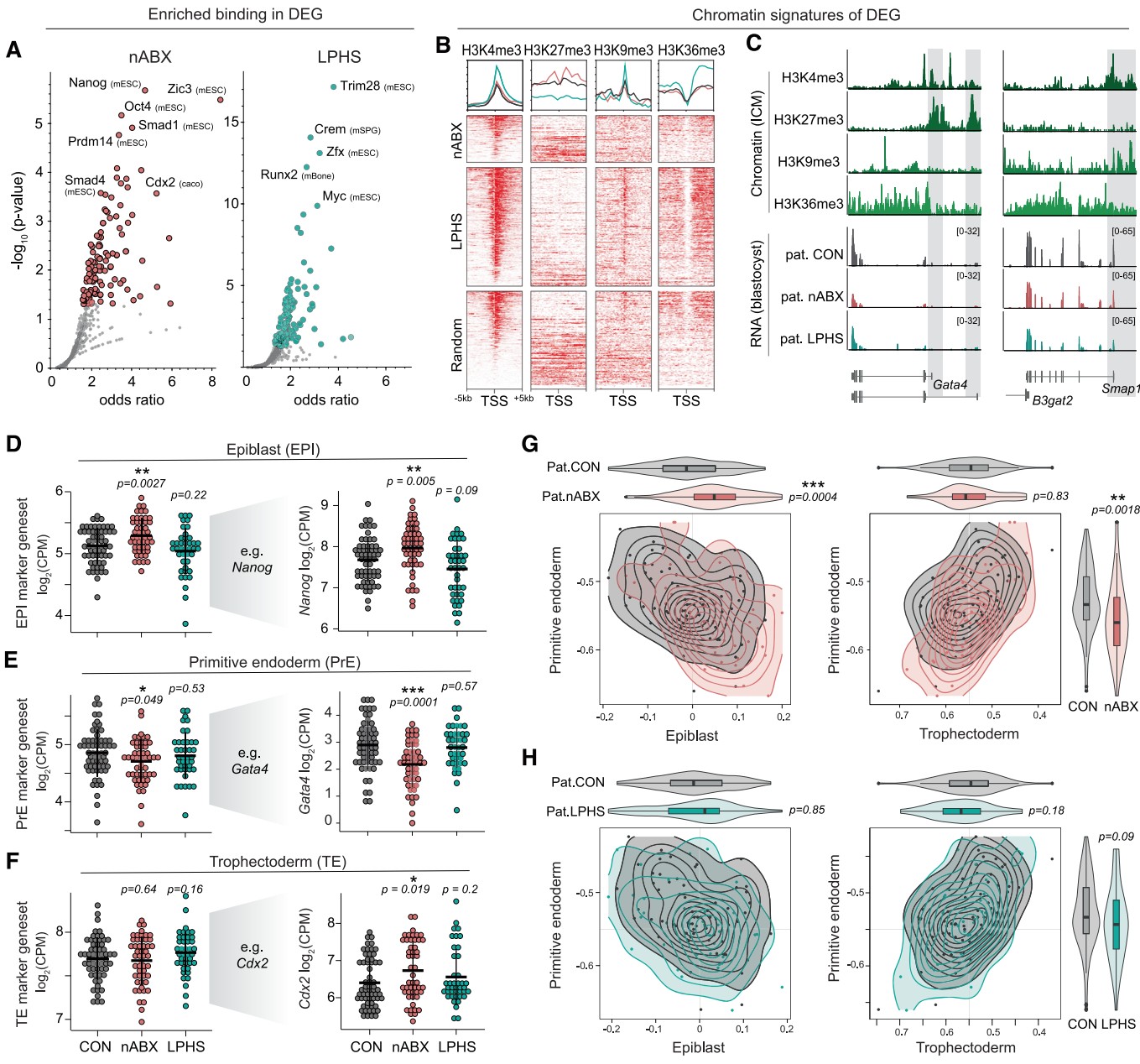

**Figure 2. The regulatory basis of intergenerational transcriptomic signatures.**

(A) Scatter plot showing enrichment for binding of specific transcription factors within DEG set promoters from paternal nABX- (left) or LPHS- (right) blastocysts. (B) Heatmaps and summary profiles of H3K4me3, H3K27me3, H3K9me3 and H3K36me3 enrichment around the TSS of DEG sets in wildtype blastocysts, sorted by H3K4me3 enrichment. Random genes are shown for comparison. (C) Representative genome tracks of DE genes showing chromatin marks in wildtype blastocysts and gene expression responses across the paternal treatments. (D–F) Dot plots depicting the expression of lineage markers in embryos derived from each paternal condition. Shown left is mean expression of collective marker genes for each lineage: epiblast (top), primitive endoderm (middle) and trophectoderm (bottom). Markers used for the respective geneset (see Fig. EV2D). Shown right are representative genes for each lineage. Each datapoint indicates $\log_2$ CPM expression of a geneset/gene in a single blastocyst. P-values computed using the two-tailed Wilcoxon test. The horizontal black line is the median, and the vertical black line extends from $Q1 - 1.5 \times IQR$ to $Q3 + 1.5 \times IQR$, where Q1 and Q3 refer to the first and third quartiles, respectively, and $IQR =$ interquartile range (range of 25th to 75th percentile of the distribution). (G, H) Contour plots and violin plots of correlation coefficients of blastocyst transcriptomic profiles from indicated paternal sperm donor with the reference single-cell lineage, comparing (G) nABX versus CON, and (H) LPHS versus CON. Correlation coefficients of primitive endoderm and epiblast (left) and primitive endoderm and trophectoderm (right) are displayed. P values were computed using a two-tailed Wilcoxon test. [Box plots: horizontal black line at the centre is the median. The box extends from the first quartile (Q1: 25th percentile) to the third quartile (Q3: 75th percentile). The vertical black line extends from $Q1 - 1.5 \times IQR$ to $Q3 + 1.5 \times IQR$, where $IQR =$ interquartile range. The minima and maxima are not explicitly annotated in these figures]. (D–H) [#samples: CON = 61, nABX=50, LPHS = 41].

effect is exemplified by significantly increased expression of EPI genes *Nanog, Pecam1* and *Tfcp2l1*, and downregulation of PrE markers *Gata4* and *Fgfr2* specifically in nABX-derived embryos (Figs. 2D–F and EV2A). Immunofluorescence staining for a canonical PrE marker (SOX17) hinted that these gene expression changes are unlikely to arise from a difference in the proportion of PrE cells in nABX-derived embryos (Fig. EV2B,C). While the trophectoderm geneset did not exhibit a significant overall change, we observed that critical TE regulators, including *Cdx2, Tfap2c and Tead4*, were significantly upregulated specifically among blastocysts sired by nABX sperm donors (Fig. 2F). Notably, immunofluorescence in independent $F_1$ cohorts confirmed a significant increase in the number of CDX2$^+$ cells in blastocysts derived from nABX-treated fathers (Fig. EV2B,C). In contrast, we detected no lineage mis-regulation in blastocysts from LPHS sperm donors, at the RNA or cell proportion level, which were derived concurrently from the same oocyte pool (Fig. EV2A,C).

To further investigate the potential lineage bias in nABX-derived embryos, we constructed reference signatures of the three lineages using published single-cell blastocyst data (Deng et al, 2014) (Fig. EV2D), and projected our blastocyst profiles onto this reference. We observed a decreased primitive endoderm signature and increased epiblast signature in paternal nABX-derived blastocysts, but not in LPHS blastocysts (Fig. 2G,H). Importantly, in silico developmental staging suggested that these embryos are at equivalent developmental timepoints to control, implying lineage expression changes are not simply a consequence of altered developmental timing (Fig. EV2E). This is further supported by the precise equivalence in chronological timing from fertilisation (92+/−2 h). Taken together, multiple lines of transcriptomic enquiry identify changes preferentially in lineage-specific genes and lineage-associated signalling pathways (e.g., BMP) in nABX-derived blastocysts. These data imply paternal nABX at least partially manifests in offspring by influencing the expression of lineage genes and/or the proportion of cells allocated to each lineage during early embryogenesis.

## Batch effects and design principles to capture intergenerational signatures

Early embryonic gene expression is inherently noisy, and capturing the effects of relatively subtle paternal influences is confounded by both technical and biological variation. In order to obtain a more robust picture of the intergenerational signatures, we profiled three independent batches, each consisting of multiple blastocysts ($n = 26$–45), across each of the three paternal environments. Batches were designed such that all IVF and embryo culture stemming from differentially exposed sperm donors ($n = 3$ per batch) occurs in parallel, using a shared oocyte pool, to minimise intra-batch confounders (Fig. 3A). While prior analysis of integrated batches captured paternal-effect signatures (Fig. 1G–I), we also found blastocysts preferentially separated according to batch, implicating batch as the strongest variable (Fig. 3B). When not accounted for, batch effects can obscure *or* over-emphasise underlying biology.

Our high replicate numbers allowed us to analyse each batch independently. PCA of each batch revealed a greater level of separation between $F_1$ blastocysts from treated and control males

than observed in the integrated analysis, particularly for paternal nABX treatment (Fig. 3C,D). Moreover, independent batch analysis identified a marked number of hcDEG in blastocysts sired by nABX- or LPHS-exposed males relative to control (Figs. 3E,F and EV3A). We used these to identify 'batch robust' DE genes (broDEGs), comprising only genes that were DE in at least two batches (Figs. 3G–J; Dataset EV9). The overlaps between such broDEGs and hcDEGs (Fig. 1G–I) were modest but highly statistically significant (Fig. 3H–J), indicating that the two strategies for identifying consistent transcriptomic changes can be used complementarily. Indeed, primitive endoderm marker *Gata4* was systematically downregulated in nABX-derived blastocysts across all three batches. Moreover, *Fgfr2*, along with *Nanog, Tat* and *Tfcp2l1* were also dysregulated across independent batches derived from dysbiotic fathers (Figs. 3K and EV3A,B). Finally, developmental signalling genes like *Rac1* and *Mapk1* were reproducibly downregulated in LPHS samples (Fig. 3K).

In summary, multiple paths of analysis led us towards the same conclusion: paternal nABX and LPHS treatments propagate information to the next generation that can be detected as gene expression changes in the F1 blastocysts. However, our analyses indicate that blastocyst expression profiles inherently possess a low signal-to-noise ratio of transmitted effects (signature of paternal effect *vs* technical and batch-specific variation). This emphasises that paternal effects are subtle at the molecular level, and motivates the need for careful experimental design to guard against technical confounders or overinterpretation of intergenerational phenotypes.

## Interactions between paternal genetic background, age and environment

Epigenetic phenomena are inextricable from genetics (Cavalli and Heard, 2019). Using inbred mouse strains provides an opportunity to assess, in a controlled manner, the role of genetic background (G) on the propagation of an environmentally-induced (E) $F_1$ phenotype (intergenerational GxE). To investigate this, we repeated the paternal nABX and LPHS exposure paradigms, IVF and blastocyst transcriptomic profiling using independent batches of C57BL/6 (B6) genetic background males, with strain-matched oocytes. The direct physiological response of males was comparable in the B6 background as in FVB, with only a limited impact on overt phenotypes, but significant differences in caecum weight (Fig. EV4A–C). Moreover, there was no significant difference across B6 paternal conditions in the developmental rate, hatching or male-female ratio of IVF-derived blastocysts (Fig EV4D–H). We did however note that IVF using B6 sperm donors was markedly less efficient and more variable than FVB, irrespective of paternal conditioning, implying greater technical noise in the B6 background.

As observed in the FVB background, there was no global separation between transcriptomes of blastocysts derived from different paternal environments by PCA (Fig. 4A). However, we also observed no significant differential gene expression across both paternal conditions, indicating the absence of a DEG signature. Genes reported to be DE in FVB did not show condition-specific expression directionality in the B6 background (Fig. 4B). Using Gene Set Enrichment Analysis (GSEA), we observed that terms related to sodium ion transport, oxidative phosphorylation, and ribosomal activity, that were significantly

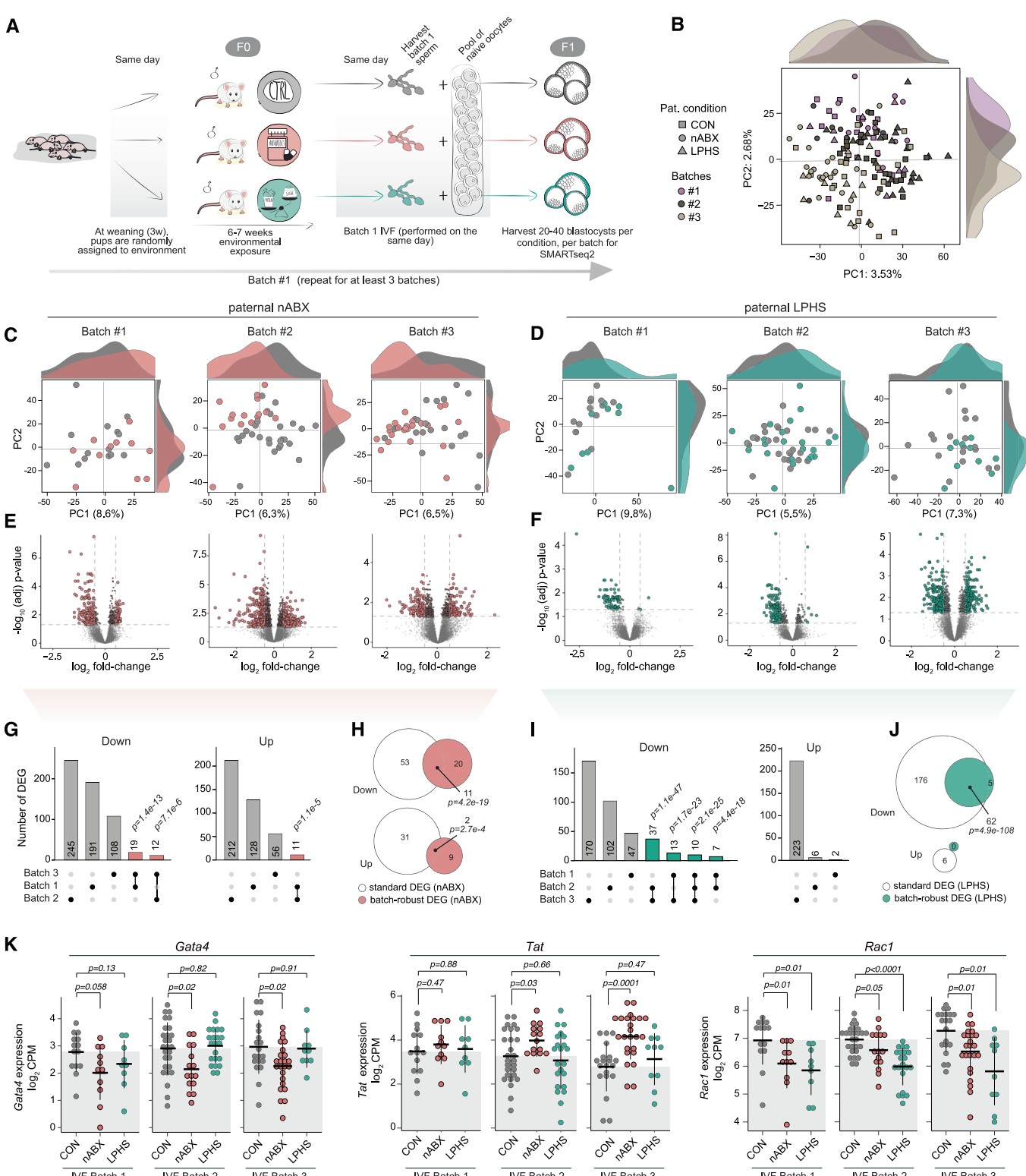

**K**

*Gata4*

*Tat*

*Rac1*

enriched in the FVB background, showed little to no enrichment in B6 background (Fig. EV4I,J). In summary, the same environmental exposures in different genetic backgrounds lead to divergent intergenerational outcomes. Whilst this could be due to a range of

biological or technical reasons (see Fig. 5), it also highlights the significance of GxE interactions in intergenerational models, and implies that modifying either variable alters (or ablates) $F_1$ outcomes.

◀

**Figure 3. Overcoming batch effects to capture consistent IEI signatures.**

(A) Schematic depicting the organisation of batches to obtain statistical power. IVFs from all sperm donors are performed on the same oocyte pool in parallel to generate embryos, which is repeated on multiple occasions (batches). (B) PCA on global transcriptomes from Fig. 1F coloured by batch. (C, D) PCA on global transcriptomes of F1 blastocysts separated by individual experimental batches, derived from (C) paternal nABX or (D) paternal LPHS exposures and relative to control fathers. Shaded histograms represent the relative density of blastocysts along PC1 (top) and PC2 (right). (E, F) Volcano plots of DE genes from nABX or LPHS paternal treatment relative to controls in the same batch. (G) Upset plots comparing differentially expressed gene (DEG) sets in each batch in the nABX model. (H) Venn diagrams showing overlap of upregulated (top) and downregulated (bottom) DE genes in the standard versus batch-robust analysis (broDEG). (I) Upset plots comparing differentially expressed gene sets in each batch in the LPHS model. (J) Venn diagrams showing overlap of DE genes in the standard analysis versus batch-robust analysis. (G–J) P values computed using Fisher's exact test. (K) Dot plot of representative examples of DE genes that show consistent differential expression across batches. P values indicate two-tailed Wilcoxon test. (E, F, K) [#genes = 14429]. The horizontal black line is the median, and the vertical black line extends from $Q1 - 1.5 \times IQR$ to $Q3 + 1.5 \times IQR$, where Q1 and Q3 refer to the first and third quartiles, respectively, and IQR = interquartile range (range of 25th to 75th percentile of the distribution).

## Impact of paternal age on embryonic gene expression

To further investigate the interactions between paternal genetic background and intergenerational epigenetic inheritance, we elected to use ageing as a stronger environmental factor. Mammalian sperm accumulate (epi)genetic changes with age, which is linked with altered offspring phenotypes (Ashapkin et al, 2023; Oluwayiose et al, 2021). We used mature (12–13 month old) males—approximating middle age in humans - as sperm donors to capture $F_1$ effects relative to young (9 weeks) males. PCA on B6-derived single-blastocyst transcriptomes ($n = 157$) revealed a distinction between embryos sired by mature or young males along the second principal component (PC2) (Fig. 4C). Moreover, we observed substantial DEGs in blastocysts derived from mature B6 sperm donors (Fig. 4D; Dataset EV10). Downregulated genes were preferentially associated with splicing and oxidative phosphorylation, which is a hallmark of impaired metabolic plasticity that impacts preimplantation embryogenesis (Moussaieff et al, 2015) (Fig. 4D). Upregulated genesets were linked with immune-related processes. To investigate the generalisability of ageing as an extrinsic factor, we also assayed the impact in the FVB genetic background ($n = 202$). We again observed a separation of $F_1$ blastocyst transcriptomes, and widespread imposition of DEGs in embryos derived from mature fathers (Fig. 4E,F; Dataset EV11). GSEA revealed shared signatures of paternal ageing manifest in $F_1$ offspring across B6 and FVB genetics related to nucleosome and mitochondrial pathways, but also identified considerable background-specific effects (Fig. 4G; Dataset EV12). The cumulative data indicate that paternal age represents a significant environmental factor that triggers differential gene expression as early as the blastocyst stage. Moreover, similarly to paternal nABX and LPHS exposure, the precise nature of intergenerational effects elicited by aging are modified by the genetic background.

## Intrinsic population variability as a readout of intergenerational effects

We next investigated the discordance between the FVB and B6 backgrounds in $F_1$ responses to paternal nABX/LPHS conditioning. We hypothesised that this GxE effect may partially stem from intrinsically different 'noise' levels in blastocyst gene expression between the two genetic backgrounds. Noisy gene expression in a dynamic model obscures real signal, making it difficult to observe a consistent gene expression phenotype, which is the basis of conventional DE analysis. We therefore next asked a complementary question, namely whether paternal nABX or LPHS trigger increased variability in offspring gene activity across individual embryos. Indeed, the postnatal growth phenotype induced by paternal nABX arises probabilistically and thus manifests as increased variation. To investigate this possibility here, we fitted a generalised linear model to the mean-variance relationship of gene expression in our data, and computed variability of a gene as the log-transformed ratio of its observed to expected variation (Figs. 5A and EV5A).

We first probed our FVB dataset and observed a significant difference in overall noise of blastocyst gene expression across paternal exposures ($P = 0.024$) (Fig. 5B). To isolate the source of this, we identified genes that exhibit substantially higher variability ($\Delta Variability_{nABX - CON} > 0.5$ or $\Delta Variability_{LPHS - CON} > 0.5$) in response to specific paternal perturbations, finding 292 in nABX and 438 in LPHS (Fig. 5C,D; Dataset EV13). Hypervariable genes showed enrichment for essential pathways like translation elongation and embryo implantation (nABX), and mRNA metabolic processes (LPHS) (Datasets EV14 and 15). Amongst variable loci, we noted that MERVL retrotransposons scored as the most hypervariable across all genes and TEs, specifically in response to paternal LPHS (Fig. 5D). MERVL and its target genes have previously been implicated in the intergenerational effects of low protein diet (Boskovic et al, 2020; Sharma et al, 2016). Indeed, no MERVL effect was observed in nABX embryos, underscoring its specificity to paternal LPHS treatment. Importantly, we observed B6 blastocysts also exhibited altered gene variability in response to paternal conditioning, which included hypervariable MERVL specifically from LPHS fathers (Fig. EV5B–D; Dataset EV16). These data imply paternal conditioning across independent backgrounds manifests as changes in the variability of offspring gene expression profiles. High variability in essential pathways is detrimental for the success of the population, and may contribute to previous observations of disease susceptibility (Argaw-Denboba et al, 2024; Dalgaard et al, 2016).

We next compared variability between FVB and B6 background, first asking whether blastocysts exhibit equivalent baseline variation. Gene expression variability from both genetic backgrounds revealed that B6 blastocysts derived from control males are significantly noisier than their control FVB counterparts (Figs. 5E and EV5E). This implies that B6 genetics is inherently linked to more variation in expression landscapes during early development. We further found that noise in blastocysts derived from nABX or LHPS fathers is significantly greater in B6 background than FVB (Fig. 5E). This increased gene expression noise could explain the difficulty in capturing consistent $F_1$ transcriptomic responses to our paternal nABX and LPHS paradigms specifically in the B6 background.

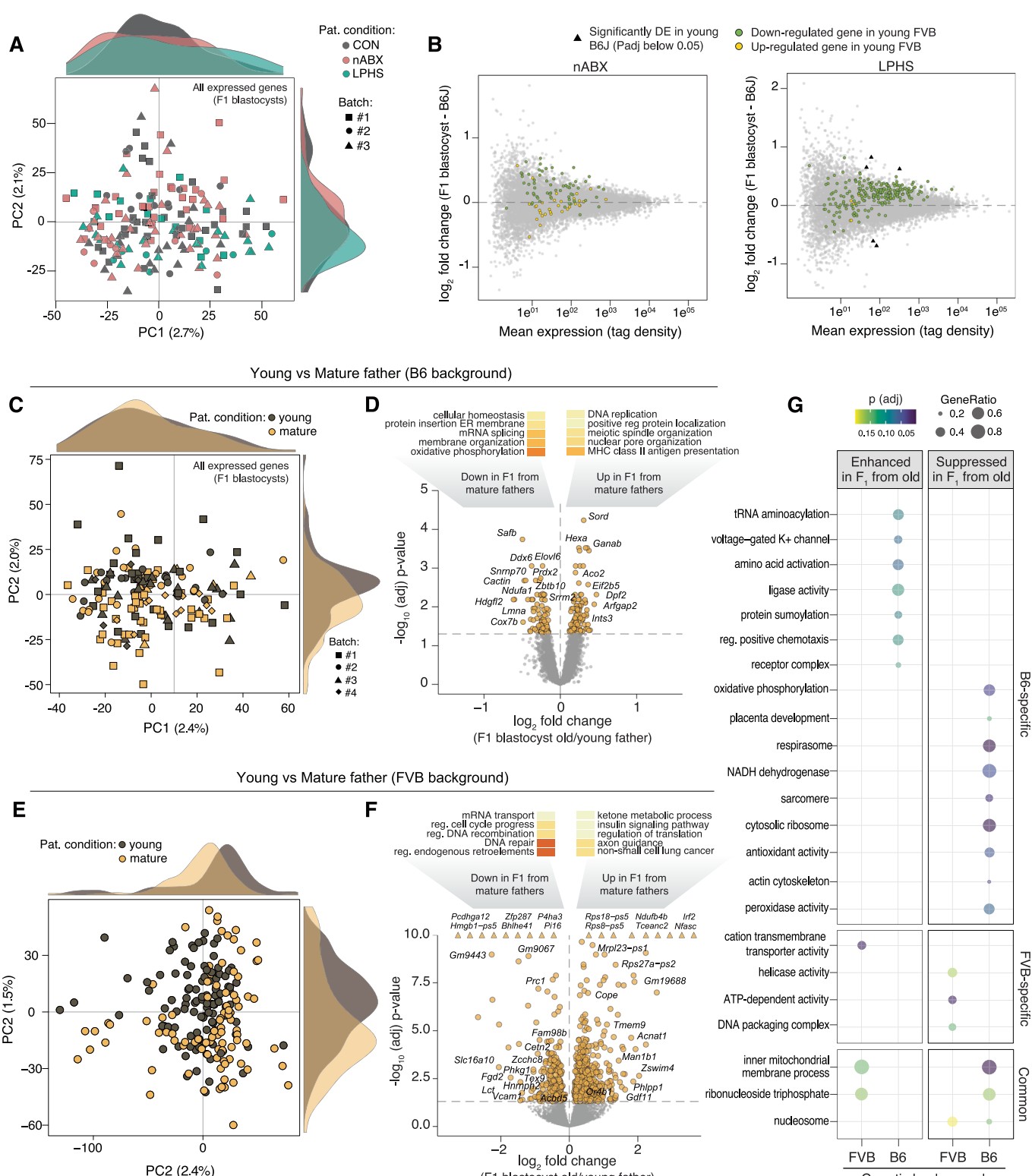

Finally, we used gene expression variability as a score to identify genes that exhibit reproducibly elevated F1 variability in response to paternal perturbations across genetic contexts. We identified a signature of genes that were hypervariable in both FVB and B6 backgrounds in blastocysts derived from dysbiotic nABX fathers.

These included key lineage genes such as *Hand1*, and were significantly enriched with gene ontology terms including 'embryonic organ morphogenesis' and 'response to external stimulus' (Fig. 5F; Dataset EV17). We further identified a set of hypervariable genes triggered by LPHS-treated fathers across both B6 and FVB

**Figure 4. The impact of genetic background and age on intergenerational epigenetic inheritance.**

(A) PCA plot of blastocyst transcriptomes of C57BL/6 (B6) background across the three paternal treatments. (B) MA plot of (left) nABX and (right) LPHS blastocyst gene expression relative to B6 controls, showing relative expression of the IEI signature identified in the FVB genetic background. (C) PCA on whole transcriptomes of individual blastocysts derived from young or mature males in the B6 genetic background. (D) Volcano plot showing differential expression between blastocysts from mature and young B6 males. Shown above is a deconstructed heatmap with GO terms enriched in DE genes downregulated (left) and up-regulated (right) in blastocysts from mature fathers. [#genes = 12663] (E) PCA on whole transcriptomes of blastocysts from young versus mature FVB males. (F) Volcano plot showing differential expression between blastocysts derived from FVB background mature vs young males. Thresholds for significance are set at adjusted $P$ value < 0.05 and abs(log2FoldChange) > 0.5. [#genes = 10882]. Triangular shapes signify genes with $-\log10(adj)$ $P$ value > 10. (G) Bubble plot showing relative geneset enrichment for up- and downregulated genes in blastocysts from aged fathers across both B6 and FVB genetic backgrounds.

backgrounds. These included 2-cell stage genes *Tcstv3* and *Mervl*, and were overall enriched in 'regulation of endocrine processes' (Fig. 5G). We conclude that gene expression variability is a relevant readout to capture intergenerational responses, and may represent a mechanistic route through which paternal exposures present in early embryogenesis.

## Paternal reproductive responses across environmental exposures

Given that our environmental perturbation paradigms introduce detectable F1 signatures, we finally asked how they affect the upstream molecular properties of the reproductive system in exposed fathers. We profiled the testis transcriptome, and noted that responses were driven primarily by genetic background. For example, nABX-induced dysbiosis was associated with geneset enrichment for 'steroidogenesis' and 'fatty acid metabolic' genes in FVB testes, consistent with altered lipid profiles in testis of nABX-treated males (Fig. 6A,B) (Argaw-Denboba et al, 2024). In contrast, B6 were more closely linked with immune-associated processes (Fig. 6C,D), suggesting there is a subtle response to environmental exposures at the level of bulk transcriptome (Fig. 6C and EV6A). We also profiled testes of aged males, which elicited wholesale changes in gene expression. The majority of the DEGs in mature testes were upregulated, and corresponded to immune activation, implying increased inflammatory responses in the aged reproductive system (Fig. 6E).

Multiple paradigms of intergenerational inheritance have reported differential abundance of small RNAs in sperm in response to environmental exposures (Argaw-Denboba et al, 2024; Chen et al, 2016; Gapp et al, 2014; Sharma et al, 2016; Tomar et al, 2024), which are proposed to partly or wholly underpin phenotypic transmission. To evaluate the signature of small RNAs in our perturbations, we adapted RNA-isolation methods in order to sequence sperm-borne small RNA from individual males (as opposed to several males pooled), and used sufficient replicate numbers ($n = 5$–$6$ per condition) to ensure robustness. We confirmed that, from individual male sperm samples, we obtained the expected proportions of tRNAs, miRNAs and piRNAs, and appropriate sequencing depth (median=5.8mio post-filtering/sample) to perform differential small RNA expression analysis (Fig. EV6B,C). We observed only limited statistically significant changes in the abundance of small RNA of any family in either the FVB or B6 backgrounds, in both the nABX and LPHS treatment paradigms (Figs. 6F and EV6D,E). A small number of tRNA-derived small RNAs and miRNAs were significantly abundant upon LPHS treatment in the B6 background

(Fig. 6F). Paternal age was, by some distance, the strongest perturbation, with miRNAs and tRNAs showing significant expression changes in sperm from older males (Figs. 6F and EV6F; Dataset EV18).

To further understand the limited changes in miRNA and tRNA upon paternal environmental exposure in young males, we examined the expression of small RNA within each individual. We identified considerable heterogeneity in expression levels in sperm from young males (Fig. 6F). In contrast, mature sperm samples exhibited a much more consistent expression profile across replicates (Fig. 6F). These results imply an inherent variability in small RNA levels in sperm of younger males, which are typically deployed in intergenerational studies. These data indicate that sperm-borne small RNA do not exhibit consistent and significant environmental responses when using relatively high-powered experimental designs, while paternal age acts as a strong modulator of the reproductive system transcriptome and sperm small RNA payloads. This consideration should be taken into account when dissecting the mechanisms of transmission for intergenerational phenotypes and indeed, the triggers of the F1 molecular signatures reported in the present study.

## Discussion

In this study, we executed a highly controlled strategy with large replicate numbers to comprehensively characterise the earliest molecular signatures of mammalian paternal effects. Across all IVF experiments, we generated transcriptomic profiles from 1006 individual embryos spanning three paternal treatments, two paternal ages and two genetic backgrounds. The results reveal a transcriptomic signature of the preconception paternal condition emerges in F1 blastocysts. The data further show that such signatures are typically subtle, and interact with underlying genetics, age, and potentially additional factors to greatly influence overall outcome. This study therefore reinforces that transmission of paternal environmental factors constitutes an important source of F1 phenotypic variation, but also that complex and confounding influences play into the penetrance and detection of such effects.

One intriguing observation is that nABX-induced paternal gut microbiome dysbiosis may mechanistically operate by influencing early lineages in embryos. We found that across independent experimental batches, paternal dysbiosis led to altered expression of extra-embryonic lineage markers, particularly reduced primitive endoderm (exemplified by *Gata4)*, with concurrent higher expression of epiblast lineage genes. This could reflect an altered ratio of

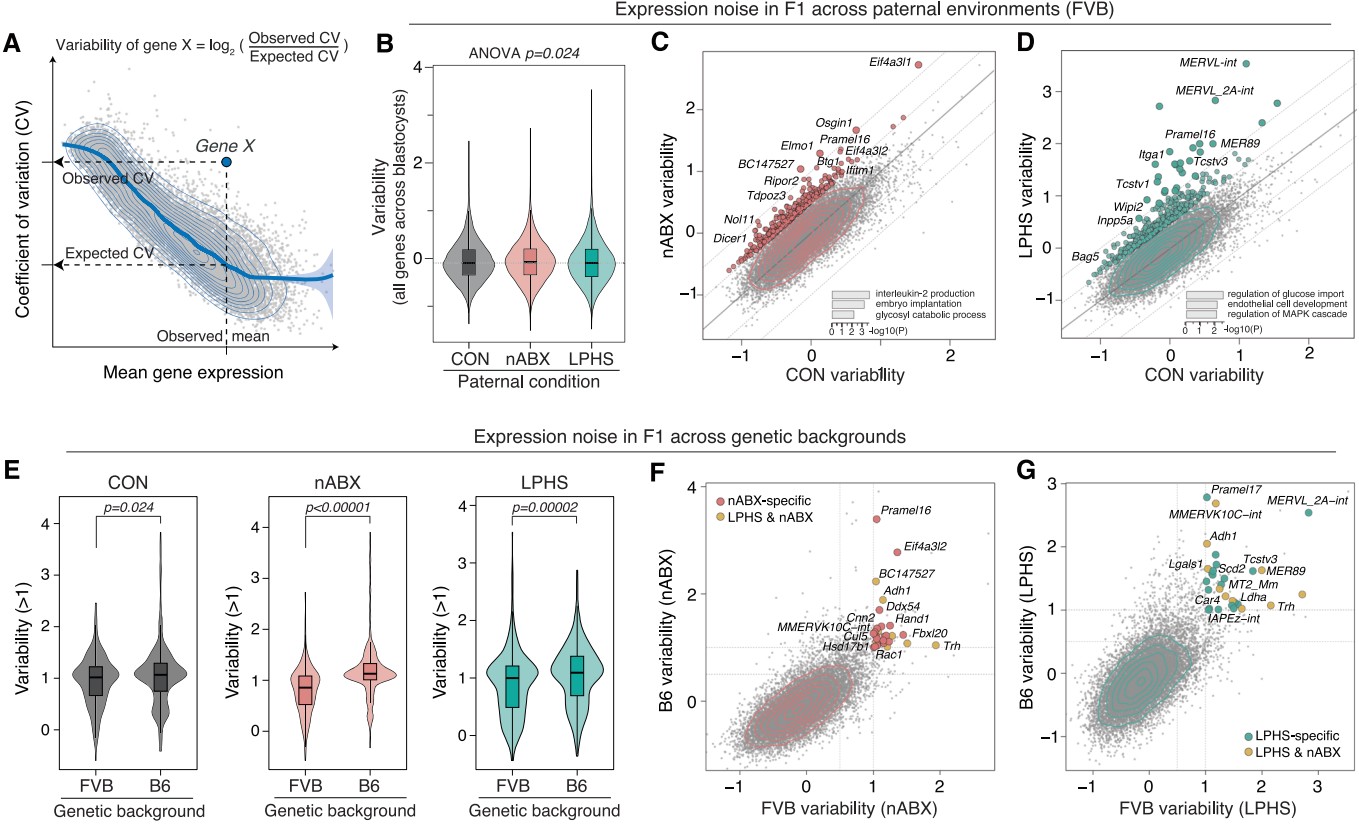

Figure 5. **Intrinsic population variability as a readout of intergenerational epigenetic inheritance.**

(A) Scatter schematic depicting the calculation of variability of a gene "X" using the coefficient of variation (CV) versus mean expression. (B) Violin and box plots comparing the distribution of variability of all expressed genes in the F1 blastocyst transcriptomic profiles of FVB males across the three paternal treatments. *P* value computed using ANOVA. [#genes = 14728]. (C, D) (top) Scatter plots contrasting the variability of genes in the nABX-derived blastocysts (C) and LPHS-derived blastocysts (D) relative to controls. Genes showing higher variability in blastocysts of treated males are highlighted. (C) Genes with nABX variability – CON variability > 0.5 are shown with bigger dots. Genes with nABX variability – CON variability >1 are shown in smaller dots. (D) Genes with LPHS variability – CON variability >0.5 are represented with bigger dots. Genes with LPHS variability – CON variability >1 are represented with smaller dots. (bottom) Bar chart depicting selected significantly enriched GO terms upon overrepresentation analysis of all highlighted genes in nABX (C) and LPHS (D) paternal conditions. (E) Violin and box plots comparing the distribution of variability of highly variable genes (see 'Methods') in F1 blastocyst transcriptomic profiles between the FVB and B6J backgrounds derived from control (left), nABX-treated (middle) and LPHS-treated (right) males. *P* values were computed using a two-tailed unpaired *t* test. [nABX *P* value < 2.2e-16]. [#genes: CON = 654, nABX=662, LPHS = 856]. (F, G) Scatter plots contrasting the variability of genes in FVB and B6 backgrounds in F1 blastocyst transcriptomic profiles from nABX-treated (F) and LPHS-treated (G) males. Genes showing high variability (Variability > 1) specific to paternal treatment in both genetic backgrounds are highlighted in brown. (B, E) [Box plots: horizontal black line at the centre is the median. The box extends from the first quartile (Q1: 25th percentile) to the third quartile (Q3: 75th percentile). The vertical black line extends from Q1 − 1.5 × IQR to Q3 + 1.5 × IQR, where IQR = interquartile range. The minima and maxima are not explicitly annotated in these figures.

(extra-)embryonic lineage allocation and/or specific changes in the expression of master regulators within lineages. In principle, even a modest imbalance in lineage identity and/or embryo size could propagate through development and elicit a temporally-uncoupled emergent phenotype in mature tissues, such as the placenta (Greenberg et al, 2017; Perez-Garcia et al, 2018; Simon et al, 2018). Indeed, paternal microbiome dysbiosis has been shown to affect F₁ phenotype probabilistically, via impacting placental development (Argaw-Denboba et al, 2024). In contrast to nABX, we found paternal exposure to a low-protein, high-sugar (LPHS) diet could be linked with a modest growth effect, as judged by smaller embryos and reduced expression of ribosomal/MAPK genes, which regulate proliferation. Such effects of paternal diet on embryonic growth/proliferation have been noted before, as have MERVL responses (Sharma et al, 2016), and can lead to impaired

implantation (Modzelewski et al, 2021). Taken together, these observations imply that that distinct paternal environments elicit relatively specific effects on progeny, at least for the initial presentation.

Our study also marks a note of caution in interpreting the impact of intergenerational effects. For example, we found that the genetic background fundamentally affects the prevalence and signature of paternal effects. This implies that disentangling GxE is not only key to understand phenotypic traits in individuals, but also for their propagation to the next generation (intergenerational GxE or iGxE). The extent to which the difference between backgrounds represents biology or technical challenges is unclear, however. Indeed, B6 embryos are inherently more variable in their gene expression and developmental success rates, which impairs the ability to detect consistent and subtle paternal effects. Conversely,

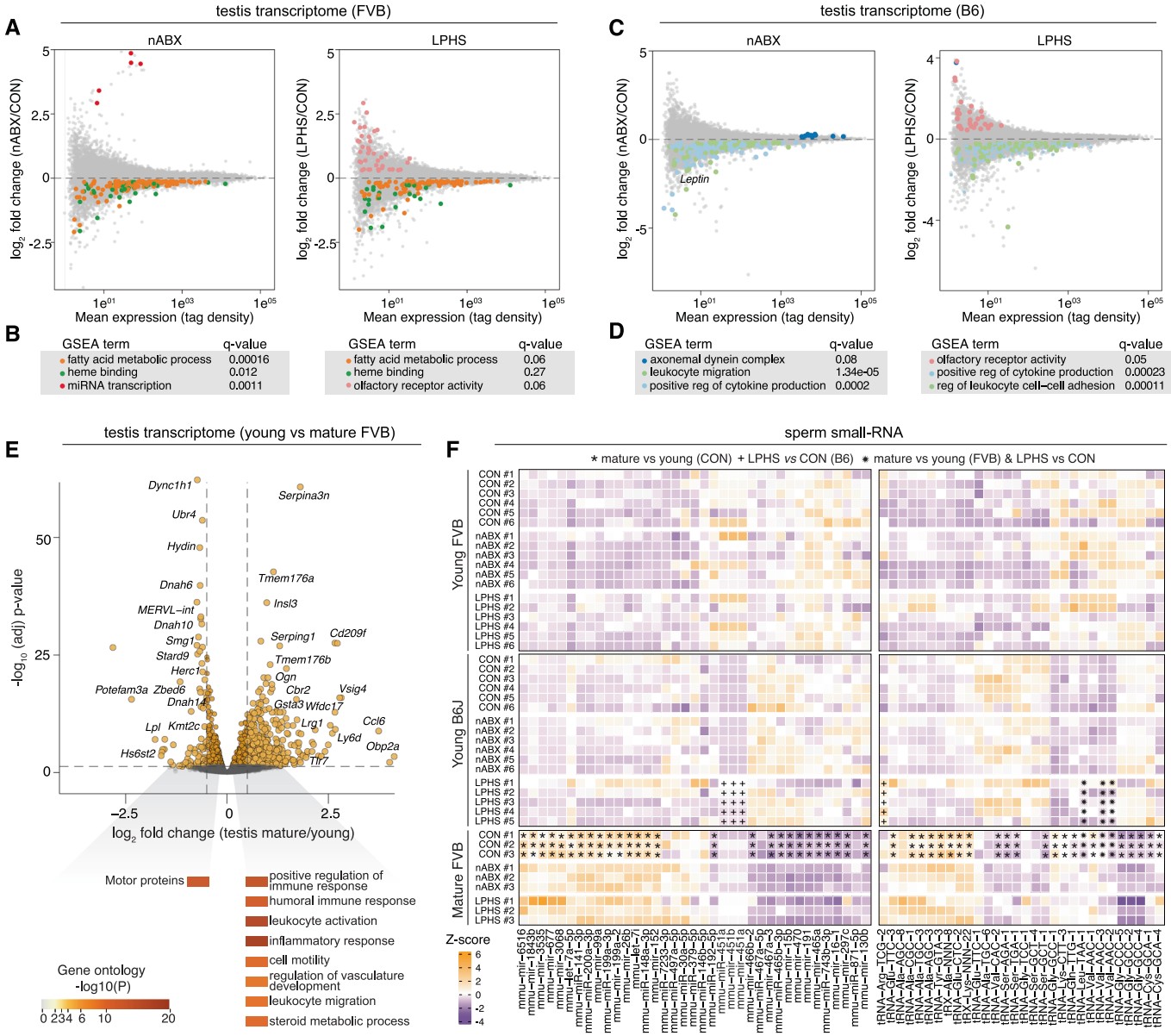

**Figure 6. Paternal reproductive responses to environmental exposures.**

(A) MA plots of (left) nABX and (right) LPHS testes of FVB males versus controls, highlighting genes involved in (B) enriched GSEA terms with their respective *q* values. (C) MA plots of (left) nABX and (right) LPHS testes of B6 males versus controls, highlighting genes involved in (D) enriched GSEA terms with their respective *q* values. (E) Volcano plot showing differential gene expression between testes of young and mature FVB males; (bottom) deconstructed heatmap highlighting select enriched GO terms in DE genes. [#genes = 16618]. (F) Heatmap showing relative expression of selected small RNAs across paternal treatments, ages and genetic backgrounds. Statistical significance determined using DESeq2 adjusted *P* value < 0.05. [∗: Significant DE small RNA in CON mature FVB versus CON young FVB; +: Significant DE small RNA in young B6J LPHS versus CON; ✳: Significant DE small RNA in CON mature FVB versus CON young FVB AND in young B6J LPHS versus CON].

we propose that the use of noise (or excess variability) actually represents an important phenotyping strategy. Paternally-induced high embryo-to-embryo variation in expression was prevalent across our paradigms, and could help explain observations of partial penetrance in intergenerational studies. We also envision that deploying other modalities as readouts of early F$_1$ phenotypes, such as metabolomics or proteomics, will capture a fuller spectrum of F$_1$ effects. Finally, our data indicate it is critically important to

ensure intergenerational readouts are sufficiently powered, especially given the subtle impact of paternal effects. Smaller-scale experiments run a risk of under-sampling the distribution of molecular/phenotypic outcomes, and consequently over- or underestimating penetrance and effect size.

Thus, we propose some recommendations that could strengthen the conclusions of intergenerational studies: (i) it is important to profile high numbers of F$_1$ embryos/progeny, across multiple experimental batches

with multiple fathers, to maximise signal-to-noise ratio, (ii) examining several genetic backgrounds reveals both biological insights for iGxE and generalisability of transmitted effects, and (iii) partial phenotypic penetrance (or excess variability) could serve as an important strategy for observing intergenerational effects. In conclusion, our study provides evidence for a molecular signature of paternal environmental exposures at the earliest stages of offspring development, and suggests a template for future studies to achieve a mechanistic understanding of mammalian intergenerational epigenetic inheritance.

# Methods

## Exposures and embryology

### Animal husbandry

All experiments involving mice were carried out in accordance with the approved protocols and guidelines by the laboratory animal management and ethics committee of the European Molecular Biology Laboratory (EMBL) under license no. 22017RMJH and 23009RMAB. The inbred FVB, C57BL/6J (B6), and C57BL/6NJ (B6) strains were used in this study. Mice were housed under a 12 h light/dark cycle (light from 07:00 to 19:00), with ad libitum access to food and beverages (see details in the "Mice environmental treatment" 'Methods' section). Mature male mice (1 year) were kept as much as possible with their littermates in the same cage, but

**Reagents and tools table**

| Reagent/resource | Reference or source | Identifier or catalogue number |
|---|---|---|
| **Experimental models** | | |
| FVB (*M. musculus*) | EMBL Rome _ Animal facility, colony production | N/A |
| C57BL/6J (*M. musculus*) | EMBL Rome _ Animal facility, colony production | N/A |
| C57BL/6NJ (*M. musculus*) | EMBL Rome _ Animal facility, colony production | N/A |
| **Antibodies** | | |
| Rabbit αCDX2 | Abcam | Ab76541 |
| Goat αSOX17 | R&D | AF1924 |
| **Oligonucleotides and other sequence-based reagents** | | |
| Oligo-dT | This study | 5'AAGCAGTGGTATCAACGCAGAGTA CTTTTTTTTTTTTTTTTTTTTTTTTTTTTTTVN -3' |
| Template switching oligo | This study | 5'-AAGCAGTGGTATCAACGCAGA GTACATrGrG+G-3' |
| IS PCR Primer | This study | 5'-AAGCAGTGGTATCAAC GCAGAGT-3' |
| Tn5ME-A | This study | 5'-TCGTCGGCAGCGTCAGATGTGT ATAAGAGACAG-3' |
| Tn5ME-B | This study | 5'-GTCTCGTGGGCTCGGAGATGTGT ATAAGAGACAG-3' |
| Tn5MErev | This study | 5'-[phos]CTGTCTCTTATACACATCT-3' |
| **Chemicals, enzymes and other reagents** | | |
| HTF media | Millipore | MR-070-D |
| MBCD media | Sigma-Aldrich | C4555 |
| Mineral oil (need to be batch tested) | Sigma | M8410 |
| KSOM media | Millipore | MR-106-D |
| GSH: L-Glutathione reduced | Sigma | G6013 |
| M2 media | Sigma | M7167-50ML |
| Acid tyride's solution | Sigma | T1788 |
| Vectashield containing DAPI | Novus Biologicals | H-1200-NB |
| RNase OUT 0.5U/μL | Invitrogen | 10777-019 |
| SuperScript IV RT 200 U/μL | Thermo Scientific | 18091050 |
| Betaine 5 M | Sigma | B03000 1VL |

| Reagent/resource | Reference or source | Identifier or catalogue number |
|---|---|---|
| RNAse inhibitor | Clontech Takara | 2313A |
| KAPA HiFi HotStart ReadyMix | Thermo Scientific | 18090050 |
| Nextera XT Index Kit v2 Set B, REF | Illumina | 15052164, LOT 20756419 |
| mirVana miRNA isolation kit | Thermo Fisher Scientific | AM1560 |
| NextFlex Small RNA-Seq Kit v4 | PerkinElmer | N/A |
| Trizol | Thermo Scientific | 15596026 |
| **Software** | | |
| COPAS vision FlowPilot (version 3.091) | Union Biometrica | NA |
| bedtools (version 2.31.1) | Quinlan AR and Hall IM, 2010 | NA |
| bowtie (version 1.0.0) | Langmead B et al, 2009 | NA |
| DESeq2 (version 1.38.3) | Love MI et al, 2014 | NA |
| fastqc (version 0.11.8) | Andrews S, 2010 | NA |
| multiqc (version 1.15) | Ewels P et al, 2016 | NA |
| R (version 4.2.2) | R Core Team (2022) | NA |
| samtools (version 1.15.1) | Danecek P et al, 2021 | NA |
| star (version 2.7.11a) | Dobin A et al, 2013 | NA |
| subread (version 2.0.6) | Liao Y et al, 2014 | NA |
| wiggletools (version 1.2.1) | Zerbino DR et al, 2014 | NA |
| **Other** | | |
| Skirted 96-well plate | Fisher Scientific | E951020401 |
| DNA SPRI beads | Beckman Coulter | B23318 |
| BeadBug™ 6 microtube homogenizer | Merck | Z742684 |
| Accessories Beads for BeadBeater® Zirconia/glass beads, 0,1 mm | Roth | N033.1 |
| Nextseq2000 sequencer | Illumina | N/A |
| NextSeq500 sequencer | Illumina | N/A |

isolated upon signs of aggression. Animals were sacrificed by cervical dislocation for all experiments.

### Mice environmental treatment

At the onset of each treatment, male mice were randomly allocated to one of three groups: Control (CON), non-absorbable antibiotics (nABX), and low protein high sugar diet (LPHS). Each group received the appropriate exposure regimen after 3 weeks or 12–13 months, respectively, for young or mature mice (Figs. 1A and 4A). CON cages were supplied with water and a rodent standard diet (protein 18,5%, sucrose 3,7%, see extensive Mucedola composition in Dataset EV19), nABX cages received non-absorbable antibiotics in drinking water (Neomycin trisulfate salt hydrate (Sigma N1876) (final concentration: 2.5 mg/mL), Bacitracin (Sigma B0125) (final concentration: 2.5 mg/mL), and Pimaricin (Sigma P9703) (final concentration: 1.25 μg/mL)), in addition to the standard diet (composition provided above). Finally, LPHS cages were provided with water and a western-style, low-protein high-sugar (LPHS) diet (sucrose 19%, casein 9.2%, based on ref: Carone, Cell, 2010, Dataset EV19). Mice were monitored daily for well-being and weighed weekly. Following 6–7 weeks of continuous treatment, mice were collected for downstream experiments.

### In vitro fertilisation (IVF)

In vitro fertilisations were performed based on published protocols, with some modifications to optimise throughput and equalise developmental timing (Guan et al, 2014). Briefly, cauda epididymis and vasa deferentia from males were dissected and sperm was gently squeezed/dissected out into preequilibrated capacitation media (HTF supplemented with MBCD at final concentration of 0.75 mM) under mineral oil (HTF: Millipore MR-070-D; MBCD: Sigma-Aldrich C4555; mineral oil: sigma M8410, need to be batch tested), and allowed to swim up for 1 h. In the meantime, cumulus-oocyte complexes were collected from superovulated females in the fertilization drop containing KSOM media with GSH (final concentration 1 mM) under mineral oil (KSOM: Millipore MR-106-D; GSH: ʟ-Glutathione reduced, Sigma G6013). Superovulated females (6–12 weeks for FVB and 3–4 weeks for B6) were intraperitoneal injected with 5–10 IU of PMSG and 5–10 IU of hCG 64 h and 16 h before oocyte collection, respectively. Then,

sperm were counted, and cumulus-oocyte complexes were inseminated with 0.2 million sperm in a 200 μl fertilization drop (KSOM + GSH). Co-incubator proceeded for four hours in a low oxygen condition (hypoxic) incubator (5% $CO_2$, 5% $O_2$, 37 °C). Following co-incubation, zygotes were cleaned from the surrounding cumulus cells and sperm by 5–8 washes in KSOM covered by mineral oil, then cultured in KSOM in a 96-well plate (without mineral oil) until the blastocyst stage (92 h+/−2 h after completed fertilization) in low oxygen condition incubator (5% $CO_2$, 5% $O_2$, 37 °C) or regular condition incubator (5% $CO_2$, 20% $O_2$, 37 °C). As a side note, the genetic background of the female always matched the genetic background of the male (e.g., for the paternal condition involving mature FVB, *IVF* was conducted using FVB oocytes).

### Blastocyst analysis with COPAS vision

FVB control and treated blastocysts were generated by *IVF* (as described) and cultured in a normoxic incubator (5% $CO_2$, 20% $O_2$, 37 °C). Blastocysts were washed in 1× PBS, fixed in 4% PFA for 20 min at 37 °C (with their zona pellucida), and transferred in a 50 mL Falcon tube containing 20 mL of 1× PBS. Then, blastocysts were resuspended in solution by vigorously shaking the Falcon tube immediately before starting the analysis by the COPAS vision. Analyses were conducted with a flow cell of 500 μm, and 8-bit images were captured for each object. Blastocysts were characterised based on Time Of Flight (TOF) and Extinction (optical density, meaning light reflection capacity). The COPAS vision software (FlowPilot 3.091) automatically computed various morphological features of the blastocyst embryo, including perimeter in μm, area in μm², roughness in arbitrary unit (A.U.) (roughness represents the deviation from 1 with 1 being completely round); and mean grayscale in A.U. (for 8-bit images, grayscale should be an integer value between 0 and 255 representing the grey level of the pixel with 0 being black and 255 white).

### Immunofluorescence on blastocysts

FVB blastocysts were generated by IVF and cultured in normoxic conditions. Blastocysts were washed twice in warm M2 media (Sigma M7167-50ML), and zona pellucida was removed by a quick incubation in warm acid Tyrode's solution, followed by 2× rinse in M2 media to neutralise the acid. Embryos were fixed in 4% PFA (in PBS) for 20 min at 37 °C and then transferred to 1× PBS. Subsequently, embryos were permeabilized in 0.5% Triton-X100 for 20 min at RT, washed three times in 1× PBS. Blocking of unspecific epitopes was performed in 3% BSA for 2–3 h at RT. Embryos were incubated overnight in primary antibody solution 3% BSA (αCDX2 – Abcam Ab76541 1:200; αSOX17 – R&D AF1924 5:100). The following day, the embryos were washed three times in 1× PBS, incubated with secondary antibodies (dilution 1:500) for 1 h at RT, washed three times in 1× PBS and mounted in drops of Vectashield containing DAPI (Novus Biologicals H-1200-NB) on glass-bottom dishes and imaged. Finally, the embryos were imaged (inside the drops of Vectashield) using a confocal microscope. Positive cells for DAPI, CDX2 and SOX17 were manually counted.

## Omics

### Single embryo RNA-sequencing

Single-embryo RNA-sequencing was performed based on the previously published Smart-seq protocol (Hennig et al, 2018; Picelli et al, 2014; Trombetta et al, 2014) with some modifications outlined below.

*Embryo collection for Smart-Seq.* Blastocysts were generated from *IVF* and cultured in a hypoxic environment (5% $CO_2$, 5% $O_2$, 37 °C) for all experiments and conditions, except for comparison of young and mature C57BL/6N (B6) paternal states (Fig. 4E,F), where normoxic incubation was performed (5% $CO_2$, 20% $O_2$, 37 °C). Embryos were washed 2–3 times in warm M2 media (Sigma M7167), and individually mouth pipetted into 5 μL of collection buffer (Trition-X-100 0.5%; RNase OUT 0.5 U/μL - Invitrogen 10777-019; dNTPs 2.5 mM and oligo-dT 2.5 μM (5′-AAG-CAGTGGTATCAACGCAGAGTACTTTTTTTTTTTTTTTTTTTTT TTTTTTTTTTTVN-3′) in RNase-free water. Embryos were stored in the collection buffer at −80 °C prior to library preparation.

*Smart-seq2 library preparation (protocol 1; all data except* Fig. 4E,F*).* Smart-seq2 library preparation was performed following these steps. The samples were organised into a skirted 96-well plate (Fisher Scientific E951020401), incubated for 3 min at 72 °C, immediately placed on ice, and then centrifuged. The first step was the reverse transcription of mRNA. An RT master mix (final volume: 6.15 μL) was prepared with the following components: 2 μL 5× SuperScript IV buffer (Thermo Scientific 18091050), 0.50 μL DTT 100 mM, 2 μL Betaine 5 M (Sigma B03000 1VL), 0.1 μL $MgCl_2$ 1 M, 0.25 μL RNAse inhibitor (Clontech Takara 2313 A), 0.25 μL SuperScript IV RT 200U/μL (Thermo Scientific 18091050), 0.1 μL template switching oligo 100 μM (5′-AAGCAGTGGTATCAACG-CAGAGTACATrGrG+G-3′), and 1.15 μL of water. This master mix was added to each well, containing a single embryo in 5 μL of collection buffer. Each well was thoroughly mixed by pipetting, sealed, centrifuged, and incubated in a thermocycler under the following conditions: 42 °C for 15 min, 80 °C for 10 min, then stored at 10 °C.

The second step, involving whole transcript amplification (WTA) and post-PCR cleanup, was immediately performed. The plate was unsealed, and a WTA mix was added to each sample (final volume: 15 μL; 12.5 μL KAPA HiFi HotStart ReadyMix 2× (Roche); 0.20 μL IS PCR Primer 5 μM (5′-AAGCAGTGGTAT-CAACGCAGAGT-3′), and 2.30 uL water). The samples were mixed well by pipetting, sealed, and incubated in a thermocycler under the following conditions: Initial step: 98 °C for 3 min; 17–18 cycles: 98 °C for 20 s, 67 °C for 15 s, 72 °C for 6 min; Extension: 72 °C for 5 min; and then storage at 10 °C. The PCR products were then cleaned up using DNA SPRI beads (Beckman Coulter) at a 0.6 volume ratio. The mixture was incubated 5 min at RT with the beads, separated using a magnet, and immediately eluted in 13 μL of water. As a note, the usual ethanol bead wash steps were not necessary since the cDNA was highly diluted, and washing seemed to decrease the cDNA yield. The cDNA concentration and profile were verified using an HS DNA Bioanalyzer chip. For the young FVB paternal condition, samples from the same 96-well plate were pooled and run only once in the Bioanalyzer chip. The median concentration was calculated and applied for the later dilution of all samples from this plate. For the mature FVB and young B6J paternal condition, each cDNA sample was quantified one by one and diluted.

During the third step, called tagmentation, homemade reagents were used in the process.

First, the linker oligonucleotides were annealed. In details, the oligos Tn5ME-A (5′-TCGTCGGCAGCGTCAGATGTGTATAAG

AGACAG-3′), Tn5ME-B (5′-GTCTCGTGGGCTCGGAGATGTG-TATAAGAGACAG-3′), Tn5MErev (5′-[phos]CTGTCTCTTATA-CACATCT-3′) were resuspended in an annealing buffer (50 mM NaCl, 40 mM Tris-HCl pH8.0); equally mixed (Tn5ME-A + Tn5MErev; or Tn5ME-B + Tn5MErev) at a final concentration of 50uM; annealed in the thermocycler for 5 min at 95 °C, slowly cool down to 65 °C (0.1 °C/s), 5 min at 65 °C and slowly cool down to 4 °C (0.1 °C/s); and diluted to 35 µM in water.

Second, the Tn5 was loaded with the annealed linkers. Homemade Tn5$_{(R27S),E54K,L372P}$ was diluted at 0.5 mg/mL. Annealed linkers (0.7 µL at 35 µM) were mixed with 10 µL of diluted Tn5 (0.5 mg/mL), incubated at 23 °C under constant shaking at 350 rpm for 30–60 min and store on ice. Finally, loaded Tn5 was diluted in water of a final dilution 1:200 to 1:300.

Third, cDNA was diluted to 100 pg/µL to 200 pg/µL and 1.25 µL of diluted cDNA was transferred into a new 96-well plate. Tagmentation reaction was added to each sample 2.5 µL of tagmentation mix (1 volume of 4× tagmentation buffer (40 mM Tris-HCl pH 7.5, 40 mM MgCl$_2$) and 1 volume of 100% DMF (Sigma Aldrich)), 1.25 µL of Tn5 (1:200 to 1:300), mixed well (final volume 5 µL); incubated at 55 °C for 3 min and place immediately on ice. The reactions were stopped by adding 1.25 µL of 0.2% SDS, incubated 5 min at RT, and stored on ice.

Final PCR was performed. In details, 10uL of the following PCR mix was added to each samples (6.75 µL of 2× KAPA buffer (Roche), 0.75 µL of 100% DMSO, 1.25 µL of 10 µM i5 adaptor index primer, and 1.25 µL of 10 µM i7 adaptor index primer) and incubated in the thermocycler under the following conditions: 72 °C for 3 min; 95 °C for 30 s; 12 cycles: 98 °C for 20 °C, 58 °C for 15 s, 72 °C for 30 s; 72 °C for 3 min; and store at 10 °C. Then, all samples were pooled (5–8 µL/sample), and the final pool was cleaned with 0.8x SPRI beads and eluted in 12 µL. The concentration and quality of the pooled library were assessed using the Agilent HS DNA BioAnalyzer D1000 and the Qubit dsDNA HS assay.

*Smart-seq library preparation (protocol 2) – (*Fig. 4E,F*).* Young and mature B6N paternal condition Smart-seq library preparation was made only using commercial reagents. Briefly, the samples were organised into a skirted 96-well plate (Fisher Scientific E951020401), incubated for 3 min at 72 °C, immediately placed on ice, and then centrifuged (all centrifugation steps were done at 800× *g* for 1 min). The first step was the reverse transcription of mRNA. An RT master mix (final volume: 6.65 µL) was prepared with the following components: 2 µL Betaine 5 M (Sigma B03000 1VL), 2 µL 5× SuperScript IV buffer (Thermo Scientific 18090050), 0.25 µL DTT 100 mM, 0.9 µL MgCl2, 1 µL template switching oligo 10 µM (5′-AAGCAGTGGTATCAACGCAGAGTACATrGrG+G-3′), and 0.5 µL SuperScript IV RT 200U/µL (Thermo Scientific 18090050). This master mix was added to each well, which already contained a single embryo in 5 µL of collection buffer. Each well was thoroughly mixed by pipetting up and down, sealed, centrifuged, and incubated in a thermocycler under the following conditions: initial step: 42 °C for 90 min; 10 cycles: 50 °C for 2 min and 42 °C for 2 min; inactivation: 70 °C for 15 min; and then stored at 4 °C.

Next, the second step, which involved whole transcript amplification (WTA) and post-PCR cleanup, was performed. The plate was unsealed, and a WTA mix was added to each sample (final volume: 14 µL; 13 µL KAPA HiFi HotStart ReadyMix

(Thermo Scientific 18090050); 1 µL IS PCR Primer 10 µM (5′-AAGCAGTGGTATCAACGCAGAGT-3′)). The samples were mixed well by pipetting, sealed, and incubated in a thermocycler under the following conditions: Initial step: 98 °C 3 min; 20 cycles: 98 °C 15 s, 67 °C 20 s, 72 °C 6 min; Extension: 72 °C 5 min; and then storage at 4 °C. The PCR products were then cleaned up using DNA SPRI beads (Beckman Coulter B23318) at a 0.8 volume ratio with 80% EtOH. The elution step was performed in 20 µL of clean water, and the eluents were transferred to a new skirted 96-well plate. Each sample was quantified using the Qubit dsDNA HS Assay Kit with Qubit assay tubes and then diluted to 0.5 ng/µL in clean water in a new skirted 96-well plate.

The third step involved constructing the Nextera XT sequencing library (Illumina 15032354). Firstly, the i5 and i7 primers were combined in a 96-well plate as detailed in the commercial protocol. Secondly, the tagmentation reaction was performed. In a new skirted 96-well plate, the tagmentation mix (2.5 µL TD buffer and 1.25 µL ATM) was combined with 1.25 µL of diluted product from samples (concentration 0.5 ng/µL) and mixed well. Tagmentation was carried out in a thermocycler for 10 min at 55 °C and then cooled down to 10 °C. As soon as the tagmentation reaction reached 10 °C, 1.25 µL of NT buffer was added to each well and incubated for 5 min at room temperature to stop the reaction. Thirdly, indexes were incorporated (Nextera XT Index Kit v2 Set B, REF 15052164, LOT 20756419), and the products were amplified. In detail, each sample received 3.75 µL of NPM and 2.5 µL of the specific combined index primer solution. The samples were mixed well by pipetting, centrifuged, and incubated in a thermocycler with the following programme - initial annealing: 72 °C 3 min; denaturation: 95 °C 30 s; 12 cycles: 95 °C 10 s, 50 °C 30 s, 72 °C 1 min; extension: 72 °C 5 min; and then storage at 4 °C. The final step is the DNA pooling and cleanup. Briefly, 2.5 µL of each sample was pooled into a 1.5-mL Eppendorf tube, washed with DNA SPRI select beads at a 0.9 volume ratio with 80% EtOH, and then eluted in 30 µL of clean water. This washing step was repeated again with DNA SPRI select beads at a 0.9 volume ratio with 80% EtOH, and the DNA was eluted in 30 µL of clean water. Finally, the quality and the concentration of the pooled library were assessed using the Agilent HS DNA BioAnalyzer D1000 and the Qubit dsDNA HS assay.

### Smart-seq library sequencing

After Smart-seq library preparation of single embryos, the indexed pooled samples were sequenced together in one or two runs using the NextSeq 2000 sequencer with a P3 or P4 (from FVB young and mature sperm donors on control conditions) flow cell in 50 bp paired-end mode. Numbers of obtained reads after QC are available in Figs. EV1J and EV4G.

### Testis RNA-seq

Whole testes from FVB, C57BL/6J control, and treated mice were collected, snap-frozen in liquid nitrogen, and stored at −80 °C. Testes were then homogenised in Trizol using a BeadBug™ 6 microtube homogenizer with 1 mm glass beads (Accessories Beads for BeadBeater® Zirconia/glass beads, 0.1 mm—Roth No. N033.1). Total RNA was extracted according to the manufacturer's recommendations. RNA was quantified using Nanodrop and RNA TapeStation kit. RNA-seq libraries were performed with a starting material ranging from 100 to 500 ng/µL of total RNA using

the NEBNext Ultra II directional RNA kit with the Beckman Coulter hybrid workstation i7 (liquid handling robot). Adaptors were diluted 1 in 30. Libraries were amplified through 11 PCR cycles and sequenced with a 40 bp, pair-end reading mode on a NextSeq500 sequencer.

### Sperm small RNA-seq

*Sperm collection.* The cauda epididymides from FVB, C57BL/6J control, and treated male mice were dissected and placed into a 1.5-mL Eppendorf tube containing warm M2 media (Sigma M7167). The caudae and M2 media were then transferred into a 6 cm dish and gently cleaned of any surrounding tissue or fat. After cleaning, the caudae were briefly washed in clean M2 and placed in a 3.5-cm dish containing 1–1.5 mL of M2 media. The caudae were then cut open ~20 times with scissors and incubated for 30 min at 37 °C, 5% $CO_2$, to allow spermatozoa to swim out. Under the binocular loupe, the areas of tissue still containing spermatozoa were identified and pinched with needles to release the remaining spermatozoa and incubated for an additional 10 min at 37 °C, 5% $CO_2$. Following incubation, the M2 media containing the spermatozoa was pipetted up and down a few times and transferred into a fresh 1.5-mL Eppendorf. It is important to perform this procedure carefully to avoid excessive disruption of the tissue and, therefore, prevent somatic cell contamination of the sperm collection. The sperm solution was filtered through a 40-µM filter to remove any tissue fragments. The filtered solution was then centrifuged at 8500 rpm, 5 min, 4 °C. The pellet was resuspended in 1 mL of somatic cell lysis buffer (0.1% SDS, 0.5% Triton-X-100, in water) and incubated on ice for 20 min. Following somatic cell lysis, the sperm solution was pelleted (first at 8500 rpm and then at 13500 rpm, both for 5 min at 4 °C) and resuspended in 1 mL of 1× PBS each time. Finally, the spermatozoa were pelleted again (13,500 rpm, 5 min, 4 °C), quickly dried, and frozen at −80 °C for storage.

*Sperm total RNA extraction.* Sperm RNA extraction was performed using an optimised version of mirVana miRNA isolation kit (Thermo Fisher Scientific AM1560). Spermatozoa were first resuspended in 600 µL of mirVana lysis buffer and transferred into a screw-cap tube containing 1 mm beads (Accessories Beads for BeadBeater® Zirconia/glass beads, 0.1 mm—Roth No. N033.1). The spermatozoa were homogenised for 60 s at 4000 rpm for a total of five cycles using the BeadBug™ 6 microtube homogenizer. Samples were centrifuged at RT, 5000 rpm, 5 min, and cooled on ice for 10 min to complete sperm lysis. Avoiding bead collection, the solution was transferred to a new 1.5-mL Eppendorf tube. Following this, the "organic extraction" and "Total RNA isolation procedure" sections from the mirVana protocol were precisely followed. Briefly, the extracted RNA was incubated on ice for 10 min with 1/10 volume of miRNA Homogenate Additive and isolated with an Acid-Phenol: Chloroform step. The aqueous phase containing the RNA was recovered and mixed with 1.25 volume of 100% RT ethanol. The lysate/ethanol mixtures were applied to a filter cartridge and washed three times. RNA was eluted with 50 µL of pre-heated nuclease-free water (95 °C). RNA concentration and profile were measured using the RNA High Sensitivity TapeStation kit (measured RNA concentration between 1000 and 8000 pg/µL). Sperm RNA can be stored at −80 °C for weeks to months.

*Small RNA library construction and sequencing.* Libraries were constructed using the NextFlex Small RNA-Seq Kit v4 by PerkinElmer. The starting material for the libraries varied in concentration for each sample, while the volume remained consistent across all samples. Libraries were sequenced at 100 bp, single end using a Nextseq2000 sequencer, yielding between 100,000 and 7 million reads per sample, with the vast majority of the sample containing ~1–5 million reads.

## Single-embryo transcriptomics analysis

### Constructing the reference genome

We used STAR (version 2.7.11a) *genomeGenerate* function to modify the default mm39 genome (which is based on a B6 background) by adding an FVB-specific SNP mask obtained from dbSNP1 (Sherry et al, 2001) lifted over to the mm39 genome. To map transcriptomic sequences obtained from B6J and B6N backgrounds, the mm39 genome was used as is. TE annotations were obtained from UCSC RepeatMasker and added to the respective genome's GTF file prior to running *genomeGenerate*. Key redefined parameters:

- --genomeTransformType Haploid
- --sjdbOverhang 99
- --genomeChrBinNbits 15

### Mapping and alignment of single-embryo transcriptomics data

Raw reads were mapped to the FVB SNP-masked or standard mm39 genome with TE annotations using STAR (version 2.7.11a). To include transposable elements in our dataset, we followed recommendations from Teissandier et al, 2019. Briefly, we allowed for multi-mapping up to a maximum of 5000 candidate loci per read and randomly assigned the mapped read to any of its candidate loci. Key redefined parameters:

- --outFilterMultimapNmax 5000
- --outFilterMismatchNoverLmax 0.06
- --winAnchorMultimapNmax 5000
- --outSAMmultNmax 1

### Quantification of reads from genes and transposable elements

All samples were filtered for a minimum of 1 million mapped reads, except in the case of mature vs young FVB (minimum 0.5 million reads). Read quantification was performed using featureCounts (v2.0.6) separately for genes (at default settings) and transposable elements (with "-M" parameter to include multi-mapped reads).

### Principal component analysis (PCA)

All downstream analysis was performed using R (4.2.2). Briefly, counts were normalised to counts-per-million (CPM) using the edgeR (3.40.2) package's *cpm* function and subsequently $log_2$-transformed with a pseudocount of 1. Log-normalised counts were then input into the *prcomp* function of the irlba (2.3.5.1) package at default parameters. Density plots lining the PCA plots were generated using the *geom_density* function of ggplot2 (3.5.1).

### Differential expression analysis

Differential expression analysis was performed using DESeq2 (1.38.3). For all analyses, paternal condition ("Condition") was the only term used in the design formula; see GitLab link ('Code availability' section) for full code and parameters. Genes showing an adjusted *P* value < 0.05 and log2FoldChange > 0.5 were identified as DE genes in all cases. When comparing young vs mature F1

blastocysts (in both B6 and FVB backgrounds), we used an adjusted $P$ value < 0.05 without log2FoldChange threshold in order to capture highly significant but low effect-size hits. Overrepresentation analysis to obtain Gene Ontology terms amongst DEG (Fig. 1G,H) was performed using String v12.0.

### Clustering of DE gene sets

Expression of DE genes obtained from both nABX and LPHS conditions in the young FVB background was averaged over all samples in the same condition. For each gene, the averaged expression in each condition was then scaled by its average expression in Ctrl samples. We then constructed a gene-gene Euclidean distance matrix using *dist* with *method = "euclidean"*, and clustered the distance matrix by complete linkage hierarchical clustering using *hclust* with *method = "complete"*. The *cutree* method was used to cut the dendrogram and obtain distinct clusters, which were then manually merged based on directionality of expression relative to controls to obtain the 8 clusters in Fig. 1I. Overrepresentation analysis to identify enriched Gene Ontology (GO) terms in DE gene sets was performed using Metascape (v3.5.20240101) "Express Analysis" mode at default settings (Zhou et al, 2019).

### Gene set enrichment analysis (GSEA)

GSEA was performed using clusterProfiler (4.6.2). Briefly, genes were ordered based on log2FoldChange output from the differential expression analysis above with *nPermSimple = 10000* & *pAdjustMethod = "BH"*.

### Construction of a single-cell reference and computing correlations

Single-cell transcriptomic profiles were downloaded from Deng et al, 2014. Specifically, mid blastocyst (GSM1112706 - GSM1112765) and late blastocyst (GSM1112664 - GSM1112693) profiles were downloaded and subset for expression of lineage markers. Lineage marker expression profiles of single-cells were centred and scaled, then used to compute a cell–cell distance matrix, which was then clustered using *hclust*. Cells belonging to epiblast and trophectoderm lineages were obtained by cutting the dendrogram using *cutree* while primitive endoderm cells were annotated based on expression of primitive endoderm-specific markers. Single-cell expression profiles were then averaged over cells of the same cluster to obtain an average expression signature of lineage-specific marker genes for each lineage. Expression of our single-embryo data over the lineage markers was then correlated (Spearman correlation) with the average expression signatures of each lineage to obtain the correlation coefficients observed in Fig. 2A.

### Profiling binding of transcription factors in our DE gene sets

Binding of transcription factors to DEG sets identified in nABX- or LPHS-derived blastocysts was performed using the ChEA module (2022) in the Enrichr tool (Kuleshov et al, 2016). In brief, this tests for enrichment of transcription factor occupancy in genesets, based on direct ChIP measurement across various cell and tissue types. DEG sets were analysed under default parameters and visualised using the appyter and the volcano plot option.

### Remapping published ChIP-seq datasets to obtain chromatin mark profiles

ICM ChIP-seq FASTQ files were downloaded from Liu et al, 2016 (H3K4me3 and H3K27me3), Wang et al, 2018 (H3K9me3), and Xu et al, 2019 (H3K36me3). Raw reads were trimmed using TrimGalore (v0.6.10) with a quality threshold of 20 to clip the 1 base from the 5' end of both paired reads. Trimmed reads were then mapped to the mm39 reference genome using bowtie2 (v2.4.2). Blacklist regions for mm39 were obtained from excluderange (Dozmorov et al, 2023) EV6 and excluded prior to quantification. Duplicate reads were marked and removed using *MarkDuplicates* (v3.1.0). Finally, coverage files were generated using *bamCoverage* module of deeptools (v3.5.5) with RPKM normalisation and replicates were merged. Enrichment of reads around the TSS of DE genes was computed using *computeMatrix reference-point* mode of deeptools with the following parameters: *-a 5000, -b 5000, -binSize 500*. Heatmaps and enrichment profiles seen in Fig. 2D were plotted using the *plotHeatmap* module of deeptools at default settings.

### In silico developmental staging

Raw FASTQ files of early blastocyst, mid blastocyst and late blastocyst stage single cells were downloaded from Deng et al, 2014, and were remapped and quantified using our pipeline described above. Cells arising from the same embryo were averaged in expression to obtain a pseudo-bulk single-embryo transcriptomic representation. Principal component analysis was applied to these averaged expression profiles to obtain the loadings of the first two components, which captured developmental progression. Our young FVB-derived single-embryo transcriptomic profiles were then multiplied with the loadings of the first two principal components to project our samples onto this space. The resulting projected coordinates of our blastocysts were visualised together with the pseudo-bulk embryos from Deng et al (2014) using ggplot2 (3.5.1).

## Variability analysis

### Calculating the variability of expression of a gene

Our variability analysis is inspired by work from Conine et al, 2020. For each gene, mean expression and coefficient of variation were calculated and a generalised additive model (GAM) was fit to the CV-mean distribution using the *gam* function of the R package mgcv (1.8-42) using the following parameters: *formula = CV ~ s(Mean, bs = "cs", fx = TRUE, k = 20), fit = TRUE, method = "GCV.Cp"*. Expected CV was obtained using the *predict* function of R stats using the following parameters: *se = TRUE, type = "link"*. Variability of a gene was calculated using the following formula:

$$Variability = \log_2 \left( \frac{Observed\ CV}{Expected\ CV} \right)$$

All overrepresentation analyses were performed as described above.

## Testis transcriptomics analysis

### Mapping and alignment of testis transcriptomics data

Testis transcriptomics data were mapped and aligned as described above for the single-embryo transcriptomic data.

### Quantification of reads from genes and transposable elements

Read quantification was performed as for single-embryo transcriptomics data with one key difference—counting was performed in a strand-specific manner using the parameter *-s 2*.

PCA, differential expression analysis and GSEA were performed as described above for the single-embryo transcriptomic data.

## Analysis of small RNA profiles from sperm

### Trimming, filtering, QC, mapping and alignment of small RNA sequencing data

Raw reads were trimmed using TrimGalore (v0.6.10) in smallRNA mode with quality threshold of 20 and restricting length of reads from 18 to 40 bases. Key redefined parameters:

- --small_RNA
- --quality 20
- --length 18
- --max_length 40

Trimmed reads were mapped using bowtie (v1.0.0) allowing for no mismatches and reporting only a single best alignment per read. Key redefined parameters:

- -v 0
- -k 1
- --best.

### Quantification of reads from miRNAs and tRNAs

miRNA annotations were obtained from miRBase (Kozomara et al, 2019) and lifted over to mm39 using CrossMap (v0.7.0) at default settings. tRNA annotations for the mm39 genome were obtained from GtRNAdb (Chan et al, 2016). miRNA and tRNA quantification were performed by counting intersections between the respective reference locations and locations of trimmed reads on the mm39 reference genome using bedtools (v2.31.1) intersect at default settings.

### Differential expression analysis

Differential expression analysis was performed as described above for single-embryo transcriptomics data.

## Data availability

All data supporting the findings of this study have been deposited into ArrayExpress and European Nucleotide Archive (ENA) under the following accessions: E-MTAB-14635; E-MTAB-14636; E-MTAB-14637; E-MTAB-14638; E-MTAB-14639; E-MTAB-14640; PRJEB88453. All code used for data analysis in this study is documented and freely available on GitLab: https://git.embl.de/grp-hackett/iei-transcriptomics. The variability analysis module has been provided as a fully documented R package "GenEVA - Gene Expression Variability Analysis" on GitLab.

The source data of this paper are collected in the following database record: biostudies:S-SCDT-10_1038-S44318-025-00556-4.

## Peer review information

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

## Acknowledgements

Funding for this study was supported by the EMBL Human Ecosystems Transversal Theme (59100) and programme funding to JAH and AB. We thank the GEVF and GBCS core facilities for experimental and technical support. We also thank Michael Bonadonna for the support of the Advanced Mobile Lab by EMBL.

## Author contributions

**Mathilde Dura**: Formal analysis; Investigation; Visualisation; Writing—original draft; Writing—review and editing. **Bobby Ranjan**: Data curation; Formal analysis; Visualisation; Writing—original draft; Writing—review and editing. **Joana B Serrano**: Formal analysis; Investigation. **Rossella Paribeni**: Investigation. **Violetta Paribeni**: Investigation. **Laura Villacorta**: Investigation. **Vladimir Benes**: Resources. **Olga Boruc**: Resources. **Ana Boskovic**: Conceptualisation; Supervision; Funding acquisition; Writing—review and editing. **Jamie A Hackett**: Conceptualisation; Formal analysis; Supervision; Funding acquisition; Visualisation; Writing—original draft; Writing—review and editing.

Source data underlying figure panels in this paper may have individual authorship assigned. Where available, figure panel/source data authorship is listed in the following database record: biostudies:S-SCDT-10_1038-S44318-025-00556-4.

## Funding

## Disclosure and competing interests statement

The authors declare no competing interests.

# Expanded View Figures

**Figure EV1.   The impact of specific environmental exposures on paternal physiology and F1 blastocysts.**

(A) Line plot showing growth curves of FVB males (F0 fathers) across CON, nABX and LPHS treatments. Error bars indicate standard deviation of mouse weights at each timepoint. Unpaired t-test; ns: p-value non-significant; *P value < 0.05. (B) Kaplan–Meier plot showing male mice's survival during the 6 weeks of indicated environmental exposure. (C) Testis to bodyweight ratio. Bars represent the median. (D) Representative images of males with open body cavity showing overt physiological responses after 6–7 weeks of indicated treatment. (E) Images of full (top) and empty (bottom) male ceca after 6–7 weeks of indicated treatment. (F) Representative images of F1 FVB blastocyst embryos generated through IVF. (F, G) Analysis of F1 blastocyst embryos with Copas Vision Sorter. Inset are images of blastocysts that had different TOF values demonstrating equivalence. (G) Scatter plot showing the time of flight (TOF) and Extinction of each blastocyst embryo colour by condition and by batch of experiments. Representative images of blastocysts from both groups are shown. No morphological differences were observed between the groups. Violin plot showing quantification of TOF or extinction. Unpaired t-test; ns: P value non-significant. (H) Perimeter, Roughness and mean greyscale of F1 FVB embryos (see 'Methods'). Unpaired t-test; ns: p-value non-significant. (G, H) [#samples: CON = 80, nABX=73, LPHS = 65]. [Box plots: horizontal black line at the centre is the median. The box extends from the first quartile (Q1: 25th percentile) to the third quartile (Q3: 75th percentile). The vertical black line extends from $Q1 - 1.5 \times IQR$ to $Q3 + 1.5 \times IQR$, where IQR = interquartile range. The minima and maxima are not explicitly annotated in these figures.] (I) Bar chart depicting efficiency of blastocyst development used for Smart-seq across multiple batches in the FVB background. (J) Table with numbers of embryos collected for Smart-seq (top) and passed quality thresholds for Smart-seq bioinformatic analysis (bottom). (K) Dot plot showing depth of blastocyst sequencing obtained across batches coloured by paternal condition. (L) Grouped bar chart showing number of male and female blastocysts across batches. (M) Upset plot showing overlap of DE genes between nABX-derived and LPHS-derived blastocysts. *P* values are calculated using Fisher's exact test. (N) Bubble plot showing gene set enrichment analysis (GSEA) in nABX-derived (left) and LPHS-derived (right) blastocysts. + indicates activated pathways upon paternal treatment, − indicates suppressed pathways upon paternal treatment. (O) Principal Component Analysis (PCA) on genes identified to be DE in either nABX-derived or LPHS-derived blastocysts coloured by paternal treatment. Shaded histograms represent relative density of blastocysts along PC1 (top) and PC2 (right).

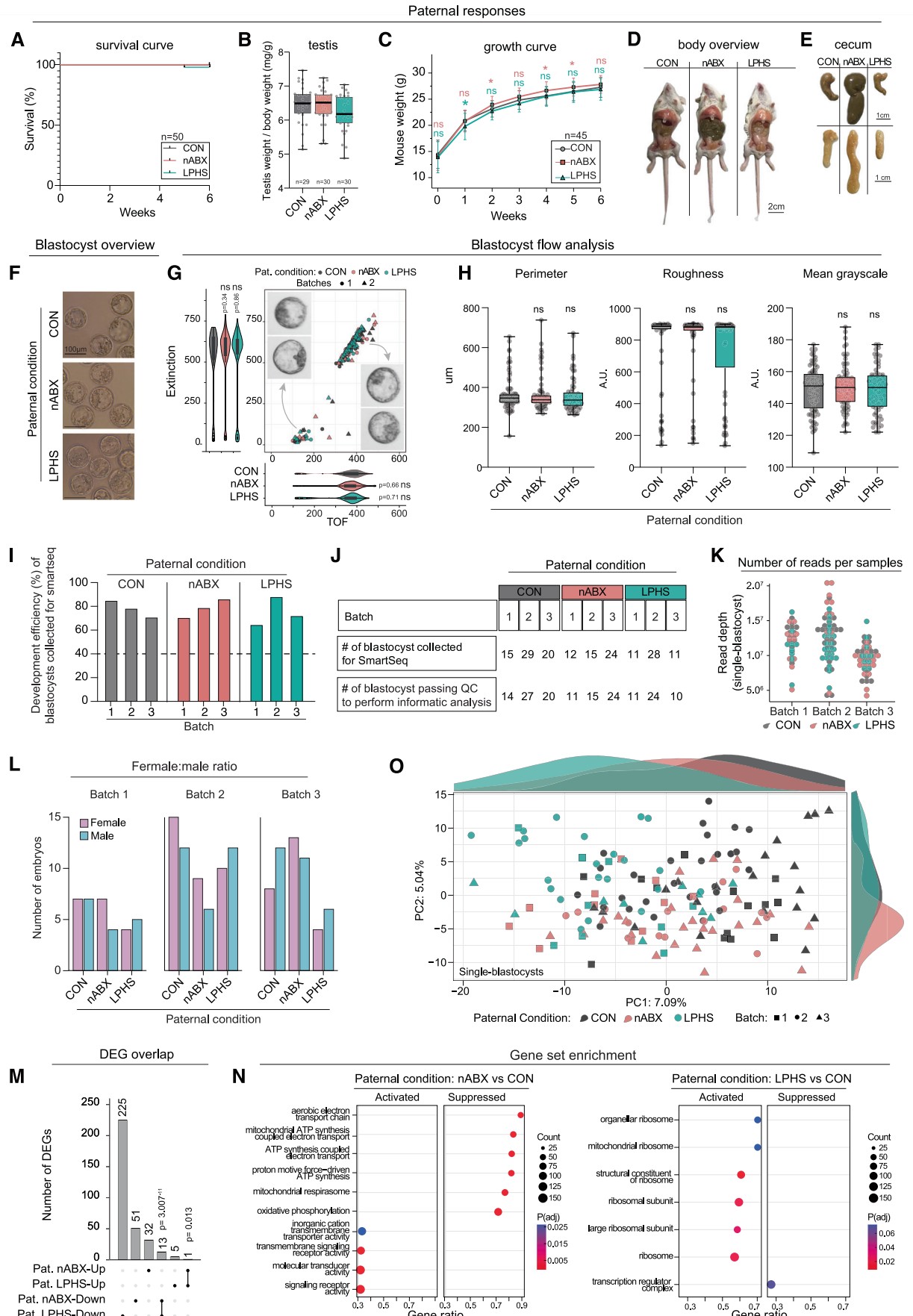

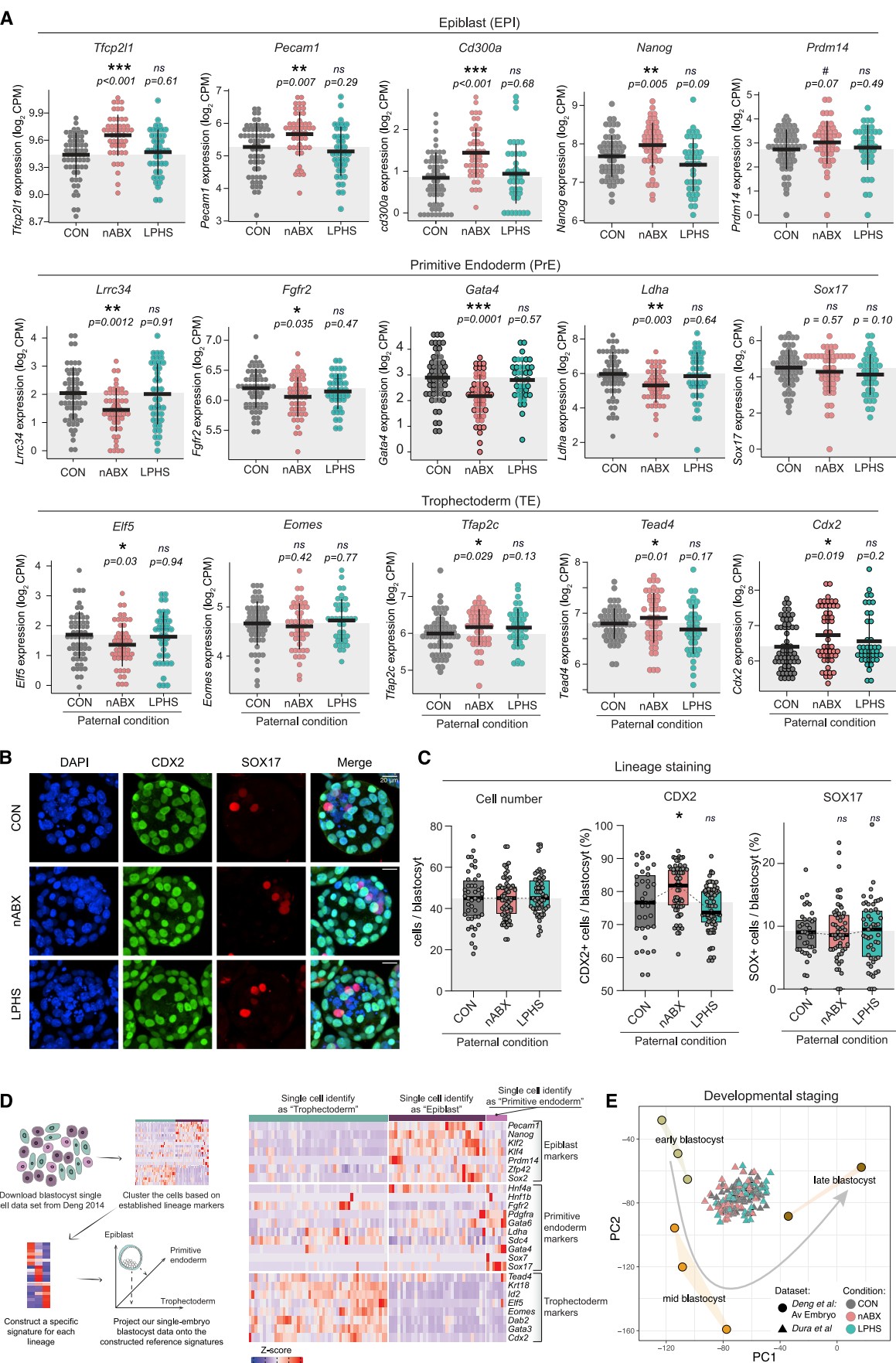

**Figure EV2.  Characterising the IEI lineage-associated impact in F1 blastocysts.**

(A) Dot plots showing expression of lineage-associated DE genes in F1 blastocysts, driven by at least one paternal condition. Each datapoint is a single blastocyst. Shown are classical and regulatory genes for epiblast, primitive endoderm and trophectoderm lineages. *P values* were computed using a two-tailed Wilcoxon test. [*Tfcp21* nABX *P* value = 6e-06; *Cd300a* nABX *P* value = 5.4e-06]. [#samples: CON = 61, nABX=50, LPHS = 41]. The horizontal black line is the median and the vertical black line extends from Q1 − 1.5 × IQR to Q3 + 1.5 × IQR, where Q1 and Q3 refer to the first and third quartiles respectively and IQR = interquartile range (range of 25th to 75th percentile of the distribution). (B) Representative immunofluorescence images of blastocysts sired by control, nABX or LPHS exposed fathers. Staining for CDX2 and SOX17 is shown. (C) Quantification of blastocyst cell number and cell lineages as indicated from two independent batches. Each datapoint is an independent blastocyst. %CDX2 *P* value = 0.034. *P* values were computed using a two-tailed unpaired *t* test. (D) Schematic explaining the strategy to obtain lineage-specific signatures from published single-cell data (Deng, 2014). (E) Projection of our single-blastocyst transcriptomes onto the first two principal components of averaged single-cell transcriptomics data from Deng et al, (2014) (see "Methods": In silico developmental staging).

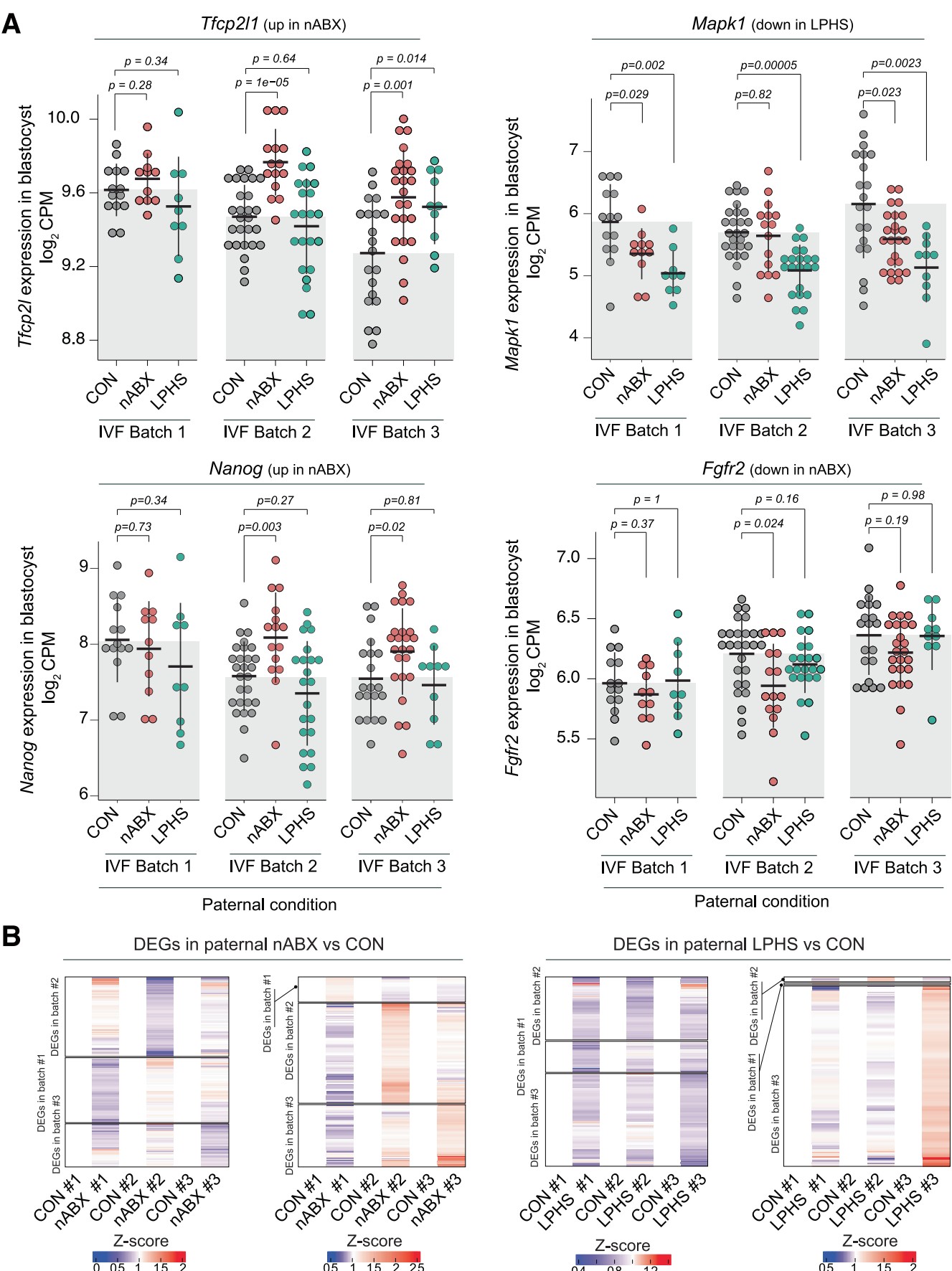

**A**

*Tfcp2l1* (up in nABX)

*Mapk1* (down in LPHS)

*Nanog* (up in nABX)

*Fgfr2* (down in nABX)

Paternal condition

**B**

DEGs in paternal nABX vs CON

DEGs in paternal LPHS vs CON

◀ **Figure EV3. Assessing batch-effects in IVF-derived F1 blastocysts.**

(A) Dot plots showing log expression of genes in in F1 blastocysts derived from the indicated paternal condition, stratified by experimental batch: *Tfcp1l1, Nanog (epiblast markers), Mapk1 and Fgfr2 (signalling components)*. Each datapoint indicates a single blastocyst. *P-values* were computed using a two-tailed Wilcoxon test. [#samples Batch 1: CON = 14, nABX=11, LPHS = 9; Batch 2: CON = 27, nABX=15, LPHS = 22; Batch 3: CON = 20, nABX=24, LPHS = 10]. The horizontal black line is the median and the vertical black line extends from Q1 − 1.5 × IQR to Q3 + 1.5 × IQR, where Q1 and Q3 refer to the first and third quartiles respectively and IQR = interquartile range (range of 25th to 75th percentile of the distribution). (B) Heatmaps showing fold changes of batch-specific DE genes relative to controls across all batches, split by batch of origin of the DE genes.

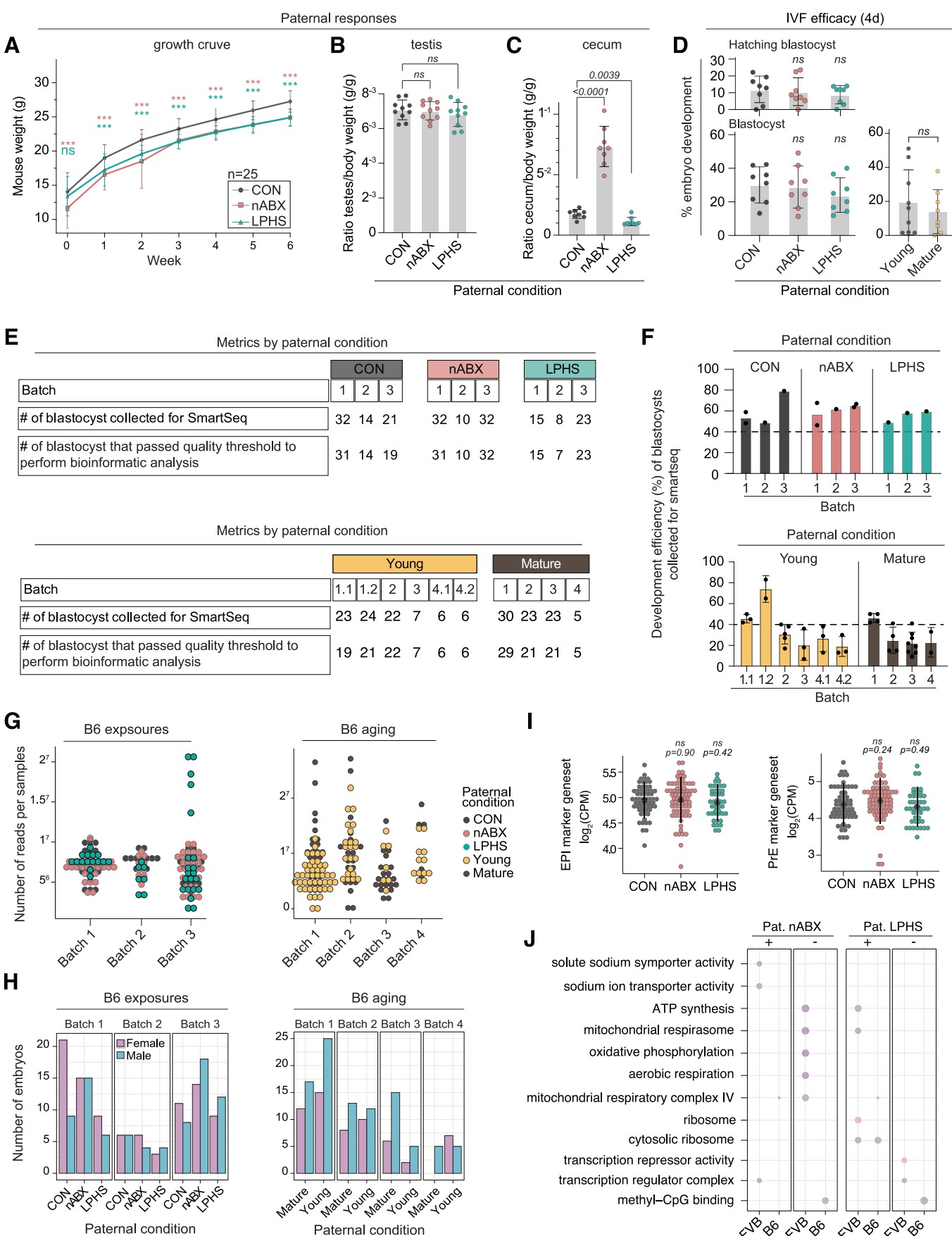

**Figure EV4.   Interrogating the effect of age and genetic background on the IEI response in F1 blastocysts.**

(A) Line plots showing growth curve of B6 males across CON, nABX and LPHS paternal treatments. Error bars indicate standard deviation at each timepoint. (B) Dot and bar plots showing testis/body weight ratio for B6 males after treatment. Error bars indicate standard deviation. (C) Dot and bar plots showing caecum/body weight ratio for young B6 males after treatment. Error bars indicate standard deviation. (D) Dot and bar plots showing rate of F1 blastocyst development and hatching from B6 across paternal treatment (left) and across ages (right). Error bars indicate standard deviation at each timepoint. Bars represent the mean. (E) Tables showing numbers of embryos collected for Smart-seq and passed quality thresholds for Smart-seq bioinformatic analysis from B6 across paternal treatment (top) and across ages (bottom). All batches shown. (F) Bar charts showing developmental efficiency of blastocysts collected for Smart-seq across batches from B6 (top) and across ages (bottom). All batches are shown. Each dot represents the efficiency of blastocyst development in each culture well. Boxes represent the mean, and error bars indicate the standard deviation. (G) Dot plots showing depth of blastocyst sequencing obtained from young B6 fathers across paternal treatment (left), and across paternal ages (right). All batches are shown. (H) Grouped bar chart showing number of male and female blastocysts obtained from young B6 fathers across paternal treatment (left), and across paternal ages (right). All batches are shown. (I) Total expression ($\log_2$CPM) of selected epiblast (EPI) (left) and primitive endoderm (PrE) (right) geneset markers. Markers used for the respective geneset (see Fig EV2D). P-values were computed using a two-tailed Wilcoxon test. (J) Bubble plot showing gene set enrichment analysis (GSEA) comparing the paternal treatments across FVB and B6 backgrounds. (A–D) $P$ values were computed using an unpaired t-test; [ns: not significant; *$P$ value < 0.05; **$P$ value < 0.005; ***$P$ value < 0.0005].

Computation of per-gene expression noise by genetic background

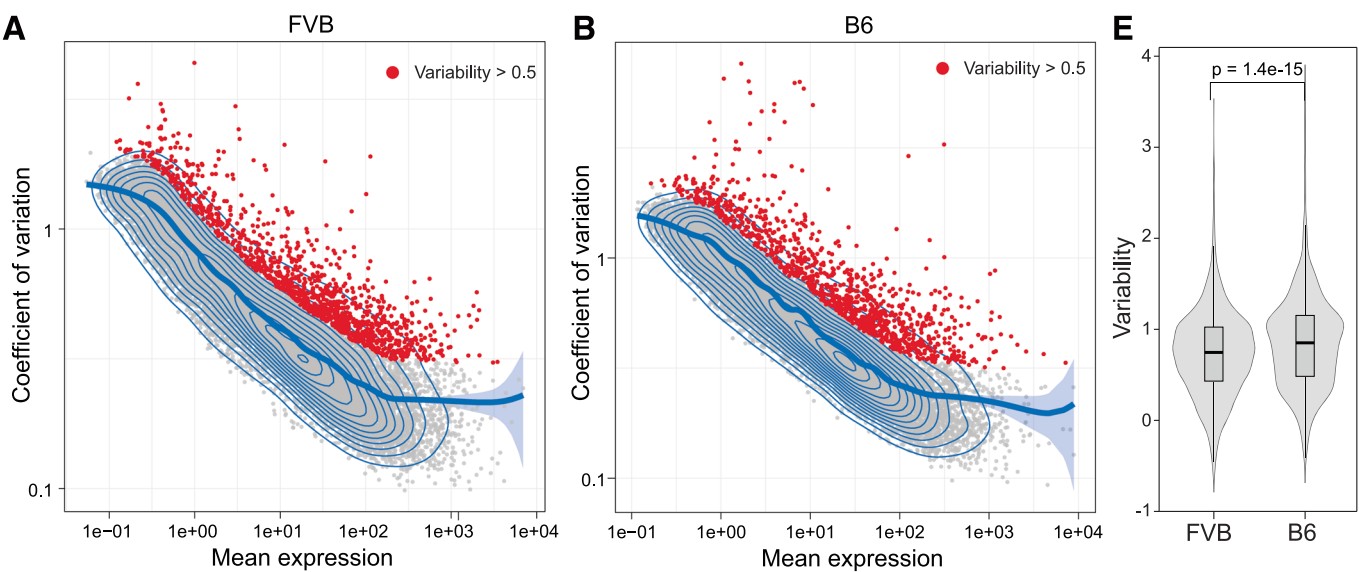

Expression noise in F1 across paternal environments (B6)

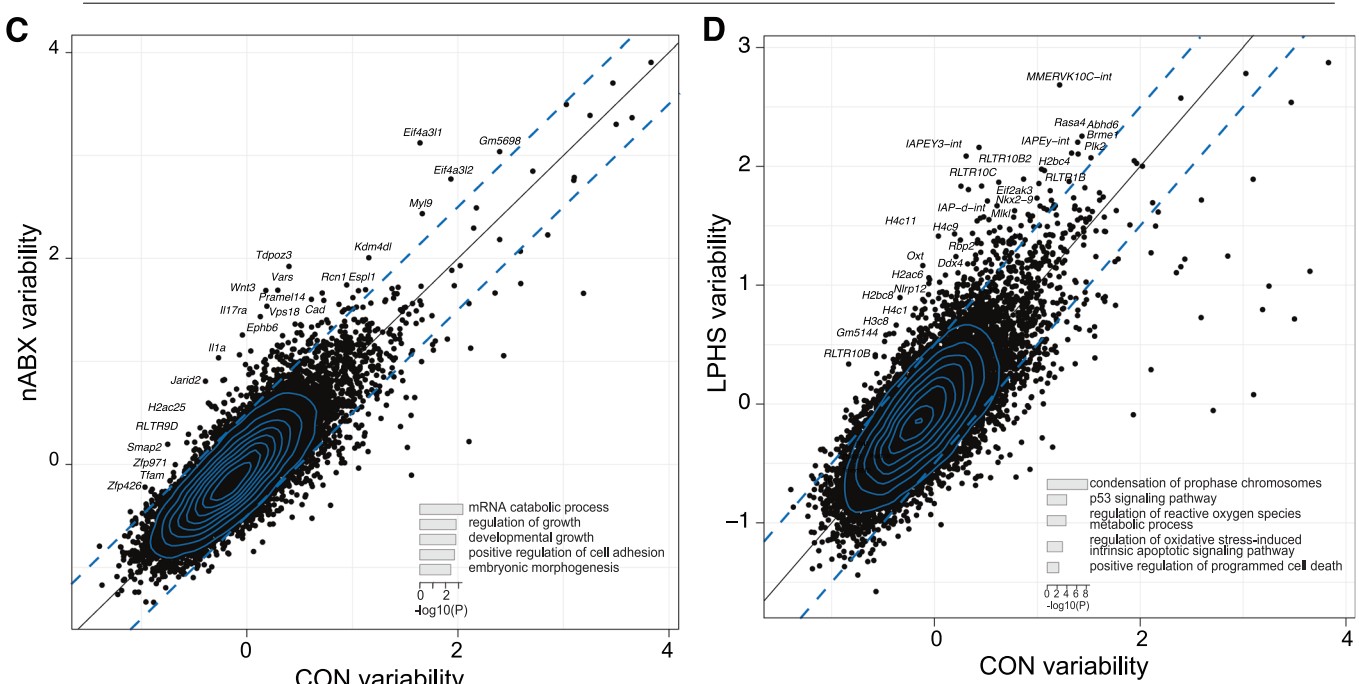

**Figure EV5. Gene expression variability across FVB and B6 backgrounds.**

(A, B) Scatter plots plotting coefficient of variation (CV) versus mean expression in FVB (A) and B6 (B) blastocysts. Blue line represents a generalised additive model (GAM) fit to the CV vs mean expression distribution. Red dots indicated genes with a variability score > 0.5 (see "Methods"). (C, D) Scatter plots contrasting the variability of genes in the nABX-derived blastocysts (C) and LPHS-derived blastocysts (D) relative to controls in the B6 background. Genes showing higher variability in blastocysts of treated males are highlighted. Horizontal bar chart highlighting enriched Gene Ontology (GO) terms in genes showing high variability in nABX-derived blastocysts (left) and LPHS-derived blastocysts (right) relative to controls in the B6 background. (E) Violin and box plot demonstrating the relative distribution of variability between the FVB and B6 backgrounds. P-value was computed using paired two-tailed t test. [#genes = 3738]. [Box plots: horizontal black line at the centre is the median. The box extends from the first quartile (Q1: 25th percentile) to the third quartile (Q3: 75th percentile). The vertical black line extends from $Q1 - 1.5 \times IQR$ to $Q3 + 1.5 \times IQR$, where IQR = interquartile range. The minima and maxima are not explicitly annotated in these figures.]

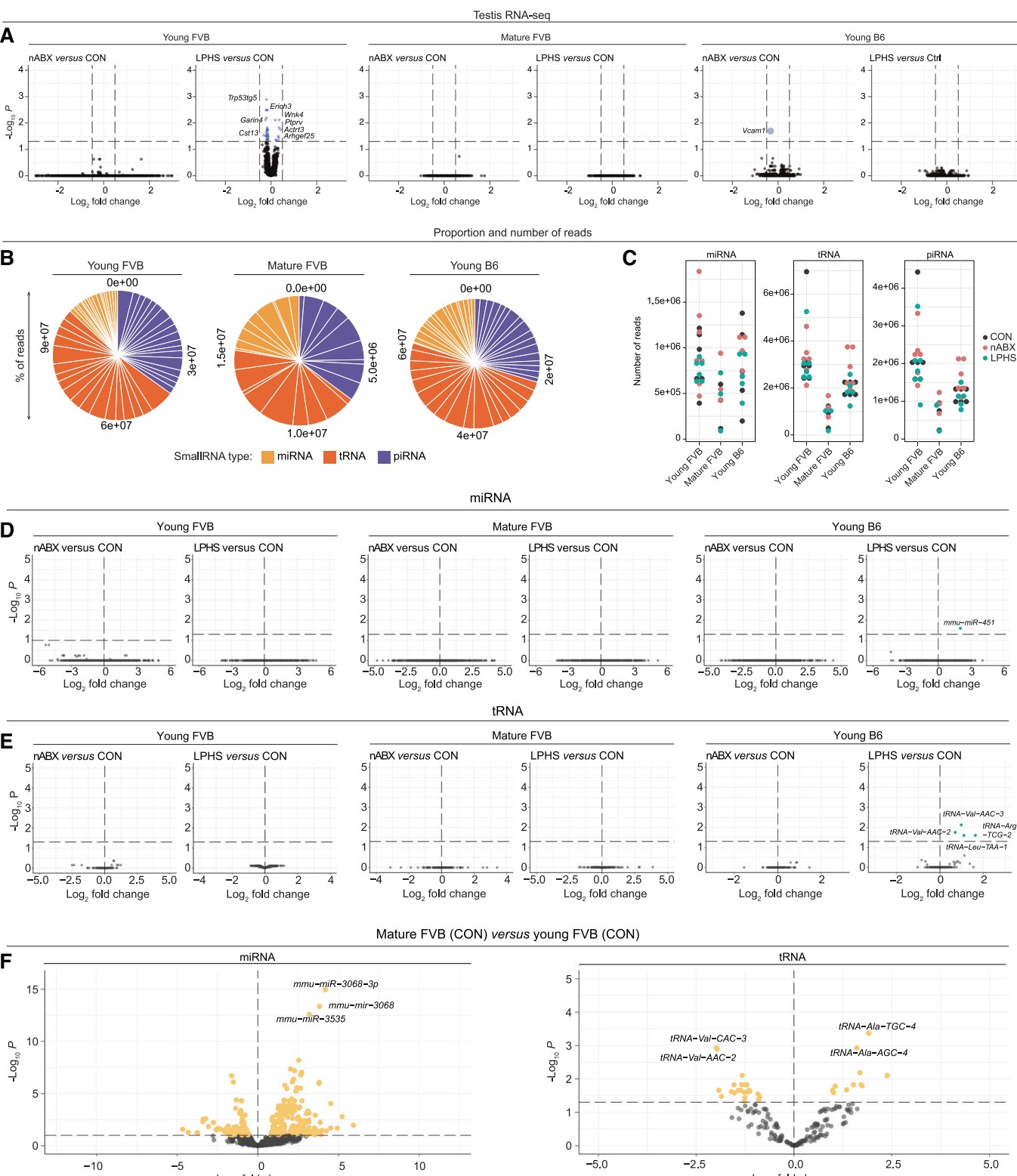

◀ **Figure EV6.  Paternal reproductive response to nABX, LPHS and aging across genetic backgrounds.**

(A) Volcano plots depicting differential gene expression in testis transcriptomes upon nABX and LPHS treatments in young FVB (left), mature FVB (middle) and young B6 (right) genetic backgrounds. [#genes: YoungFVB=21231, matureFVB=16586, youngB6 = 16529]. (B) Pie charts showing relative distributions of different types of small RNAs in sperm of young FVB (left), mature FVB (middle) and young B6 (right) males. (C) Dot plots showing depth of small RNA sequencing split into miRNA (left), tRNA (middle) and piRNA (right) reads split across genetic background and coloured by condition. (D) Volcano plots of differential expression of miRNAs in sperm of nABX (left) and LPHS (right) males relative to controls in young FVB, mature FVB and young B6 genetic background. [#miRNAs: YoungFVB=1271, matureFVB=1258, youngB6 = 1256]. (E) Volcano plots of differential expression of tRNAs in sperm of nABX (left) and LPHS (right) males relative to controls in young FVB, mature FVB and young B6 genetic background. [#tRNAs: YoungFVB=207, matureFVB=224, youngB6 = 198]. (F) Volcano plots of differential expression of miRNAs (left) and tRNAs (right) in sperm of untreated (CON) mature FVB versus young FVB males. [#miRNAs=1148, #tRNAs=264]. (A, D–F) All volcano plots plot adjusted P values versus $\log_2$ fold-change values. Thresholds for significance are set at adjusted P value < 0.05.

