## [Peer Review File · The EMBO Journal]

Embryonic signatures of intergenerational epigenetic inheritance across paternal environments and genetic backgrounds

Mathilde Dura, Bobby Ranjan, Joana Serrano, Rossella Paribeni, Violetta Paribeni, Laura Villacorta, Vladimir Benes, Olga Boruc, Ana Boskovic, and Jamie Hackett

Corresponding author: Jamie Hackett (jamie.hackett@embl.it)

Review Timeline:

Submission Date:	22nd Nov 24
Editorial Decision:	27th Jan 25
Revision Received:	23rd Apr 25
Editorial Decision:	12th Jun 25
Revision Received:	24th Jun 25
Accepted:	22nd Jul 25

Editor: Ieva Gailite

Transaction Report:

Dear Jamie,

Thank you for submitting your manuscript for consideration by the EMBO Journal. We have now received comments from two reviewers, which are included below for your information. Since the third reviewer was not able to return their comments in a timely manner due to health issues, I am taking the decision on the basis of the input at hand.

As you will see, both reviewers are generally positive in their assessment and appreciate the contribution of the study to the research field. At the same time, they indicate a number of concerns that would be important to address in the revised study. From my side, especially the concerns regarding the potential differences in developmental timing (reviewer #1) and the used statistical analysis (reviewer #2) appear particularly important to address. On the other hand, point 4 by reviewer #1 does not have to be addressed in full. Point 1 by reviewer #2 is also an important one, please consider including a comparison to B6 background in a few more key figures if feasible. Finally, both reviewers would appreciate an immunofluorescence staining-based confirmation for the reported alterations in blastocyst lineage allocation, although reviewer #1 classifies this request as non-essential. From the editorial side, I find that such data would strengthen the conclusions.

Based on these generally positive assessments, I invite you to address these comments in a revised manuscript. I think that it would be useful to discuss the revision in more detail via email or phone/videoconferencing - please let me know which option you prefer.

We generally allow three months as standard revision time, which can be extended to six months in the case of major revisions. Should you foresee a problem in meeting this deadline, please let us know in advance to discuss an extension. As a matter of policy, competing manuscripts published during this period will not negatively impact on our assessment of the conceptual advance presented by your study. However, please contact me as soon as possible upon publication of any related work to discuss the appropriate course of action.

When preparing your letter of response to the referees' comments, please bear in mind that this will form part of the Review Process File and will therefore be available online to the community. For more details on our Transparent Editorial Process, please visit our website: <https://www.embopress.org/page/journal/14602075/authorguide#transparentprocess>. Please also see the attached instructions for further guidelines on preparation of the revised manuscript.

Please feel free to contact me if have any further questions regarding the revision. Thank you for the opportunity to consider your work for publication, and I look forward to discussing your revision with you.

With best wishes,

Ieva

At EMBO Press we ask authors to provide source data for the main manuscript figures. Our source data coordinator will contact you to discuss which figure panels we would need source data for and will also provide you with helpful tips on how to upload

and organize the files.

We realize that it is difficult to revise to a specific deadline. In the interest of protecting the conceptual advance provided by the work, we recommend a revision within 3 months (27th Apr 2025). Please discuss the revision progress ahead of this time with the editor if you require more time to complete the revisions.

Referee #1:

In this manuscript, Dura and colleagues investigated the effects of paternal antibiotic exposure (a cocktail of Neomycin, Bacitracin, and Pimaricin) and nutritional alteration (low protein, high sugar diet) on the transcriptome of F1 blastocysts. The authors identified subtle yet significant differential gene expression (DGEs) in all experimental treatments compared to the control group in the FVB strain. Intriguingly, the DGEs associated with each experimental condition were linked to distinct Gene Ontology (GO) terms, enriched for different transcription factor binding reported for mouse ESCs, exhibited unique chromatin states as reported for wild-type embryos, and showed enrichment for different lineage-specific signatures. These findings suggest that various paternal exposures may impact distinct molecular pathways in F1 embryos. Interestingly, these effects were less pronounced in the B6 strain, potentially highlighting the complexity of genotype-environment (GxE) interactions. Furthermore, the authors observed differences in gene expression variability across strains and treatments.

The reviewer commends the study's robust experimental design (high replication, large sample size, and the use of two genetic backgrounds) and the rigorous bioinformatic analyses (integrative approaches and batch-specific considerations). This manuscript represents a significant contribution to the field of intergenerational epigenetic inheritance (IEI), setting high standards for experimental and analytical practices. It also emphasizes key technical, biological, and analytical considerations for such studies, while introducing gene expression variability as a potential mechanistic explanation for IEI phenomena. However, variations in the developmental timing of embryos - an important aspect which can introduce prominent variability - has not been taken into account in this study.

Major Comments

1) A major concern for this work is the developmental time of the blastocysts. While authors collected the blastocysts for all experiments at fixed time point of 92 (+/- 2) hpf, it cannot be excluded that certain blastocysts are developmentally more or less advanced than others at this time point. Such developmental differences may be driven by biological interventions as the authors noted a size decrease of LPHS group blastocyst, which could argue to developmental delays, but also from stochastic effects. Ideally, the collection of the blastocysts should have been performed at multiple different time points, so the authors could have constructed a developmental pseudotime, which then could have been taken into account in the differential gene expression models. Nevertheless, the authors might be still able to compute the developmental pseudotimes of the individual embryos taking advantage of publicly available RNAseq data collected at various time points. Taking developmental timing effects into consideration should be prominently discussed as part of their recommendations in the discussion.

2) In the blastocyst flow analysis (Supplementary Figure 1G), two distinct groups of blastocysts are visible (TOF ~400 and TOF <200). Additionally, some outliers with high TOF. Do these groups represent developmental differences? In other words, authors should investigate how changes in blastocyst flow characteristic analysis relate to the number of cells of the blastocysts. And more generally, is there also any relation to the number of cells in the trophectoderm and inner cell mass, something that could

relate to the concluded skewed differentiation and biased lineages? Were the blastocysts analyzed for transcriptomic profiling derived from one or both populations?

3) How do the authors explain the observation that LPHS treatment results in smaller blastocysts ("we noted a small but significant decrease in the size of blastocysts derived from LPHS-exposed males, pointing to a potential developmental delay") without corresponding differences in lineage marker expression (Figure 2D-2H), while nABX treatment shows significant misregulation of lineage markers without affecting blastocyst size?

Given the temporal separation of the inner cell mass and the trophectoderm prior to the separation of the epiblast and primitive ectoderm, how does the comparison of ICM vs TE marker genes look like? Can one detect the proposed delay of LPHS samples compared to controls by examining this earlier lineage decision?

4) The experiment involving aged versus young B6 animals (Figures 4E, F and 6E, F) appears tangential to the central question of this work. While aging is indeed a critical component of GxE interactions and the observed effects are intriguing, the inclusion of this experiment raises additional questions:

a) Could age-related effects be recapitulated in the FVB strain or other genetic backgrounds?

b) Would intermediate or older ages produce qualitative different DGE profiles, or do DGEs change quantitative with age progression?

c) This experiment also shows a reduced developmental efficiency of blastocyst in both young and old animals which is not explained by the authors.

Minor Comments

1) Figure 1G-H: it is not clear from the material and methods description whether the batch was taken into account when constructing the model for differential gene expression. Please clarify.

2) Figure 1G-H: Please show fold change expression as a function of absolute expression (MA plot).

3) Related to text describing gene function enrichments analysis: please provide actual data for functional enrichment scores including which genes are part of a given gene set / GO term cluster.

4) Figure 2A: Please indicate which TF binding profiles are enriched for UP and DOWN regulated genes separately. Which genes have the corresponding binding sites?

5) Figure 2B: What is the relevance of Trim28 motif and H3K9me3 enrichments at promoters of LPHS DEGs given that most of such genes are down regulated in the LPHS condition?

6) Figure 3: The authors examine the batch effects. Have the authors performed DGE analysis of controls only among the different batches? What is the output of such analysis in terms of numbers of DEGs and associated GO terms. If any DEGs are found between control sample batches, could they be explained by differences in developmental time? How would such results translate to DGEs between controls and experimental conditions when considering developmental timing as part of the differential expression model?

7) The experimental scheme in Figure 3A initially appeared to depict a new experiment. However, the PCA plots in Figures 1F and 3B suggest this is the same experiment analyzed differently. This caused some confusion while reading the manuscript. The authors should consider revising the schemes in Figures 1A and/or 3A for clarity.

8) Supplementary Figure 4A: The description in the manuscript says: "with negligible impact on overt phenotypes and bodyweights but opposite and significant differences in caecum weight (Fig S4A-C)". Why do the authors argue that there are no differences in bodyweight, while the bodyweights of growing nABX and PHLs treated males are smaller? Actually, why are the nABX males smaller at t=0?

9) Figure 4B: The LPHS group DEG response seen in FVB (mostly negative log₂FC) seems to be inverted in LPHS group of B6. Even though the FVB-LPHS DE genes do not significantly change in B6 they seem to have mostly positive log₂FC. How do the authors explain this?

10) Figure 5C-5D: What about the genes that show higher variability in the control?

11) Were there batch effects in terms of size or other flow parameters of the analyzed blastocysts? If so, how were these accounted for?

12) Can the authors provide a batch-specific analysis of the data presented in Figure 4A to ensure robustness?

13) How does gene expression variability change in the transcriptomes of testis and sperm in B6 and FVB mice across different treatments? This could offer additional insight into strain-specific differences.

14) Figure 4 and S4: The comparison of FVB vs B6 was presumably not done contemporarily. The authors should further discuss the absence of DEGs in the B6 embryos also in terms of timing of harvesting of blastocysts, developmental time differences of strains, blastocyst cell number differences among strains. Have the B6 embryos been harvested slightly earlier than the FVB embryos, explaining no DEGs in the B6 embryos since they may relate to the initiation of Epi/PE programs?

15) Figure 4B: Similar to the first minor comment: It is not clear from the material and methods description whether the batch was taken into account when constructing the model for differential gene expression. Please clarify.

16) Figure 5E: The authors mention in the manuscript: "Gene expression variability from both genetic backgrounds revealed that B6 blastocysts derived from control males are significantly noisier than their control FVB counterparts (Fig 5E and S5E). This implies that B6 genetics is inherently linked to more variation in expression landscapes during early development." To what extent is this influenced by proliferation and differentiation rates, or more broadly speaking by developmental timing? Have the authors counted the number of cells/nuclei in blastocysts, to assess developmental timing?

17) Finally, the authors use quite strong language in the manuscript to summarize their results of the experiments. For example: "Overall, our results indicate we can capture a robust molecular signature of paternal conditioning in early embryos." The

"robust" signature is influenced greatly by the batch and the genetic background. The reviewer recommends that authors should be more careful with their word choices.

Additional Non-Essential Suggestions

Since RNA-seq was performed on whole blastocysts, it would be interesting to assess lineage specification at the single-cell level using immunofluorescence staining. Such an approach could provide further insight into whether lineage specification is altered in response to paternal exposures. Alternatively, single blastomere RNA sequencing could be employed to validate lineage bias. However, both of these experiments require significant investment and are not necessary for the current work.

Referee #2:

Summary

The study by Dura et al examines the effects of paternal environments, genetics and age on early embryogenesis in mice. It presents comprehensive analyses of the transcriptome of >800 mouse blastocysts from male mice exposed to gut dysbiosis induced by antibiotics (nABX) or a low-protein high-sugar (LPHS) diet, and from males of two different ages or from two different genetic backgrounds. The purpose was to determine if overlapping molecular processes are associated with father-to-offspring inheritance of phenotypes induced by the selected exposure or condition. The results show distinct transcriptional signatures in blastocysts in these various conditions. Although significant batch effects in the RNA-seq data are observed, the results appear mostly reproducible across several experiments. The authors propose a variance-based analysis to identify hypervariable genes that may drive probabilistic F1 phenotypes. They then examine long and small RNA-seq data from the testes of fathers using the same methods and find that genetic background and age have strong effects on their transcriptome, which can potentially confound the effects of environmental exposures. Overall, the results are solid, the figures are clear and have helpful illustrations and annotations and the findings are valuable for the field of intergenerational epigenetic. The manuscript however remains incomplete in some parts and present statistical analyses that can be improved.

Major concerns:

1. The observed iGxE interactions in blastocysts suggest that the genetic background of the father influences the offspring' phenotypes. Based on the cited references, offspring phenotypes in the used conditions have been reported in B6 but not in FVB animals. This study does not conduct the analyses systematically in both genotypes, which is a limitation. Its relevance and usefulness would be higher if phenotypes would be examined in offspring of both genetic backgrounds.
2. The authors report a shift in (extra-)embryonic lineage allocation in blastocysts after paternal nABX exposure. It would be useful to examine if the results in Fig 2D-F remain valid if the analyses are performed individually for the three experimental batches mentioned in Fig 3. It would also increase the validity of this observation if the results could be confirmed with another method such as microscopy with lineage marker staining.
3. The authors present an unusual variance-based RNA-seq analysis that is used to explain the probabilistic nature of the offspring phenotypes, as they did in previous studies. Although this provides an interesting interpretation of the results, the absence of observable phenotypes in blastocysts questions the biological relevance of such method. It would be necessary to validate and benchmark the variance-based analysis for instance using RNA-seq data from animals with naturally variable phenotypes such as *Nnat*^{+/-p} (<https://doi.org/10.1038/s42255-022-00629-2>)
4. GxE interactions in testis and embryo are based on independent analyses in each genetic background followed by a comparison of results e.g. overlapping and non-overlapping genes and pathways. Although useful, this approach does not properly assess statistically the interaction between paternal exposure and genetic background. For a valid and quantitative evaluation of interaction effects, a unified statistical model should be used and the data from both genetic backgrounds together should be analysed with a model that includes a GxE interaction term e.g. Gene expression ~ Exposure + GeneticBackground + Exposure:GeneticBackground.
5. The experimental design involves analyzing embryos derived from the sperm of fathers with different exposures, creating a hierarchical structure with individual embryos nested within paternal groups. Embryos from the same father are not independent thus such nested structure needs to be accounted for to avoid an underestimation of variability, false positives and an inflated data significance. The method used to take this issue into account for differential expression analyses needs to be provided.

Minor concerns

6. p5: The authors state that "The direct physiological response of males was similar between the FVB and B6 backgrounds, with negligible impact on overt phenotypes and bodyweights, but opposite and significant differences in cecum weight"? However, a comparison of Fig. S1A and S4A suggests a strain-specific difference: FVB males have no significant change in body weight but B6 males have significant weight reduction (~10% mean difference) in response to both exposures. Additionally, data supporting the claim of opposite differences in cecum weight between FVB and B6 cannot be found in the manuscript. Comparing Fig. 1C and S4C, the effects of exposure appear to be consistent across strains. This needs to be clarified and the discrepancies explained.
7. p6: The authors state that "PCA on single blastocyst transcriptome revealed a distinction between embryos sired by mature or young males along the second principal component". However, the visual differences in the PCA plot are minimal, with no clear separation or clustering based on paternal age. The use of the term "distinction" seems overstated. Proper statistical analyses need to be conducted to quantify the effect of paternal age on PC2.
8. p17: The methods section "Profiling binding of transcription factors in our DE gene sets" lacks details. The description and

available code do not clearly explain how the analysis was conducted. While Enrichr performs enrichment analyses across multiple databases, which databases were used is not indicated. The methodology should also be clarified to ensure reproducibility.

9. p3: The sentence "we noted a significant decrease in the size of blastocysts derived from LPHS-exposed males, pointing at a potential developmental delay" needs a reference to support this observation.

10. For Fig. 11, labeling the clusters directly in the figure would improve clarity and make it easier to follow the corresponding explanation in the text.

11. p7: The authors refer to sequencing "sperm-borne small RNA." Given that small RNAs traffic from the epididymis to sperm cells, it is unclear how it was determined that the observed small RNAs are sperm-borne. This needs to be clarified and evidence to support the claim needs to be provided.

12. The description of IVF mentions an "oocyte pool." What such pool means needs to be clarified as this suggests that all oocytes from different mothers are mixed at the start of the IVF procedure. Then whether embryos can be traced back to the oocytes of individual mothers or not needs to be indicated.

Response to Reviewers

Dura, Ranjan et al.,

We thank the Editor and the Reviewers for taking the time to provide balanced and thoughtful feedback. We believe we have addressed all the points raised by the reviewers, through generating new *in vivo* experimental data and *in silico* analyses. Below we summarise the major experimental additions, and then address each reviewer comment (C) with a full response (R).

Summary of major additions:

- 1) We generated a major new single-embryo transcriptomics dataset of >200 blastocysts sired by young or mature FVB males, further revealing that aging represents an 'environmental' factor that influences F1 phenotype. This new data enabled comprehensive investigation into intergenerational GxE effects (B6vFVB background), which uncovered how genetics and environment interact. The new data form the bulk of an almost entirely new figure in the revised manuscript (**Fig 4**) and considerably strengthen the insights.
- 2) We have performed *in silico* developmental staging of embryos analyzed in our study. This validates that the hundreds of sequenced embryos exhibit synchronous transcriptomic profiles, and provides evidence that paternal preconception conditioning does not significantly impact developmental pace. This result supports the conclusion that lineage changes and other measured effects are unlikely to be indirect readouts of developmental asynchrony.
- 3) We performed immunofluorescence and cell count experiments on new and independent F1 cohorts from nABX and LPHS sires, which support the results of developmental staging. We have further performed staining for lineage markers which supports altered expression of lineage regulators.
- 4) We have validated and benchmarked our variance-based analysis using published datasets of samples with naturally variable phenotypes, strengthening our conclusions about gene-expression variance as an important new biological metric in intergenerational epigenetic studies. This approach is now freely available to the community as a fully documented R package on GitLab for users to download and apply to their own datasets.
- 5) As per the Reviewers' constructive comments, we made substantial changes to the text, adding to the clarity and precision of our conclusions. Additional figures for Reviewers from requested analyses have been added to this document.

In sum, we have added considerable experimental data that improves the manuscript and conclusions. We now profile embryonic signatures of intergenerational epigenetic inheritance across three paternal environments at two ages in two genetic backgrounds, serving as a foundational resource for the field. Additionally, this work provides analytical tools and a codebase for systematic reproduction and analysis of transcriptomics datasets.

Referee #1

C1: *In this manuscript, Dura and colleagues investigated the effects of paternal antibiotic exposure (a cocktail of Neomycin, Bacitracin, and Pimaricin) and nutritional alteration (low protein, high sugar diet) on the transcriptome of F1 blastocysts. The authors identified subtle yet significant differential gene expression (DGEs) in all experimental treatments compared to the control group in the FVB strain. Intriguingly, the DGEs associated with each experimental condition were linked to distinct Gene Ontology (GO) terms, enriched for different transcription factor binding reported for mouse ESCs, exhibited unique chromatin states as reported for wild-type embryos, and showed enrichment for different lineage-specific signatures. These findings suggest that various paternal exposures may impact distinct molecular pathways in F1 embryos. Interestingly, these effects were less pronounced in the B6 strain, potentially highlighting the complexity of genotype-environment (GxE) interactions. Furthermore, the authors observed differences in gene expression variability across strains and treatments. The reviewer commends the study's robust experimental design (high replication, large sample size, and the use of two genetic backgrounds) and the rigorous bioinformatic analyses (integrative approaches and batch-specific considerations). This manuscript represents a significant contribution to the field of intergenerational epigenetic inheritance (IEI), setting high standards for experimental and analytical practices. It also emphasizes key technical, biological, and analytical considerations for such studies, while introducing gene expression variability as a potential mechanistic explanation for IEI phenomena. However, variations in the developmental timing of embryos - an important aspect which can introduce prominent variability - has not been taken into account in this study.*

R1: We appreciate the reviewer dedicating their time and effort to assess our study, and concluding that “*this manuscript represents a significant contribution to the field of intergenerational epigenetic inheritance (IEI), setting high standards for experimental and analytical practices*”.

C2: *A major concern for this work is the developmental time of the blastocysts. While authors collected the blastocysts for all experiments at fixed time point of 92 (+/- 2) hpf, it cannot be excluded that certain blastocysts are developmentally more or less advanced than others at this time point. Such developmental differences may be driven by biological interventions as the authors noted a size decrease of LPHS group blastocyst, which could argue to developmental delays, but also from stochastic effects. Ideally, the collection of the blastocysts should have been performed at multiple different time points, so the authors could have constructed a developmental pseudotime, which then could have been taken into account in the differential gene expression models. Nevertheless, the authors might be still able to compute the developmental pseudotimes of the individual embryos taking advantage of publicly available RNAseq data collected at various time points. Taking developmental timing effects into consideration should be prominently discussed as part of their recommendations in the discussion.*

R2: The reviewer is correct in stating that certain blastocysts - irrespective of the conditions - could be more or less developmentally advanced from others, as cleavages of mouse preimplantation embryos are inherently not synchronous. Indeed, we designed the study to minimise this by performing IVF with precisely chronologically staged blastocysts (92 hrs post-IVF). Moreover, the large replicate number of individual blastocysts - fertilised and cultured in parallel between fathers - is designed to mitigate against stochastic differences in developmental rate. Nonetheless, to further determine if paternal preconception exposure to nABX or LPHS causes changes in developmental

timing of the resulting blastocysts, we have followed the reviewer's suggestion and quantified the developmental pseudotime of our blastocysts by leveraging a publicly available dataset (Deng et al, 2014), comprising early, mid and late-blastocyst samples. Our new analysis, included in the revised MS in the **Figure S2E**, and below (**Figure R1**) shows that PC1 captures developmental progression between early and late blastocysts, while early and mid-blastocysts separate along the PC2 (**Figure R1**). Projecting our transcriptomic data onto this plot clearly shows that our embryos cluster in the middle of the first PC, appropriately corresponding to embryonic day E4 (i.e. 92 h post-IVF). Importantly, we do not detect additional separation within our dataset, indicating that paternal preconception state does not significantly alter developmental timing of the embryos. This result is also consistent with **R4** (below), where we determined the number of cells per blastocyst across the three experimental groups (using FVB mice) and did not observe significant changes in cell numbers for any condition analyzed. Thus, changes in gene expression likely do not stem from indirect effects of altered developmental staging between conditions.

Figure R1. Principal component projection for *in silico* developmental staging of our single-embryo transcriptomes of young FVB background coloured by paternal condition using published data from Deng et al. 2014.

C3: *In the blastocyst flow analysis (Supplementary Figure 1G), two distinct groups of blastocysts are visible (TOF ~400 and TOF <200). Additionally, some outliers with high TOF. Do these groups represent developmental differences? In other words, authors should investigate how changes in blastocyst flow characteristic analysis relate to the number of cells of the blastocysts. And more generally, is there also any relation to the number of cells in the trophectoderm and inner cell mass, something that could relate to the concluded skewed differentiation and biased lineages? Were the blastocysts analyzed for transcriptomic profiling derived from one or both populations?*

R3: The reviewer correctly observes that the flow analysis of blastocysts in Figure S1G shows two distinct groups characterized by different TOFs. We therefore performed a detailed re-analysis of the raw imaging data of the embryos used in this experiment. Despite our best efforts, we could not determine the source of the measured differences in the time of flight: none of the characteristics which we could assess (batch, size, morphological features, etc) could explain the measured TOF differences. As we know the identity of each blastocyst in this experiment, in the revised manuscript we now include the representative images of corresponding embryos in either TOF ~400 or TOF <200 groups taken directly from the flow analysis (Fig. S1G), also shown below, for the reviewer's

convenience. Regarding the potential differences in cell numbers and lineage bias between the inner cell mass (ICM) and trophectoderm (TE) allocation, we kindly refer the review to the **R4**, below. In brief, we do not find evidence for altered cell numbers. Finally, we see no evidence of bimodality in the transcriptomic data that would explain TOF. Taken together, we believe TOF is capturing a metric unrelated to relevant or overt phenotypes - possibly a technical feature of flow analysing blastocysts. Crucially however, it is not differential between paternal groups.

Figure R2. Scatter plot showing the time of flight (TOF) and Extinction of each blastocyst embryo colour by condition and by batch of experiments, and corresponding representative images of analysed blastocysts. Violin plot showing quantification of TOF or extinction.

C4: How do the authors explain the observation that LPHS treatment results in smaller blastocysts ("we noted a small but significant decrease in the size of blastocysts derived from LPHS-exposed males, pointing to a potential developmental delay") without corresponding differences in lineage marker expression (Figure 2D-2H), while nABX treatment shows significant misregulation of lineage markers without affecting blastocyst size? Given the temporal separation of the inner cell mass and the trophectoderm prior to the separation of the epiblast and primitive ectoderm, how does the comparison of ICM vs TE marker genes look like? Can one detect the proposed delay of LPHS samples compared to controls by examining this earlier lineage decision?

R4: There could be several potential explanations behind the observed smaller size of LPHS-derived blastocysts, despite no observed lineage gene expression changes. For instance, slower cell cycle progression could lead to a slight developmental delay without an impact on lineage allocation, as reported in PMID: 34644528. Furthermore, the size of the blastocoel directly impacts blastocyst size, as its functional role in the mouse embryo is to induce expansion of the embryo so that it can hatch out of the zona pellucida and implant at the appropriate time (PMID: 2540052, PMID: 29920273). Similarly, small differences in cell volume (in any or all of the lineages) could contribute to the overall measured size of a blastocyst composed of ~100 cells. To investigate this further, we have performed new experiments using multiple independent blastocyst cohorts to determine cell number and lineage allocation across batches. Our cell count analyses in Control versus LPHS-derived blastocyst did not detect significant changes in the total cell numbers between two groups of embryos

(**Fig R3**). This suggests the smaller size is not a developmental delay but a growth defect. Note that the growth quantification occurred in multiple batches each with sufficient replicates.

Figure R3. Quantitation of lineage staining on single blastocysts (shown fully in Fig S2) revealing no change in cell number per embryo but a difference in lineage allocation. Each data point is an individual blastocyst taken from two independent experiment replicates.

The same analysis, this time for Control versus nABX-derived embryos, also revealed no significant impact of paternal preconception nABX exposure on the total cell number in the resulting embryos. However, consistently with transcriptomic results, we observe an enrichment of CDX2-positive cells in nABX-derived embryos, suggesting that paternal dysbiosis can manifest its effect onto the next generation through modulation of cell fate decisions during early development (**Figure S2C**). Following the reviewer’s suggestion, we have also generated contour plots using ICM and TE markers for nABX vs Control and LPHS vs Control comparisons (**Figure R4**). While in the LPHS case we do not observe any lineage skew, paternal nABX exposure led to significantly increased expression of epiblast markers, as shown in **Fig. 2**.

Figure R4. Contour plots and violin plots of correlation coefficients of blastocyst transcriptomic profiles from indicated sperm donor condition with the reference single-cell lineage of epiblast and trophoblast.

C5: *The experiment involving aged versus young B6 animals (Figures 4E, F and 6E, F) appears tangential to the central question of this work. While aging is indeed a critical component of GxE interactions and the observed effects are intriguing, the inclusion of this experiment raises additional questions:*

- a) Could age-related effects be recapitulated in the FVB strain or other genetic backgrounds?*
- b) Would intermediate or older ages produce qualitative different DGE profiles, or do DGEs change quantitative with age progression?*
- c) This experiment also shows a reduced developmental efficiency of blastocyst in both young and old animals which is not explained by the authors.*

R5: We answer here the tripartite question from the reviewer:

i) As suggested by the reviewer, we have now performed a major new experiment to profile F1 blastocysts from young and mature (1 year old) males in the FVB background, which are included in the revised Manuscript (*see new Figure 4*). Similarly to what we observed for B6 genetic background, paternal age represents an important ‘environmental’ factor influencing early embryonic gene expression in FVB mice. We detect a clear separation of ‘old’ versus ‘young’ transcriptomic signatures along the PC1, as well as 306 DEGs (204 downregulated and 102 upregulated genes) in blastocysts from mature FVB sperm donors. The new data enable us to rigorously compare the effects of different paternal environments (age) across genetic backgrounds. Gene set enrichment analysis (GSEA) indicates a common signature irrespective of background, highlighting generic or conserved F1 effects. It further shows that that genetic background modifies the precise nature of transcriptomics effects, implying intergenerational GxE is a major player in F1 outcomes. For example, pathways for metabolic OxPhos are highly downregulated in blastocysts from aged fathers in both B6 and FVB, while effects on ribosomal pathways appear B6-specific.

ii) The question of whether age quantitatively or qualitatively impacts gene expression patterns in F1 blastocysts remains open. In this study, we opted to use animals of ages corresponding to human puberty/young adulthood and middle-age (45-60 years of age), both of which are relevant in terms of reproductive susceptibility to external stimuli. Although it would be interesting to extend our study to include additional timepoints in between, and even beyond 1-year, these experiments would require years of work to generate enough appropriately aged animals and resulting embryos, and go beyond the scope of the current study.

iii) The reviewer notes ‘reduced developmental efficiency’ of B6 embryos compared to FVB ones. Here, we are not able to distinguish between the impact of IVF success versus efficiency of downstream development on the overall rates of B6 blastocyst formation. Indeed, the efficiency of in vitro fertilization as well as ex-vivo culturing robustness of B6 gametes/embryos is reduced compared to the FVB strain. It is well-accepted that B6 is not the best model for early developmental studies, and most of the IVF and ex-vivo development protocols have been developed and optimized for FVB or CD1 mouse strains due to their favorable reproductive traits (large litters, good maternal instincts, and high implantation success: <https://link.springer.com/article/10.1023/A:1022166921766>, <https://www.informatics.jax.org/silver/chapters/6-2.shtml>). However, once fertilized and channelled through proper development, B6 embryos develop to the blastocyst stage at comparable speed as their FVB counterparts, arguing against reduced developmental efficiency in our analyzed samples. Importantly, at the time of blastocyst collection, visual inspection of the embryos ensured that only blastocysts with normal morphological appearance were harvested for downstream transcriptomic studies.

Minor Comments

1) **Figure 1G-H:** it is not clear from the material and methods description whether the batch was taken into account when constructing the model for differential gene expression. Please clarify.

In the revised manuscript, this has been clarified in the Methods Section III: Single-embryo transcriptomics analysis - Differential Expression analysis, which now states: ‘For all analyses, paternal condition (“Condition”) was the only term used in the design formula; see GitLab link (Code availability section) for full code and parameters’. We identified batch-robust differentially expressed genes (broDEGs) by performing differential analysis in each batch individually. Please also see R5 to reviewer 2 for further information on this.

2) **Figure 1G-H:** Please show fold change expression as a function of absolute expression (MA plot).

Please find the requested MA plots below (**Figure R5**).

Figure R5. MA plots showing fold-change of differential expression as a function of mean expression: (left) nABX vs CON and (right) LPHS vs CON. DE genes are coloured in red ($\log_2FC > 0.5$ & adjusted p -value < 0.05).

3) **Related to text describing gene function enrichments analysis:** please provide actual data for functional enrichment scores including which genes are part of a given gene set / GO term cluster.

The data requested by the reviewer has now been added to **Supplementary Tables** in the revised Manuscript. Please find the table legends below:

Table S2 – String overrepresented terms in blastocyst DE genes by young FVB paternal environment

Table S3 – Gene set enrichment analysis of nABX-derived blastocysts in young FVB background

Table S5 – GO overrepresentation analysis in blastocyst DE gene clusters by young FVB paternal environment

Table S12 – Gene set enrichment analysis of mature vs young male-derived blastocysts across FVB and B6N backgrounds

Table S14 – Overrepresentation analysis of hypervariable genes across paternal environments in young FVB background

Table S16 – Overrepresentation analysis of hypervariable genes across paternal environments in young B6J background

Table S17 – Overrepresentation analysis of hypervariable genes specific to paternal treatment across genetic backgrounds

4) **Figure 2A:** Please indicate which TF binding profiles are enriched for UP and DOWN regulated genes separately. Which genes have the corresponding binding sites?

This data is now available in **Table S7**. For nABX DEG, we find the enrichment for binding of pluripotency (epiblast) factors such as OCT4 is driven by the upregulated genes. Downregulated genes are enriched in SMAD binding. This is in line with an increase in collective epiblast geneset expression and reduced primitive endoderm signature at the global level. For LPHS, because downregulation is the predominant effect, the binding profiles already reflect enrichment on downregulated genes.

5) **Figure 2B**: *What is the relevance of Trim28 motif and H3K9me3 enrichments at promoters of LPHS DEGs given that most of such genes are down regulated in the LPHS condition?*

The observation of enrichment of Trim28 motif and H3K9me3 over promoters of LPHS DEGs is indeed intriguing. Considering that these processes are linked, that the chromatin signatures are associated with gene repression, and that LPHS DEGs are indeed mostly downregulated, it is plausible to hypothesize that paternal exposure to LPHS could manifest its effects in the next generation through aberrant or incomplete heterochromatin formation and/or remodelling, naturally occurring during earliest stages of mouse development. Curiously, a critical function of Trim28-mediated H3K9me3 heterochromatin is the repression of transposable elements in the genome, including the MERVL retrotransposon, which is expressed exclusively during earliest stages of preimplantation development when embryonic blastomeres are considered to be totipotent. Our previous studies have shown that MERVL promoter (which is its Long Terminal Repeat (LTR)), and the cellular genes driven by MERVL-LTR (termed ‘MERVL target genes’) are responsive to paternal LPHS diet consumption (PMID: 26721685), and that heterochromatin-based mechanisms regulate MERVL expression amplitude in the early embryo (PMID: 31831626). Importantly, in this study we also identify MERVL transcripts as some of the most variable ones in response to paternal LPHS exposure. Therefore, it is plausible to hypothesize the presence of a functional mechanistic link between the impact of paternal dietary exposure on heterochromatin status across genomic transposable elements and their targets, which have potential to regulate early developmental events, including the onset of embryonic genome activation (EGA) or the duration of the totipotent state.

6) **Figure 3**: *The authors examine the batch effects. Have the authors performed DGE analysis of controls only among the different batches? What is the output of such analysis in terms of numbers of DEGs and associated GO terms. If any DEGs are found between control sample batches, could they be explained by differences in developmental time? How would such results translate to DGEs between controls and experimental conditions when considering developmental timing as part of the differential expression model?*

We have now performed DGE analysis of the controls among the different batches, and found that there is shared differential expression across all batches (**Figure R6**). However, *in silico* developmental staging shows no consistent batch-specific differences in developmental timing, implying that the batch-specific control DE genes are unlikely to be explained by differences in developmental time. Instead they likely represent technical variables relating to the inherent batch differences, sample handling, library preparations and other variations. For this reason, the identification of batch-robust DEG (broDEG) that are explained by paternal condition rather than batch, such as *Gata4*, is crucial and is an important consideration in intergenerational studies. In this

scenario genes that are consistently altered *within* batches, rather than between batches, are high-confidence.

Figure R6. (left) Venn diagram showing overlap of DE genes among control samples only across batches. (right) Heatmap of Gene Ontology terms enriched in the DE genes among control samples only across batches.

7) *The experimental scheme in Figure 3A initially appeared to depict a new experiment. However, the PCA plots in Figures 1F and 3B suggest this is the same experiment analyzed differently. This caused some confusion while reading the manuscript. The authors should consider revising the schemes in Figures 1A and/or 3A for clarity.*

We thank the reviewer for their suggestion. The schematic in **Figure 3A** is designed to depict the details behind experimental setup which allowed us to obtain statistical power for the downstream analyses and address the impact of batch-to-batch variability in the transcriptomic data, which is less commonly considered in intergenerational studies. Indeed, the PCA in **Figures 1F** and **3B** contain the same experimental samples, but while **Figure 1F** focuses on the impact of paternal exposure on blastocyst gene expression via conventional analysis, **Figure 3B** highlights the impact of the experimental batch. In the revised manuscript, we have now explicitly stated in the **Figure 3** legend that the PCA plot in **Figure 3B** is generated from the same experimental samples as those in **Figure 1F**: “(B) PCA on global transcriptomes from Fig 1F, coloured by batch.”

8) *Supplementary Figure 4A: The description in the manuscript says: "with negligible impact on overt phenotypes and body weights but opposite and significant differences in caecum weight (Fig S4A-C)". Why do the authors argue that there are no differences in bodyweight, while the bodyweights of growing nABX and LPHS treated males are smaller? Actually, why are the nABX males smaller at $t=0$?*

We thank the reviewer for pointing this out. When comparing the absolute values of bodyweight (in grams) between the three groups of males in the B6 group, the nABX and LPHS treated males show a decrease compared to Control over time. However, as the reviewer rightfully points out, at $t=0$, the weight of nABX-exposed males was, by random chance, lower compared to Control males. This represents experimental chance during the random separation of wildtype littermates into experimental groups. During the treatment period of 6 weeks, nABX males gain weight proportionally

to their starting values, and therefore the dynamics of weight-gain over time, expressed as percentage of starting weight, is comparable (actually slightly higher) than controls. This is in line with nABX-exposure in FVB and previous studies showing nABX leads to marginally elevated weight gain. In contrast, LPHS exposure in B6 leads to a slight weight loss as a dynamic over the time course. Given that this group randomly started at a similar (albeit slightly lower) weight than Control this likely represents a true effect. Indeed, it is also consistent with a marginal weight loss in FVB. Thus, B6 weight change trends are consistent with FVB, but due to the slightly different t=0 starting weights we cannot make robust conclusions on this. In line with the reviewers point, we have modified the text so that it does not claim “negligible impact on overt phenotypes and body weights”.

9) **Figure 4B:** The LPHS group DEG response seen in FVB (mostly negative log2FC) seems to be inverted in LPHS group of B6. Even though the FVB-LPHS DE genes do not significantly change in B6 they seem to have mostly positive log2FC. How do the authors explain this?

We too found it perplexing that the transcriptomic signature of LPHS in the young FVB background seems to be partially inverted in the B6J background. We have run multiple quality control steps, including comparing library sizes, read mapping rates and mitochondrial gene expression, and the results do not suggest a technical artifact. Moreover, we consistently see this effect over independent batches. We hypothesize that these pathways may be particularly sensitive to paternal effects, with the directionality influenced by genetic background. Alternatively, the increased variability in gene expression in the B6J background (**Figure 5**) could contribute to spurious non-significant fold changes when averaged over a population. Of note, the change in MERVL in response to LPHS is consistent across both genetic backgrounds. Moreover, in the new manuscript, we find that key F1 responses to aged fathers - such as OxPhos activity - is consistent across genetic backgrounds, while other responses are specific. Thus, while we cannot completely rule out technical issues - given the scale of replications (>1,000 blastocysts), this implies that the impact of each environment is tremendously sensitive to genetic background. This is a key conclusion of our study.

10) **Figure 5C-5D:** What about the genes that show higher variability in the control?

We have now included the list of genes which show higher variability in control conditions compared to experimental ones in Supplementary Table S15. Overrepresentation analysis shows that genes hypervariable in the controls against either experimental condition (nABX or LPHS) are enriched for chromosome condensation and cellular pH regulation (**Figure R7**).

Figure R7. Heatmap showing overrepresentation analysis of hypervariable genes in control samples against the two experimental conditions in the young FVB background.

11) Were there batch effects in terms of size or other flow parameters of the analyzed blastocysts? If so, how were these accounted for?

We kindly refer the reviewer to **R3**, above, where we provide a detailed answer regarding the absence of batch effects in our blastocyst FACS data, and provide raw images of the analyzed blastocysts. We restate once more that we do not observe batch effects in terms of size or other flow parameters. This is consistent with visual observations during experiments, and quantitative data showing equivalent rates of blastocyst development and hatching between independent batches (**Fig S1**).

12) Can the authors provide a batch-specific analysis of the data presented in **Figure 4A** to ensure robustness?

We provide batch-specific volcano plots of differential expression in the B6 genetic background (**Figure R8**). The batch-to-batch variability in F1 responses in B6 further emphasizes the importance of profiling multiple batches in IEI studies. Indeed, our overall conclusion is that there is an absence of *robust* F1 DEGs induced by nABX or LPHS in the B6 background. We further note that our batches were highly experimentally controlled with precise timing, high fertilisation rate, and parallel IVF from a common pool of oocytes for all sperm donors.

Figure R8. Volcano plots showing batch-specific analysis of single-embryo differential gene expression across paternal nABX (top row) and paternal LPHS (bottom row) in the young B6J genetic background. There were no batch-robust DE genes in the B6J background.

13) *How does gene expression variability change in the transcriptomes of testis and sperm in B6 and FVB mice across different treatments? This could offer additional insight into strain-specific differences.*

Although this is an interesting question, we have only performed RNA-seq and small RNA-seq on 6 replicates of testes and sperm samples each. Thus, our data is underpowered to determine the impact of different paternal environments on variability in gene expression in these tissues.

14) **Figure 4 and S4:** *The comparison of FVB vs B6 was presumably not done contemporarily. The authors should further discuss the absence of DEGs in the B6 embryos also in terms of timing of harvesting of blastocysts, developmental time differences of strains, blastocyst cell number differences among strains. Have the B6 embryos been harvested slightly earlier than the FVB embryos, explaining no DEGs in the B6 embryos since they may relate to the initiation of Epi/PE programs?*

We try to address the reviewer's point here: The comparison of FVB vs B6 transcriptomic data was done contemporarily, but the generation of experimental data used in the comparison took place on different days. For clarity, blastocysts derived from either mouse strain were collected after the same amount of chronological time post-IVF (92 +/- 2 hours), and the IVF conditions used were the same between strains. Visual inspection of the harvested embryos did not reveal any discernible morphological differences between FVB and B6 embryos, and we were rigorous in harvesting only blastocysts which exhibited normal morphology. Therefore, while we cannot formally exclude the possibility that the B6 blastocysts are slightly 'slower' or 'faster' compared to the FVB ones in terms of cleavage time - which would require continuous live-imaging of embryos during preimplantation - on all accounts the data from the two strains are directly comparable. Finally, differential expression of genes related to Epi/PrE lineages was detected in nABX-sired FVB blastocysts compared to Control conditions. Therefore, a putative developmental delay of B6 blastocysts could only explain the absence of Epi/PrE paternal signature, but would not be able to explain the absence of the LPHS DEG signature, which is also not detected in B6 embryos.

15) **Figure 4B:** *Similar to the first minor comment: It is not clear from the material and methods description whether the batch was taken into account when constructing the model for differential gene expression. Please clarify.*

We apologize for the lack of clarity in the text. Batch was not taken into account when constructing the model for differential gene expression, which we have now explicitly stated in the Methods Section III: Single-embryo transcriptomics analysis - Differential Expression analysis, indicating: 'For all analyses, paternal condition ("Condition") was the only term used in the design formula; see GitLab link (Code availability section) for full code and parameters'. We have, however, performed gene expression comparisons for all individual batches of embryos collected, and the data is presented in **Figure 3**.

16) **Figure 5E:** *The authors mention in the manuscript : "Gene expression variability from both genetic backgrounds revealed that B6 blastocysts derived from control males are significantly noisier than their control FVB counterparts (Fig 5E and S5E). This implies that B6 genetics is inherently linked to more variation in expression landscapes during early development." To what extent is this influenced by proliferation and differentiation rates, or more broadly speaking by developmental*

timing? Have the authors counted the number of cells/nuclei in blastocysts, to assess developmental timing?

In the FVB background, we have performed *in silico* developmental staging using published single-cell RNA-seq data (**Figure R1**), and immunofluorescence staining reveals no change in the number of nuclei in blastocysts across conditions (**Figure S2B-C,E**). The same *in silico* developmental staging analysis, this time performed for B6J-strain blastocysts revealed nearly-identical distribution of these samples along the PC1 as we observed for FVB blastocysts (in other words, transcriptomes clustering between early- and late-blastocysts stages, corresponding to E4), and is shown below in **Figure R9**. This analysis therefore suggests that the inherent variability we observe in the gene expression profiles of B6J blastocysts, in any of the three paternal preconception conditions, is not globally influenced by proliferation rates or developmental timing.

Figure R9. Principal component projection for *in silico* developmental staging of our single-embryo transcriptomes of young B6J background coloured by paternal condition using published data from Deng et al. 2014.

17) Finally, the authors use quite strong language in the manuscript to summarize their results of the experiments. For example: "Overall, our results indicate we can capture a robust molecular signature of paternal conditioning in early embryos." The "robust" signature is influenced greatly by the batch and the genetic background. The reviewer recommends that authors should be more careful with their word choices.

We thank the reviewer for their remarks - indeed, a central finding of our study is that drawing strong conclusions in IEI paradigms is challenging and subject to confounders, which we highlight in the abstract and throughout. In the revised manuscript, we have implemented text changes to more-appropriately reflect the magnitude of the measured transcriptional and physiological phenotypes we report, across genetic backgrounds, age and exposure conditions. We hope that these changes will render the manuscript more balanced.

Additional Non-Essential Suggestions

Since RNA-seq was performed on whole blastocysts, it would be interesting to assess lineage specification at the single-cell level using immunofluorescence staining. Such an approach could provide further insight into whether lineage specification is altered in response to paternal exposures. Alternative, single blastomere RNA sequencing could be employed to validate lineage bias. However, both of these experiments require significant investment and are not necessary for the current work.

We thank the reviewer for this suggestion. We have now performed immunostaining experiments in FVB blastocysts, where we assessed the lineage allocation bias based on paternal preconception exposure, by quantifying the number of CDX2-positive cells (CDX2 is a well-characterized marker of trophectoderm lineage) and total cell numbers per blastocyst for each of the conditions (CON, nABX and LPHS). The data, now included as the new **Figure S2B,C**, shows an increase in the number of CDX2-positive cells only in blastocysts sired by nABX-consuming males, and is consistent with the transcriptomic data (Figure 2). Note that during the course of revisions we expended considerable effort to stain embryos for GATA4 (a marker of the primitive endoderm lineage) which is also a batch-robust DEG. Unfortunately, despite efforts using two independent antibodies, the staining did not technically work in our hands - no specific signal was detected in any samples. Instead we were able to stain for SOX17, which was *not* recorded as a DEG. Consistently, we observed no significant change in SOX17 cells, albeit it trended downwards.

Referee #2

The study by Dura et al examines the effects of paternal environments, genetics and age on early embryogenesis in mice. It presents comprehensive analyses of the transcriptome of >800 mouse blastocysts from male mice exposed to gut dysbiosis induced by antibiotics (nABX) or a low-protein high-sugar (LPHS) diet, and from males of two different ages or from two different genetic backgrounds. The purpose was to determine if overlapping molecular processes are associated with father-to-offspring inheritance of phenotypes induced by the selected exposure or condition. The results show distinct transcriptional signatures in blastocysts in these various conditions. Although significant batch effects in the RNA-seq data are observed, the results appear mostly reproducible across several experiments. The authors propose a variance-based analysis to identify hypervariable genes that may drive probabilistic F1 phenotypes. They then examine long and small RNA-seq data from the testes of fathers using the same methods and find that genetic background and age have strong effects on their transcriptome, which can potentially confound the effects of environmental exposures. Overall, the results are solid, the figures are clear and have helpful illustrations and annotations and the findings are valuable for the field of intergenerational epigenetic. The manuscript however remains incomplete in some parts and present statistical analyses that can be improved.

We appreciate the reviewer providing valuable and thoughtful feedback on our manuscript and concluding that “Overall, the results are solid, the figures are clear and have helpful illustrations and annotations and the findings are valuable for the field of intergenerational epigenetics.”

Major concerns

C1: *The observed iGxE interactions in blastocysts suggest that the genetic background of the father influences the offspring' phenotypes. Based on the cited references, offspring phenotypes in the used conditions have been reported in B6 but not in FVB animals. This study does not conduct the analyses systematically in both genotypes, which is a limitation. Its relevance and usefulness would be higher if phenotypes would be examined in offspring of both genetic backgrounds.*

R1: The aim of this study was to capture molecular signatures - and hence pathways - that may respond to paternal conditioning using a highly controlled and powered experimental design. This builds on an extensive literature on intergenerational epigenetic inheritance (IEI) phenomena in mice that has been primarily described and studied in the B6 genetic background. Our goal was to build on this and include a GxE aspect by using additional strains such as FVB. Indeed, it is well-accepted that for reproductive studies and fertility, FVB strain is considered gold-standard, as the females perform well on superovulation and, when a litter is born, they exhibit high levels of maternal care for the pups. As our goal in this study was to understand the impact of paternal precondition exposure on the immediate offspring, we reasoned that starting with FVB strain, and then extending our analyses to the B6 background, would allow for the scalability of our experimental design. This allows us to obtain high sample replicate numbers (i.e. single embryos across batches) without the need for excessive animal usage, therefore adhering to the 3R principles of animal research. Moreover, evidence of intergenerational phenotypes of our exposures in FVB is actually established. A study from 2016 (PMID: 26721685) has reported the effects of paternal Low protein diet exposure and small RNA modulation on 2-cell and 4-cell stage FVB embryos, using IVF and microinjection-based approaches, providing precedent for use of this genetic background when assessing paternal contribution to developmental programming. By assessing the impact of paternal preconception exposure to nABX and LPHS as early as 4 days after fertilization in both B6 and FVB genetic backgrounds, we have a) added the missing developmental link to the previous IEI phenomena reported for B6 adult animals, and b) provided evidence of differential early embryonic responses to distinct paternal environments in the FVB genetic context, supporting further detailed characterization of post-natal outcomes, which is currently an ongoing line of research in our team, but out of scope of the study presented here.

C2: *The authors report a shift in (extra-)embryonic lineage allocation in blastocysts after paternal nABX exposure. It would be useful to examine if the results in Fig 2D-F remain valid if the analyses are performed individually for the three experimental batches mentioned in Fig 3. It would also increase the validity of this observation if the results could be confirmed with another method such as microscopy with lineage marker staining.*

R2: We observed differential expression of key lineage-specific markers *Gata4*, (primitive endoderm) and *Nanog* (epiblast) in response to nABX across multiple batches, as shown in the manuscript (**Figure 3K**). Below we further show consistent upregulated trends of critical epiblast specifier (*Tfcp2l1*) and key primitive endoderm regulator downregulation (*Fgfr2*) across batches (see also *new Fig S2*). Such trends were only prevalent for lineage markers, implying this is a specific effect.

Following the reviewer suggestion we have attempted to perform immunostaining experiments against lineage markers in entirely new cohorts of paternal exposures and IVF by staining for Sox17, Gata4 and Cdx2. Of note, we encountered technical issues with antibodies against GATA4 not giving any signal. However, we did observe a clear validation of the transcriptomic increase in *Cdx2*, which manifests through a significantly increased number of CDX2+ cells in blastocysts derived from nABX-exposed spermatozoa compared to Control embryos (new **Figure S2C**). Moreover, we were able to stain for SOX17, which was *not* recorded as a DEG. Consistently, we observed no significant change in SOX17+ cells, albeit it trended downwards. Overall, we observe relatively consistent transcriptomic changes in key lineage markers in response to nABX, which may manifest at the level of changes in absolute gene expression or changes in cell proportions. For CDX2, it appeared to be the latter.

Figure R10. Dot plots showing mean expression of marker genes of epiblast (left) and primitive endoderm (right) by batch (column) in Control versus nABX samples. This supplements batch analysis of *Gata4* and other genes in Figure 3.

C3: *The authors present an unusual variance-based RNA-seq analysis that is used to explain the probabilistic nature of the offspring phenotypes, as they did in previous studies. Although this provides an interesting interpretation of the results, the absence of observable phenotypes in blastocysts questions the biological relevance of such method. It would be necessary to validate and benchmark the variance-based analysis for instance using RNA-seq data from animals with naturally variable phenotypes such as Nnat+/-p (<https://doi.org/10.1038/s42255-022-00629-2>)*

R3: We appreciate the reviewer's comment and have now benchmarked our approach on a published dataset of epigenetically inherited phenotypes. We decided to use the 2-cell stage single-embryo SMART-Seq data generated from paternal high-fat diet (HFD) and low-fat diet (LFD) exposures due to the large dataset size and similarity to our own experimental design (PMID: 38839949). Note that the *Nnat* dataset does not have enough samples for a robust variance analysis of this nature. Tomar et al report a sexually-dimorphic mitochondrial gene-associated bimodality in HFD-derived embryos. It therefore stands to reason that HFD-derived embryos should have higher variability in their gene expression profiles when compared with LFD-derived embryos. Indeed, gene expression in HFD-derived embryos shows increased variability relative to LFD-derived embryos (**Figure R11 - top**).

We further demonstrate the power of this approach by identifying genes demonstrating increased expression variability specifically in response to paternal HFD/LFD diet. Hypervariable genes in both groups are associated with “developmental process” and “response to stimulus” Gene Ontology terms, while genes specific to LFD show an enrichment for “reproductive process” and genes specific to HFD show an enrichment for “homeostatic process” (**Figure R11 - bottom**). Because this analysis is at the 2-cell stage, which is highly transcriptionally dynamic as a function of time, we cannot distinguish whether this variance reflects time-dependent variation or paternal condition. However, it serves as a valuable independent validation of the strategy to detect high variance genes within *or* between large experimental datasets.

Figure R11. (Top row) Comparison of gene expression variability between HFD- and LFD-derived embryos in the form of a Box-violin plot (left) and grouped density plot (right). (Bottom row) Scatter plot highlighting genes showing paternal treatment-specific gene expression variability (left) and overrepresentation analysis of these gene sets (right).

C4: *GxE interactions in testis and embryo are based on independent analyses in each genetic background followed by a comparison of results e.g. overlapping and non-overlapping genes and pathways. Although useful, this approach does not properly assess statistically the interaction between paternal exposure and genetic background. For a valid and quantitative evaluation of*

interaction effects, a unified statistical model should be used and the data from both genetic backgrounds together should be analysed with a model that includes a GxE interaction term e.g. Gene expression ~ Exposure + GeneticBackground + Exposure:GeneticBackground.

R4: Genetic background is a major modifier of overall gene expression, which reflects highly divergent genomes. For example, approximately 50% of transposable element sites are different between genetic backgrounds, while precise levels of gene expression vary considerably. For this reason, integrating genetic background into a unified statistical model is driven almost exclusively by genetic background alone, with the relatively subtle effects of paternal perturbation not detected or confounded. We therefore chose to analyse each genetic background independently and then integrate the effects through overlapping DEG, GSEA and hypervariation outputs. In the new manuscript this is additionally demonstrated by interesting new insights into intergenerational GxE (e.g. *see new* Fig 4G).

C5: *The experimental design involves analyzing embryos derived from the sperm of fathers with different exposures, creating a hierarchical structure with individual embryos nested within paternal groups. Embryos from the same father are not independent thus such nested structure needs to be accounted for to avoid an underestimation of variability, false positives and an inflated data significance. The method used to take this issue into account for differential expression analyses needs to be provided.*

R5: We agree with the reviewer's comment that paternal group can be a strong contributor to embryonic variability and would thereby influence the results of our analyses. For this reason, our experimental design considers embryos from each father as an independent batch. Thus, our batch-specific analysis inherently takes paternal variability into account, since each batch is a father (**Figure 3**). As an alternative analysis strategy during data analysis optimisation, we also attempted to include the father of origin in the DESeq2 design formula for differential expression analysis as suggested. However, correcting for batch using the DESeq2 design formula results in a DE gene set that is heavily biased towards Batch 1 (**Figure R12**). This highly skewed weighting led to numerous spurious results, possibly because of challenges in parsing subtle effects with variable (single-embryo) datasets. For this reason, we opted to analyse blastocysts of each batch individually and then integrate the results of each differential expression analyses in order to obtain batch-robust DE genes (broDEGs; **Figure 3**) that produced a consistent effect in response to paternal conditioning.

Figure R12. Upset plots showing overlap of DEGs identified using DESeq’s built-in batch-correction (“BatchTermIncluded_DESeq”) as compared with DEGs identified in each batch individually, starting from nABX downregulated genes (top left), nABX upregulated genes (top right), LPHS downregulated genes (bottom left) and LPHS upregulated genes (bottom right).

Minor concerns

6. **p5:** *The authors state that "The direct physiological response of males was similar between the FVB and B6 backgrounds, with negligible impact on overt phenotypes and bodyweights, but opposite and significant differences in caecum weight"? However, a comparison of Fig. S1A and S4A suggests a strain-specific difference: FVB males have no significant change in body weight but B6 males have significant weight reduction (~10% mean difference) in response to both exposures. Additionally, data supporting the claim of opposite differences in cecum weight between FVB and B6 cannot be found in the manuscript. Comparing Fig. 1C and S4C, the effects of exposure appear to be consistent across strains. This needs to be clarified and the discrepancies explained.*

We have now corrected this point in the text of the revised manuscript, for clarity and precision. It is important to note however, that the B6 response to nABX data does not indicate a reduction despite statistical significance. This is because - due to stochastic experimental variation - at $t=0$ the weight of the B6 males randomly selected to be exposed to nABX was lower compared to Control males. This represents experimental chance during the random separation of wildtype littermates into experimental groups. During the treatment period of 6 weeks, nABX males gain weight proportionally to their starting values, and therefore the dynamics of weight-gain over time, expressed as percentage of starting weight, is comparable (actually slightly higher) than controls. This is in line with modestly increased rate of weight gain upon nABX-exposure in FVB, and previous studies showing nABX leads to marginally elevated weight gain. In contrast, LPHS exposure in B6 leads to a slight weight loss relative to Control. Given that this group randomly started at a similar (albeit slightly lower) weight than Control this likely represents a true effect. Indeed, it is also consistent with a marginal weight loss in FVB. Thus, B6 weight change trends are consistent with FVB, but due to the slightly different $t=0$ starting weights we cannot make robust conclusions on this. In line with the reviewers point, we have modified the text so that it does not claim negligible impact on overt phenotypes and body weights, and then specify that the nABX and LPHS consumption causes significant differences in caecum weight with opposite vector directionality (nABX results in increased, and LPHS in decreased caecum weight), a response which is shared between the two mouse strains analyzed.

7. **p6:** *The authors state that "PCA on single blastocyst transcriptome revealed a distinction between embryos sired by mature or young males along the second principal component". However, the visual differences in the PCA plot are minimal, with no clear separation or clustering based on paternal age. The use of the term "distinction" seems overstated. Proper statistical analyses need to be conducted to quantify the effect of paternal age on PC2.*

We strongly agree with the reviewer that there is no clear separation or clustering of samples based on PCA, hence we avoided using these terms. Nevertheless, we believe there is a visual distinction between the distribution of young vs mature sired blastocysts along PC2. This is quantified in the PCA histogram, which reveals an apparent bimodal distribution. PC1 acts as a control. Moreover, there is a strong DEG signature in mature-derived blastocysts, stemming from hundreds of sample-replicates, which serves as strong statistical evidence. Furthermore, we now show that the effect is recapitulated in mature-sired blastocysts from FVB males, with considerable overlap in responding pathways, and which also show a distinction in PCA.

8. **p17:** *The methods section "Profiling binding of transcription factors in our DE gene sets" lacks details. The description and available code do not clearly explain how the analysis was conducted.*

While Enrichr performs enrichment analyses across multiple databases, which databases were used is not indicated. The methodology should also be clarified to ensure reproducibility.

In the revised Manuscript, we have now detailed “Binding of transcription factors to DEG sets identified in nABX- or LPHS- derived blastocysts was performed using the ChEA module (2022) in the Enrichr tool (Kuleshov *et al*, 2016). This tests for enrichment of transcription factor occupancy in genesets, based on direct ChIP measurement across a large database of cell and tissue types. DEG sets were analysed under default parameters and visualised using the appyter and the volcano plot option.”

9. **p3:** *The sentence "we noted a significant decrease in the size of blastocysts derived from LPHS-exposed males, pointing at a potential developmental delay" needs a reference to support this observation.*

A previous study (PMID: 34644528) has shown that reduced cell proliferation speed (i.e. developmental delay) in the preimplantation embryo results in a decrease in blastocyst size at the time of implantation, which can adversely affect pregnancy outcomes of smaller embryos. However, as we detail in **R4** (above), the observed smaller LPHS-blastocyst size could be a result of developmental delay or changes in the cell and/or blastocoel volume, or a combination of these factors. We cannot quantitatively determine the relative contribution of these possibilities in our data, although our new cell-count analyses (**Figure S2C**) indicate that paternal preconception exposure to LPHS did not significantly alter the number of cells per blastocyst, compared to control conditions. We have therefore modified the test to read “*pointing to a potential growth defect*”, which captures a broader array of possibilities and more accurately represents our results.

10. **For Fig. II,** *labeling the clusters directly in the figure would improve clarity and make it easier to follow the corresponding explanation in the text.*

Thank you for this helpful suggestion, we have now appropriately labelled the clusters directly in the figure.

11. **p7:** *The authors refer to sequencing "sperm-borne small RNA." Given that small RNAs traffic from the epididymis to sperm cells, it is unclear how it was determined that the observed small RNAs are sperm-borne. This needs to be clarified and evidence to support the claim needs to be provided.*

We believe that there is a language misunderstanding at play here that has recently taken root in the IEI field. The term ‘sperm-borne’ signifies that these small RNAs are carried by sperm, where they were isolated from. The word ‘borne’ is the past participle of the verb ‘bear’ and typically means to be carried by. It does not refer to where RNA are ‘born’. We do not know, nor do we claim, that these RNAs originate in the testes during sperm development nor that they are delivered to the spermatozoa as they mature in the epididymis. Rather our language simply means that the small RNA are carried (or present) in mature sperm and therefore likely delivered to the oocyte during fertilisation.

12. **The description of IVF** *mentions an "oocyte pool." What such pool means needs to be clarified as this suggests that all oocytes from different mothers are mixed at the start of the IVF procedure. Then whether embryos can be traced back to the oocytes of individual mothers or not needs to be indicated.*

The reviewer is correct in their interpretation that an ‘oocyte-pool’ represents all oocytes from different females which are mixed and then randomly distributed at the start of the IVF procedure. Therefore, the resulting embryos cannot be traced back to individual females. The text of the manuscript states: ‘ (...) and IVF was performed from a **randomised common pool of strain-matched oocytes** for all sperm donors’, and later: ‘Batches were designed such that all IVF and embryo culture stemming from differentially exposed sperm donors occurs in parallel, using a **shared oocyte pool**, to minimise intra-batch confounders’.

Dear Jamie,

Thank you for submitting a revised version of your manuscript. I sincerely apologise for the protracted assessment process for your manuscript due to the delay in reviewer report submission.

We have now received input from both original reviewers, who are generally satisfied with the revisions. Therefore, I will be happy to accept the manuscript for publication after incorporation of textual edits in response to the remaining comments by reviewer #2.

Additionally, there are a few editorial points that need addressing before I can extend official acceptance of the manuscript:

1. Please submit up to five keywords.
2. Please rename supplementary figures into "Figure EV1" etc. throughout the manuscript.
3. CRedit has replaced the traditional author contributions section because it offers a systematic, machine-readable author contributions format that allows for more effective research assessment. Please remove the Authors Contributions from the manuscript and use the free text boxes beneath each contributing author's name in our online submission system to add specific details on the author's contribution. More information is available in our guide to authors.
4. Please rename "Competing interest statement" section into "Disclosure and competing interests statement" (further info: <https://www.embopress.org/page/journal/14602075/authorguide#conflictsofinterest>).
5. Please move Methods after Discussion.
6. Please move the Data Availability section after Methods and add the content from the Code Availability section. Please also include resolvable links to the individual datasets. More information about the format of this section can be found here: <https://www.embopress.org/page/journal/14602075/authorguide#dataavailability>.
7. All Materials and Methods need to be described in the main text using our 'Structured Methods' format. According to this format, the Methods section includes a Reagents and Tools Table (listing key reagents, experimental models, software and relevant equipment and including their sources and relevant identifiers) followed by a Methods and Protocols section describing the methods, ideally using a step-by-step protocol format. The aim is to facilitate adoption of the methodologies across labs. Please download and fill our Reagents and Tools Table template (.docx), which you can find in our author guidelines: <https://www.embopress.org/page/journal/14602075/authorguide#structuredmethods>. When submitting your revised manuscript, please do not include the Reagents and Tools Table in the Methods section of the manuscript but upload it as a separate file choosing the file type "Reagent Table". An example of a Method paper with Structured Methods can be found here: <https://www.embopress.org/doi/10.15252/msb.20178071>.
8. Please rename Tables S1, 4, 10, 11, 12, 13, 15, 18 into Dataset EV1 - EV8 and upload each dataset as a single file. The remaining tables should be renamed Table EV1 etc. and also uploaded as separate files. Please remove the legends from the manuscript text file and add to the corresponding table at the top of the page.
9. Please check whether the figures are mentioned in the text in a sequential order. Currently, Figure 5 is called out before Figure 4C.
10. Tables S15, S16 and S19 are not mentioned in the manuscript text; please add the corresponding callouts.
11. Our data editors have flagged the following issues in figure legends that need correcting:
 - Please define the annotated p values ****/****/**/* as well as provide the exact p-values for the same in the legend of figure S2 C as appropriate.
 - Please provide the exact p values in the legends of figures 1C, 5E, S1C, S2 A, S4 A, C.
 - Please indicate the statistical test used for data analysis in the legends of figures 1G, H, I, J; 2G, H; 4D, F, G; 5C, D; 6E, S1 N, S2 C, S6A, D, E, F.
 - Please define the box plots in terms of minima, maxima, centre, bounds of box and whiskers, and percentile in the legends of figures 1E, 2G, H; 5B, E; S1B, G, H; S2 C, S5 E.
 - Please define the box plots in terms of bounds of box and percentile in the legends of figures 1B, C.
 - Please provide information on the number and nature of replicates in the legends of figures 1D, E, G, H, J; 2D-F, G, H; 3E, K; 4D, F; 5B, E; 6E; S1G, H; S2 A, S3 A, S4F, S5 E, S6A, D, E, F.
 - Please define the error bars in the legends of figures 1D; 2D-F; 3K, S2 A, S3 A, S4 A.
 - Please define the measure of center for the error bars in the legends of figures S1C, S4 B-D.
12. Papers published in The EMBO Journal are accompanied online by a 'Synopsis' to enhance discoverability of the manuscript. It consists of A) a short (1-2 sentences) summary of the findings and their significance, B) 3-4 bullet points highlighting key results and C) a synopsis image that is 550x300-600 pixels large (width x height, jpeg or png format). You can either show a model or key data in the synopsis image. Please note that the image size is rather small and that text needs to be readable at the final size.

With best wishes,

leva

leva Gailite, PhD
Senior Scientific Editor
The EMBO Journal
Meyerhofstrasse 1
D-69117 Heidelberg
Tel: +4962218891309
i.gailite@embojournal.org

We realize that it is difficult to revise to a specific deadline. In the interest of protecting the conceptual advance provided by the work, we recommend a revision within 3 months (10th Sep 2025). Please discuss the revision progress ahead of this time with the editor if you require more time to complete the revisions.

Referee #1:

The authors have carefully addressed my concerns raised previously. The manuscript has been considerably improved. It is an important contribution to the field of IEI. I consider the manuscript now suitable for publication in the EMBO journal.

Referee #2:

The authors have improved the manuscript and their revisions make the data and the presentation stronger. A few minor points remain to be addressed for more clarity and to avoid potential misunderstanding.

1) The benchmarking of the proposed variability measure effectively highlights the strength of the approach. Selecting the dataset with 2-cell embryos showing paternal exposure-induced variability was a good choice. However, in their response to Reviewer 1's minor comment 13 and our comment R3, the authors state that certain datasets do not have enough samples for a robust variance analysis. A clarification of the basis for this claim is needed. What criteria or statistical considerations define an insufficient sample size in this context? What do the authors consider as a minimum acceptable sample size for reliable variance estimation?

2) In response to Reviewer 1 R5, the authors state that "each batch is a father." This is not clearly explained in the manuscript and rather it seems that a batch includes embryos from multiple fathers, and that batch-specific analyses are intended to address technical variability between batches rather than biological variability between exposed fathers. This should be clarified.

3) Unclear statement on page 6: "For example, nABX-induced dysbiosis was associated with geneset enrichment for 'steroidogenesis' and 'fatty acid metabolic' genes in FVB testes, consistent with altered lipid profiles in testis of nABX-treated males". Should this refer to lipid profiles in LPHS-treated males instead? Or which specific lipid profiles in Fig. 6A-B are being referred to here?

4) As per reviewer 1 minor comment 17, there are overstatements that need to be toned down:

a. On page 3, the authors define "high confidence DEGs" as DEGs with adjusted $p < 0.05$ and $\log_2FC > 0.5$. These thresholds are standard for differential expression analyses and are not particularly "high confidence". Given the inherent noise in early embryonic gene expression, simply "DEGs" is more appropriate.

b. On page 4, the authors describe certain genes as being dysregulated across multiple batches, referring to *Gata4* as "systematically downregulated" and *Nanog* as "consistently dysregulated." However, only 2 out of 3 batches show significant changes for these genes, making the statements misleading especially since these genes are presented with three others (*Tat*, *Rac1*, *Mapk1*) which do show significant changes across all three batches.

c. The same paragraph mentions *Lrrc34* and *Prdm14*, but plots showing their expression across the 3 batches could not be found. Conversely, batch-specific expression plots for *Tfcp2l1* and *Fgfr2* are included in Fig. S3A, yet these genes are not discussed in the above-mentioned paragraph. Please double-check the gene references and associated plots.

Response to Reviewers – second revision

Dura, Ranjan et al.,

We thank the reviewers for re-assessing our revised manuscript, and would like to note the constructive feedback from both reviewers throughout the process. Below we address the remaining comments from reviewer #2, which improve the clarity of the study.

C1: The benchmarking of the proposed variability measure effectively highlights the strength of the approach. Selecting the dataset with 2-cell embryos showing paternal exposure-induced variability was a good choice. However, in their response to Reviewer 1's minor comment 13 and our comment R3, the authors state that certain datasets do not have enough samples for a robust variance analysis. A clarification of the basis for this claim is needed. What criteria or statistical considerations define an insufficient sample size in this context? What do the authors consider as a minimum acceptable sample size for reliable variance estimation?

R1: To assess inter-sample variability, our minimum acceptable threshold is $N=15$ per condition, which is based on foundational work studying technical noise in single-cell analysis (Brennecke et al., 2013). This work implies that 13 transcriptomic profiles were too few to substantiate claims of biological insight from highly variable genes. The threshold is further supported by a recent study on murine sample sizes for bulk transcriptomics, which claimed that a minimum $N=8-12$ per condition is needed to reduce the likelihood of false positive discovery (Halasz et al., 2024). The datasets proposed by the reviewer were significantly below this threshold and therefore not appropriate for variance analysis. Nevertheless, we fully agree with the reviewer's principle that the method should be benchmarked and proceeded to do so using alternative datasets that had sufficient sample size, which validated the strategy.

C2: In response to Reviewer 1 R5, the authors state that "each batch is a father." This is not clearly explained in the manuscript and rather it seems that a batch includes embryos from multiple fathers, and that batch-specific analyses are intended to address technical variability between batches rather than biological variability between exposed fathers. This should be clarified.

R2: We thank the reviewer for pointing this out, and have updated the manuscript text to clarify this point by specifically stating the number of fathers in each batch. This supplements the methods section, which also makes this point.

C3: Unclear statement on page 6: "For example, nABX-induced dysbiosis was associated with geneset enrichment for 'steroidogenesis' and 'fatty acid metabolic' genes in FVB testes, consistent with altered lipid profiles in testis of nABX-treated males". Should this refer to lipid profiles in LPHS-treated males instead? Or which specific lipid profiles in Fig. 6A-B are being referred to here?

R3: We thank the reviewer for this comment. This statement refers to altered lipid profiles observed in testes of nABX-treated males in our recently published manuscript on the effect of paternal gut dysbiosis on offspring fitness (Argaw-Denboba et al., 2024). We have now updated the manuscript text with the citation.

C4: As per reviewer 1 minor comment 17, there are overstatements that need to be downtoned:

C4a. On page 3, the authors define "high confidence DEGs" as DEGs with adjusted $p < 0.05$ and $\log_2\text{FC} > 0.5$. These thresholds are standard for differential expression analyses and are not particularly "high confidence". Given the inherent noise in early embryonic gene expression, simply "DEGs" is more appropriate.

R4a. The commonly accepted definition of a significant differentially expressed gene (DEG) in transcriptomics is that it must reach statistical significance ($p < 0.05$) after multiple correction (adj $p < 0.05$), typically using a negative binomial distribution-based method, such as DESeq2 or EdgeR. Optionally, a \log_2 fold-change (LFC) threshold is additionally applied to filter for higher effect-size responses amongst significant genes. We chose to focus on the latter option to increase the confidence in DEGs by guaranteeing an effect size $> \log_2 0.5$ across F1 replicate numbers that reached $n > 50$ per paternal condition (i.e. robust). We believe this additional filter should be distinguished from the baseline definition that is based only on adjusted statistical significance. Indeed consistently, applying this filter leads to only the top 17% (nABX) and 33% (LPHS) of statistically significant DEGs remaining, supporting that they are indeed the 'high-confidence' subset.

C4b. On page 4, the authors describe certain genes as being dysregulated across multiple batches, referring to *Gata4* as "systematically downregulated" and *Nanog* as "consistently dysregulated." However, only 2 out of 3 batches show significant changes for these genes, making the statements misleading especially since these genes are presented with three others (*Tat*, *Rac1*, *Mapk1*) which do show significant changes across all three batches.

R4b. All of the genes presented are 'consistently' or 'systematically' downregulated across independent batches, as judged by our stated definition of broDEGs reaching statistical significance in 2/3 batches with $\text{LFC} > 0.5$ (see methods and text). While most of the noted genes also reach statistical significance across all three batches, two genes - *Gata4* and *Nanog* - only reach significance in 2/3 batches. In the third batch they reach $p = 0.058$ and 0.73 . Thus, all genes confirm to the definition. However, because some still did not reach significance in all three batches but rather trended, we did not refer to them as 'significant' but as 'consistent' to distinguish statistical significance from trend. To address the reviewer point directly, we have now removed the word 'consistently' from the text. More generally however, we believe that these genes together constitute a signature of differential gene expression between the conditions given that they collectively fall into coherent pathways of 'epiblast' and 'primitive endoderm' regulators, and considering the inherent noise in embryonic transcriptomes.

C4c. The same paragraph mentions *Lrrc34* and *Prdm14*, but plots showing their expression across the 3 batches could not be found. Conversely, batch-specific expression plots for *Tfcp2l1* and *Fgfr2* are included in Fig. S3A, yet these genes are not discussed in the above-mentioned paragraph. Please double-check the gene references and associated plots.

R4c. Thank you for spotting this error. *Tfcp2l1* and *Fgfr2* are now correctly called out in the text.

Dear Jamie,

Thank you for addressing the final editorial requests. I am now happy to inform you that your manuscript has been accepted for publication. Congratulations with a great study!

Before we forward your manuscript to our publishers, I would like to propose some edits in the manuscript abstract and synopsis. I have also written a short blurb that will accompany the title of your manuscript in our online system. Please take a look at these edits in the attached text file and let me know if any corrections or adjustments are needed.

If you have any questions, please do not hesitate to contact the Editorial Office. Thank you for this wonderful contribution to The EMBO Journal!

With best wishes,

Ieva

Ieva Gailite, PhD
Senior Scientific Editor
The EMBO Journal
Meyerohofstrasse 1
D-69117 Heidelberg
Tel: +4962218891309
i.gailite@embojournal.org
